# Applying causal discovery to single-cell analyses using CausalCell

Yujian Wen[1†], Jielong Huang[1†], Shuhui Guo[1], Yehezqel Elyahu[2], Alon Monsonego[2], Hai Zhang[3]*, Yanqing Ding[4]*, Hao Zhu[1,5,6]*

[1]Bioinformatics Section, School of Basic Medical Sciences, Southern Medical University, Guangzhou, China; [2]The Shraga Segal Department of Microbiology, Immunology and Genetics, Faculty of Health Sciences, Ben-Gurion University of the Negev, Beer-Sheva, Israel; [3]Network Center, Southern Medical University, Guangzhou, China; [4]Department of Pathology, School of Basic Medical Sciences, Southern Medical University, Guangzhou, China; [5]Guangdong-Hong Kong-Macao Greater Bay Area Center for Brain Science and Brain-Inspired Intelligence, Southern Medical University, Guangzhou, China; [6]Guangdong Provincial Key Lab of Single Cell Technology and Application, Southern Medical University, Guangzhou, China

**\*For correspondence:**
zhangh@smu.edu.cn (HZ);
dyqgz@126.com (YD);
zhuhao@smu.edu.cn (HZ)

†These authors contributed equally to this work

**Abstract** Correlation between objects is prone to occur coincidentally, and exploring correlation or association in most situations does not answer scientific questions rich in causality. Causal discovery (also called causal inference) infers causal interactions between objects from observational data. Reported causal discovery methods and single-cell datasets make applying causal discovery to single cells a promising direction. However, evaluating and choosing causal discovery methods and developing and performing proper workflow remain challenges. We report the workflow and platform CausalCell (http://www.gaemons.net/causalcell/causalDiscovery/) for performing single-cell causal discovery. The workflow/platform is developed upon benchmarking four kinds of causal discovery methods and is examined by analyzing multiple single-cell RNA-sequencing (scRNA-seq) datasets. Our results suggest that different situations need different methods and the constraint-based PC algorithm with kernel-based conditional independence tests work best in most situations. Related issues are discussed and tips for best practices are given. Inferred causal interactions in single cells provide valuable clues for investigating molecular interactions and gene regulations, identifying critical diagnostic and therapeutic targets, and designing experimental and clinical interventions.

## Editor's evaluation

This manuscript presents an important tool for causal inference intended for the analysis of single cell datasets but possibly with broader applications. It compares several algorithms and incorporates a number of them in the platform and offers convincing evidence of its usefulness. With the rapid expansion of large datasets, this tool is beneficial in offering several causal inference analysis options and expediting the interpretation of data.

## Introduction

RNA-sequencing (RNA-seq) has been used to detect gene expression in a lump of cells for years. Many statistical methods have been developed to explore correlation/association between transcripts in RNA-seq data, including the 'weighted gene co-expression network analysis' that infers networks of correlated genes (*Joehanes, 2018*). Since a piece of tissue may contain many different cells and the

sample sizes of most RNA-seq data are <100, causal interactions in single cells, which to a great extent are emergent events (*Bhalla and Iyengar, 1999*), cannot be revealed by these statistical methods. Averaged gene expression in heterogeneous cells also makes causal interactions blurred or undetectable. Except for some annotated interactions in signaling pathways, most causal interactions in specific cells remain unknown (e.g. in developing cells undergoing rapid fate determination and in diseased cells expressing genes aberrantly).

Single-cell RNA-sequencing (scRNA-seq) has been widely used to detect gene expression in single cells, providing large samples for analyzing cell-specific gene expression and regulation. On statistical data analysis, it is argued that '*statistics alone cannot tell which is the cause and which is the effect*' (*Pearl and Mackenzie, 2019*). Corresponding to this, causal discovery is a science that distinguishes between causes and effects and infers causal interactions from observational data. Many methods have been designed to infer causal interactions from observational data. For single-cell analysis, any method faces the three challenges – high-dimensional data, data with missing values, and inferring with incomplete model (with missing variables). The constraint-based methods are a class of causal discovery methods (*Glymour et al., 2019*; *Yuan and Shou, 2022*), and the PC algorithm is a classic constraint-based method. Testing conditional independence (CI, CI≠unconditional independence [UI]≠uncorrelation) between variables is at the heart of constraint-based methods. Many CI tests have been developed (*Verbyla, 2018*; *Zhang and Peters, 2011*), from the fast GaussCItest to the time-consuming kernel-based CI tests. GaussCItest is based upon partial correlations between variables. Kernel-based CI tests estimate the dependence between variables upon their observations without assuming any relationship between variables or distribution of data. These features of kernel-based CI tests enable relationships between any genes and molecules, not just transcription factors (TFs) and their targets, to be inferred. Thus, CI tests critically characterize constraint-based causal discovery and distinguish causal discovery from other network inferences, including 'regulatory network inference' (*Nguyen et al., 2021*; *Pratapa et al., 2020*), 'causal network inference' (*Lu et al., 2021*), 'network inference' (*Deshpande et al., 2019*), and 'gene network inference' (*Marbach et al., 2012*).

Kernel-based CI tests are highly time-consuming and thus infeasible for transcriptome-wide causal discovery. Recently other causal discovery methods are reported, especially continuous optimization-based methods (*Bello et al., 2022a*; *Zheng et al., 2018*). Thus, identifying the best methods and CI tests, developing reasonable workflows, developing measures for quality control, and making trade-offs between time consumption, network size, and network accuracy are important. This *Tools and Resources* article addresses the above issues by benchmarking multiple causal discovery methods and CI tests, applying causal discovery to multiple scRNA-seq datasets, developing a causal discovery workflow/platform (called CausalCell), and summarizing tips for best practices. Specifically, the workflow combines feature selection and causal discovery. The benchmarking includes 11 causal discovery methods, 10 CI tests, and 9 feature selection algorithms. In addition, measures for estimating and ensuring the reliability of causal discovery are developed. Our results indicate that when relationships between variables are free of missing variables and missing values, continuous optimization-based methods perform well. Otherwise, the PC algorithm with kernel-based CI tests can better tolerate incomplete models and missing values. Inferred relationships between gene products help researchers draw causal hypotheses and design experimental studies. The remaining sections describe the workflow/platform and data analysis examples, discuss specific issues, and present tips for best practices. The details of methods and algorithms, benchmarking results, and data analysis results are described in appendix files.

## Materials and methods
### Features of different algorithms

Causal discovery cannot be performed transcriptome-wide due to time consumption and the power of methods. A way to choose a subset of genes based on one or several genes of interest is feature selection. A feature selection algorithm combines a search technique and an evaluation measure and works upon one or several response variables (i.e. genes of interest). After obtaining a measure between the response variable(s) and each feature (i.e. variable, gene), a subset of features most related to the response variable(s) are extracted from the whole dataset. Using simulated data and real scRNA-seq data (*Appendix 1—table 1*), we benchmarked nine feature selection algorithms. The

**Table 1.** Performance of feature selection methods.

| Algorithm | Category | Time consumption | Accuracy | Scalability | Advantage/disadvantage |
|---|---|---|---|---|---|
| RandomForest | Ensemble learning-based methods use many trees of a random forest to calculate the importance of features, then perform regression based on the response variable(s) to identify the most relevant features. | + | ++ | ++ | These algorithms are indeterministic (the same input may generate slightly different outputs). ExtraTrees and RandomForest perform better than XGBoost. |
| ExtraTrees | | + | ++ | ++ | |
| XGBoost | | ++ | + | + | |
| BAHSIC | The three are Hilbert-Schmidt independence criterion (HSIC)-based algorithms. HSIC is used as the measure of dependency between the response variable and features. | + | +++ | + | BAHSIC and SHS are the best and second best. |
| SHS | | + | +++ | + | |
| HSIC Lasso | | ++ | ++ | ++ | Inferior to BAHSIC and SHS. |
| Lasso | Lasso is a regression analysis method that performs both variable selection and regularization (which adds additional constraints or penalties to a regression model). Lasso, RidgeRegression, and ElasticNet are three regulation terms. | +++ | + | +++ | Inferior to BAHSIC and SHS. Accuracy is not high and scalability is poor. |
| RidgeRegression | | +++ | + | +++ | |
| ElasticNet | | +++ | + | +++ | |

\# Time consumption is estimated upon simulated data (**Appendix 2—figure 1**). Accuracy is estimated upon simulated and real data (**Appendix 2—figures 2–7**). Scalability is estimated upon simulated data (**Appendix 2—figure 2**). Advantage/disadvantage is made upon accuracy together with algorithms' other properties.

properties and advantages/disadvantages of these algorithms are summarized, with '+++' and '+' indicating the most and least recommended ones (**Table 1**; **Appendix 2—figures 1–7**).

Many causal discovery methods have been proposed. Constraint-based causal discovery identifies causal relationships between a set of variables in two steps: skeleton estimation (determining the skeleton of the causal network) and orientation (determining the direction of edges in the causal network). The PC algorithm is a classic and widely recognized algorithm (**Glymour et al., 2019**). Causal discovery using the PC algorithm is different in that PC can work with different CI tests to perform the first step. We combined the PC algorithm with 10 CI tests to form 10 constraint-based causal discovery algorithms. The properties and advantages/disadvantages of the 10 algorithms are summarized, with '+++' and '+' indicating the most and least recommended ones (**Table 2**; **Appendix 3—figures 1 and 2**). In addition to constraint-based methods, there are other kinds of methods, including score-based methods that assign a score function to each directed acyclic graph (DAG) and optimize the score via

**Table 2.** Performance of causal discovery methods.

| Methods | CI tests | Category | Time consumption | Accuracy | Stability | Features |
|---|---|---|---|---|---|---|
| | GaussCItest | GaussCItest assumes all variables are multivariate Gaussian, which impairs GussCItest's performance when data are complex. | +++ | + | +++ | Fast and inaccurate |
| | CMIknn | Conditional mutual information (CMI) is based on mutual information. | +++ | ++ | + | Fast and inaccurate |
| | RCIT | | ++ | ++ | ++ | Fast and moderately accurate |
| | RCoT | Two approximation methods of KCIT (the Kernel conditional independence test). | ++ | ++ | ++ | |
| | HSIC.clust | | + | ++ | + | |
| | HSIC.gamma | Extra transformations make HSIC determine if $X$ and $Y$ are conditionally independent given a conditioning set. HSIC.gamma and HSIC.perm employ gamma test and permutation test to estimate a p-value. | + | +++ | ++ | Slow and accurate |
| | HSIC.perm | | + | +++ | + | |
| | DCC.gamma | Distance covariance is an alternative to HSIC for measuring independence. DCC.gamma and DCC.perm employ gamma test and permutation test to estimate a p-value. | + | +++ | ++ | |
| | DCC.perm | | + | +++ | + | Slow and accurate |
| PC GSP | GCM | The generalized covariance measure-based (also classified as regression-based). | + | ++ | +++ | Slower than DCC.gamma |
| GES | | Score-based causal. | ++ | ++ | ++ | Fast and moderately accurate |
| DAGMA-nonlinear | | Continuous optimization-based. | + | ++ | +++ | Performs well with complete models |

\# Time consumption is estimated upon simulated data (**Appendix 3—figure 1**). Accuracy is estimated upon the lung cancer cell lines (**Figure 2**; **Appendix 3—figure 2**). Stability is estimated upon the relative structural Hamming distance (SHD, a standard distance to compare graphs by their adjacency matrix), which is used to measure the extent an algorithm produces the same results when running multiple times (**Appendix 3—table 1**). Advantage/disadvantage is made upon accuracy.

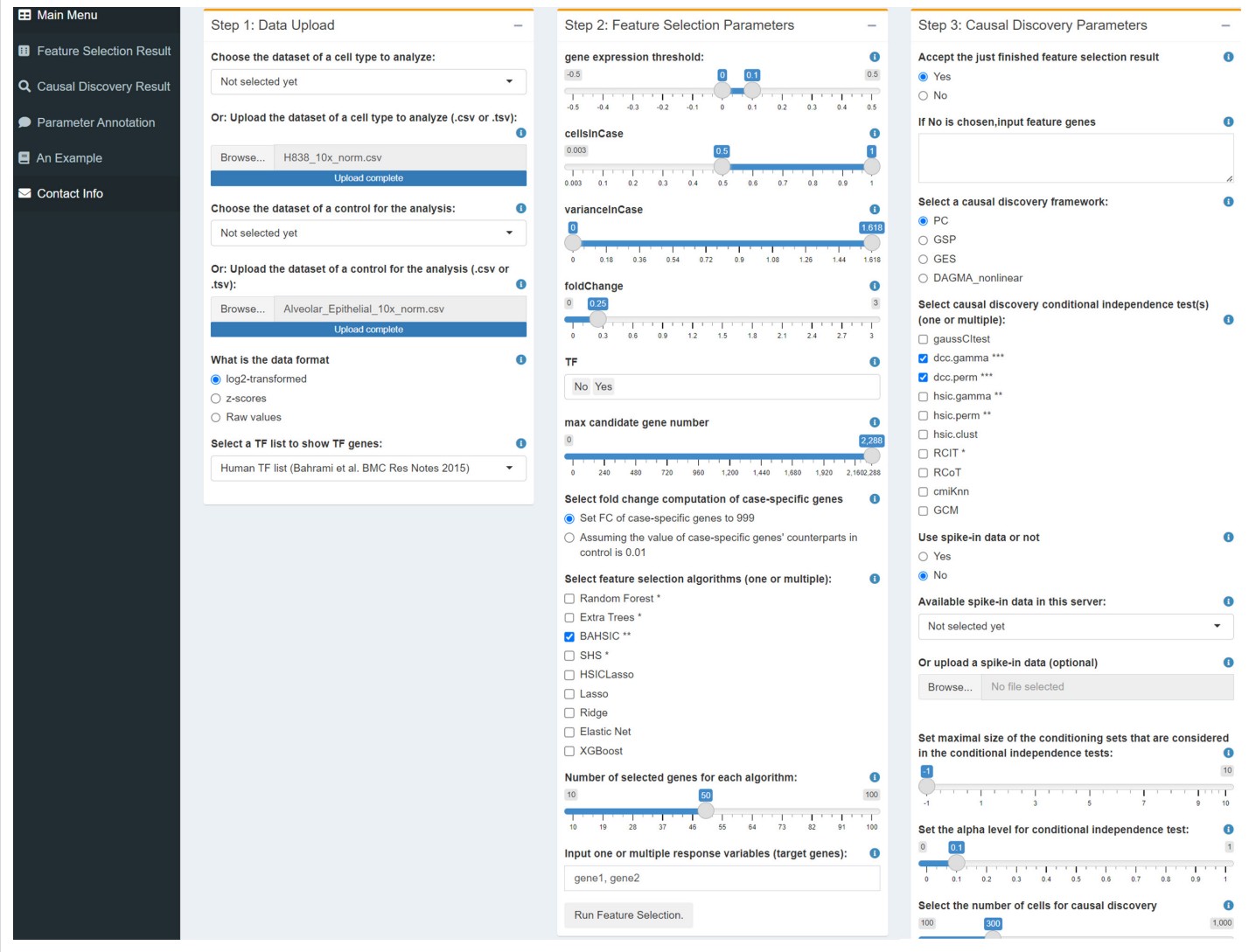

**Figure 1.** The user interface of CausalCell. Multiple algorithms and functions are integrated and implemented to facilitate and compare feature selection and causal discovery.

greedy searches (*Chickering, 2003*), hybrid methods that combine score-based and constraint-based methods (*Solus et al., 2021*), and continuous optimization-based methods that convert the traditional combinatorial optimization problem into a continuous program (*Bello et al., 2022a*; *Zheng et al., 2018*). When benchmarking the four classes of methods, multiple simulated data, real scRNA-seq data, and signaling pathways were used to evaluate their performance (*Appendix 1—table 1*).

The results of benchmarking the 11 causal discovery methods and 10 CI tests show that when causal discovery is without the problems of incomplete models (i.e. ones that miss nodes or edges from the data-generating model) and missing values, nonlinear versions of continuous optimization-based methods (especially DAGMA-nonlinear) perform better than others (*Bello et al., 2022a*). When causal discovery is applied to a set of highly expressed or differentially expressed genes in an scRNA-seq dataset (which has both missing variables and missing values), the PC algorithm with kernel-based CI tests (especially DCC.gamma) performs well. Therefore, the CausalCell platform includes 4 causal discovery methods (PC, GES, GSP, and DAGMA-nonlinear) to suit different data, together with 10 CI tests and 9 feature selection algorithms.

## Developing the workflow/platform for causal discovery

The CausalCell workflow/platform is implemented using the Docker technique and Shiny language and consists of feature selection, causal discovery, and several auxiliary functions (*Figure 1*). A parallel version of the PC algorithm is used to realize parallel multi-task causal discovery (*Le et al., 2019*). In addition, the platform also includes the GES, GSP, and DAGMA-nonlinear methods. PC and GSP can work with 10 CI tests. Annotations of functions and parameters and the detailed description of a causal discovery process are available online.

## Data input and pre-processing

scRNA-seq and proteomics data generated by different protocols or methods (e.g. 10x Genomics, Smart-seq2, and flow cytometry) can be analyzed. CausalCell accepts log2-transformed data and *z*-score data and can turn raw data into either of the two forms. A dataset (i.e. the 'case') can be analyzed with or without a control dataset (i.e. the 'control'). Researchers often identify and analyze special genes, such as highly expressed or differentially expressed genes. For each gene in a case and control, three attributes (the averaged expression value, percentage of expressed cells, and variance) are computed. Fold changes of gene expression are also computed (using the *FindMarkers* function in the *Seurat* package) if a control is uploaded. Genes can be ordered upon any attribute and filtered upon a combination of five conditions (i.e. expression value, percentage of expressed cells, variance, fold change, and being a TF or not). Since performing feature selection transcriptome-wide is unreliable due to too many genes, filtering genes before feature selection is necessary, and different filtering conditions generate different candidates for feature selection.

Batch effects may influence identifying differentially expressed genes. Since removing batch effects should be performed with raw data before integrating batches and there are varied batch effect removal methods (*Tran et al., 2020*), it should be performed by the user if necessary.

## Feature selection

Feature selection selects a set of genes (i.e. features) from the candidate genes upon one or multiple genes of interest (i.e. response variables). As above-mentioned, candidate genes are extracted from the whole dataset upon specific conditions because performing feature selection transcriptome-wide is unreliable. Based on the accuracy, time consumption, and scalability of the nine feature selection algorithms (*Table 1*), BAHSIC is the most recommended algorithm. The joint use of two kinds of algorithms (e.g. Random Forest+BAHSIC) is also recommended to ensure reliability. Feature genes are usually 50–70, but the number also depends on the causal discovery algorithms. Genes can be manually added to or removed from the result of feature selection (i.e., the feature gene list) to address a biological question specifically. The input for causal discovery can also be manually selected without performing feature selection; for example, the user can examine a specific Gene ontology (GO) term.

## Causal discovery

The PC and GSP algorithms can work with the 10 CI tests to provide varied options for causal discovery. In the inferred causal networks, direction of edges is determined by the meek rules (*Meek, 1997*), and each edge has a sign indicating activation or repression and a thickness indicating CI test's statistical significance. The sign of an edge from A to B is determined by computing a Pearson correlation coefficient between A and B, which is 'repression' if the coefficient is negative or 'activation' if the coefficient is positive. In most situations, 'A activating B' and 'A repressing B' correspond to up-regulated A in the case dataset, with up- and down-regulated B in the case dataset compared with in the control dataset.

There are two ways to construct a consensus network that is statistically more reliable. One way is to run multiple algorithms (i.e. multiple CI tests) and take the intersection of some or all inferred networks as the consensus network (*Figure 2*). The other is to run an algorithm multiple times and take the intersection of all inferred networks as the consensus network (*Figure 3*).

If a scRNA-seq dataset is large, a subset of cells should be sampled to avoid excessive time consumption. We suggest that 300 and 600 cells are suitable for reliable inference if the input is Smart-seq2 and 10x Genomics data, respectively, the input contains about 50 genes, and genes are expressed in >50% cells. Here, reliable inference means that key interactions (those with high CI test significance) are inferred (Appendices 3, 4). More cells are needed if the input genes are expressed in

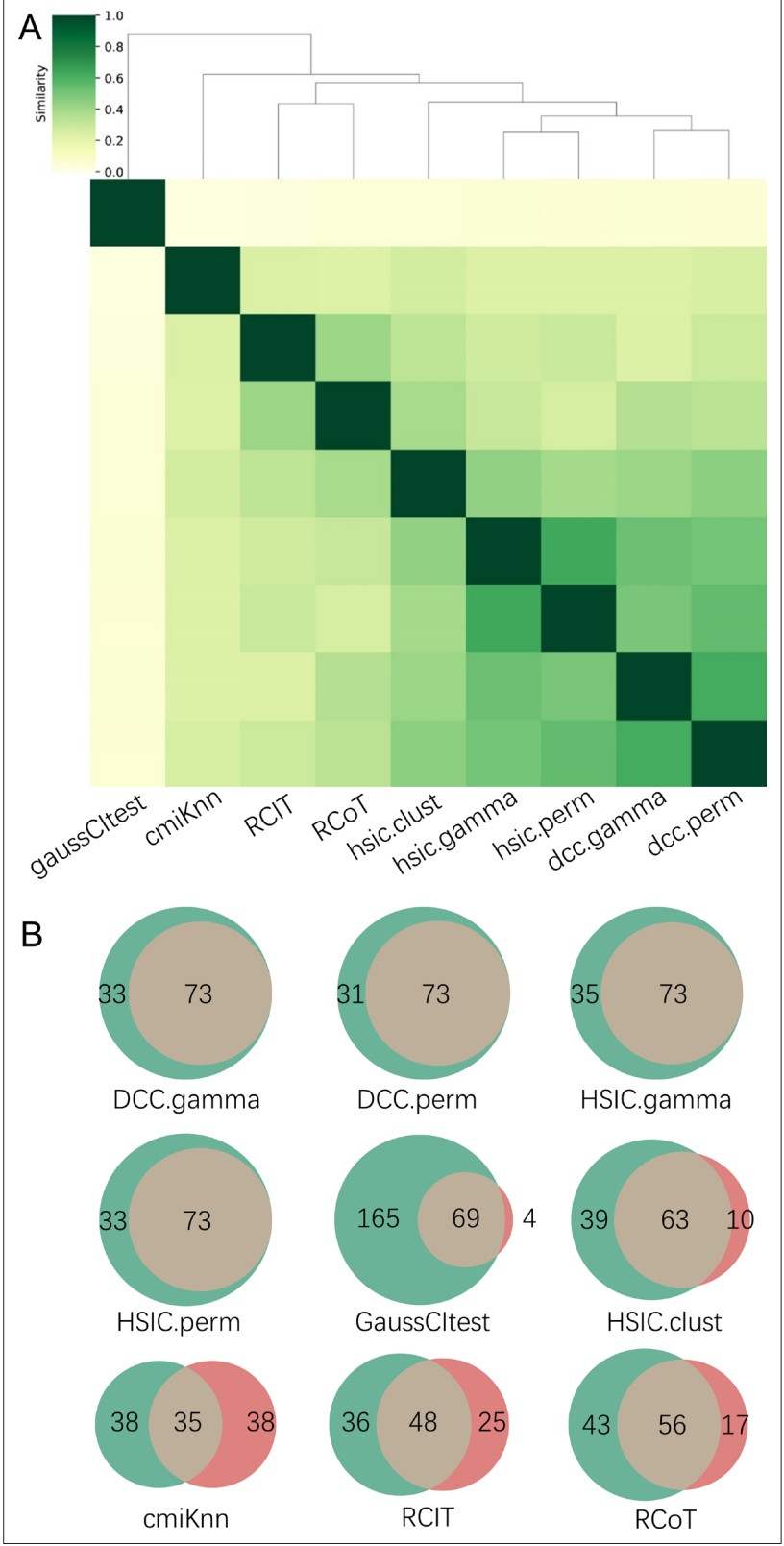

**Figure 2.** The accuracy of PC+nine CI tests was evaluated with four steps. First, nine causal networks were inferred using the nine CI tests. Second, pairwise structural Hamming distances (SHD) between these networks were computed, and the matrix of SHD values was transformed into a matrix of similarity values (using the equation *Similarity = exp(-Distance/2σ²)*, where *σ*=5). The networks of DCC.gamma, DCC.perm, HSIC.gamma, and HSIC.

*Figure 2 continued on next page*

*Figure 2 continued*

perm share the highest similarity. Third, a consensus network was built using the networks of the above four CI tests, which was assumed to be closer to the ground truth than the network inferred by any single algorithm. Fourth, each of the nine networks was compared with the consensus network. (**A**) The cluster map shows the similarity values (darker colors indicating higher similarity). (**B**) Shared and specific interactions in each algorithm's network and the consensus network. In each panel, the gray-, green-, and pink-circled areas and numbers indicate the overlapping interactions, interactions identified specifically by the algorithm, and interactions specifically in the consensus network. There are 73 overlapping interactions between DCC.gamma's network and the consensus network, and 33 interactions were identified specifically by DCC.gamma. Thus, the true positive rate (TPR) of DCC.gamma is 73/ (73+33)=68.9%. The TPRs of DCC.perm, HSIC.gamma, HSIC.perm, GaussCItest, HSIC.clust, cmiKnn, RCIT, and RCoT are 70.2%, 67.6%, 68.9%, 29.5%, 61.8%, 47.9%, 57.1%, and 56.6%, indicating that the two distance covariance criteria (DCC) CI tests perform better than others.

fewer cells and if the input contains >50 genes. Larger sample sizes (more cells) may make more interactions be inferred, but the key interactions are stable (*Appendix 3—figure 3*). As HSIC.perm and DCC.perm employ permutation to perform CI test, the networks inferred each time may be somewhat different. Our data analyses suggest that interactions inferred by running distance covariance criteria (DCC) algorithms multiple times are quite stable (*Figure 3*).

Four parameters influence causal discovery. First, 'set the alpha level' determines the statistical significance cut-off of the CI test, and large and small values make more and fewer interactions be inferred. Second, 'select the number of cells' controls sample size, and selecting more cells makes the inference more reliable but also more time-consuming. Third, 'select how a subset of cells is sampled' determines how a subset of cells is sampled. If a subset is sampled randomly, the inferred network is not exactly reproducible (but by running multiple times, the inferred edges may show high consistency, see *Figure 3*). Fourth, 'set the size of conditional set' controls the size of conditional set when performing CI tests; it influences both network topology and time consumption and should be set with care. Since some CI tests are time-consuming and running causal discovery with multiple algorithms are especially time-consuming, providing an email address is necessary to make the result sent to the user automatically.

The performance of different PC+CI tests was intensively evaluated. First, we evaluated the accuracy, time consumption, sample requirement, and stability of PC+nine CI tests using simulated data and the non-small cell lung cancer (NSCLC) cell line H2228 and the normal lung alveolar cells (as the case and control) (*Tian et al., 2019*; *Travaglini et al., 2020*). Comparing inferred networks with the

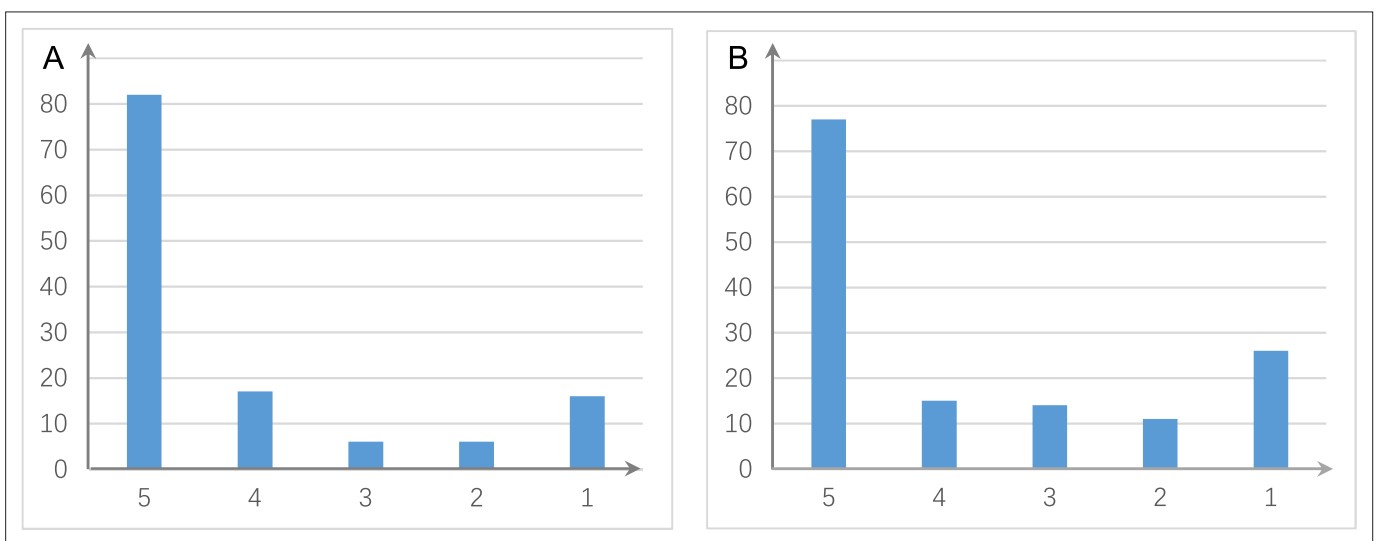

**Figure 3.** The shared and distinct interactions inferred by running causal discovery five times using the H2228 cell line dataset. Numbers on the vertical and horizontal axes represent the percentages of interactions in 1, 2, 3, 4, and 5 networks, respectively. (**A**) The results of PC+DCC.gamma. (**B**) The results of PC+DCC.perm. These results indicate that 78% and 64.3% of interactions occurred stably in ≥4 networks, suggesting that the inferred networks are quite stable.

consensus network suggests that the two DCC CI tests are most accurate and most time-consuming, suitable for small-scale network inference. RCIT and RCoT, two approximated versions of the KCIT, are moderately accurate and relatively fast, suitable for large-scale network inference. GaussCItest is the fastest and suitable for data with Gaussian distribution (*Figure 2*; *Appendix 3—figure 2*). Second, we compared the performance of PC+DCC.gamma, GSP+DCC.gamma, and GES. The former two have comparable performance, and both are more accurate and time-consuming than GES (*Appendix 3—figures 4 and 5*).

### Verification of causal discovery

We used the five NSCLC cell lines (A549, H1975, H2228, H838, and HCC827), the normal alveolar cells, and genes in specific pathways to validate network inference by PC+DCC.gamma (*Tian et al., 2019*; *Travaglini et al., 2020*). First, upon the combined conditions of (a) gene expression value >0.1, (b) gene expression in >50% cells, and (c) fold change >0.3, we identified highly and differentially expressed genes in each cell line against the alveolar cells. Second, we applied gene set enrichment analysis to differentially expressed genes in each cell line using the g:Profiler and GSEA programs. g:Profiler identified 'Metabolic reprogramming in colon cancer' (WP4290), 'Pyrimidine metabolism' (WP4022), and 'Nucleotide metabolism' (hsa01232) as enriched pathways in all cancer cell lines, and GSEA identified 'Non-small cell lung cancer' (hsa05223) as an enriched pathway in cancer cell lines ('WP' and 'hsa' indicate WikiPathways and KEGG pathways). Many studies reveal that glucose metabolism is reprogrammed and nucleotide synthesis is increased in cancer cells. Key features of reprogrammed glucose metabolism in cancer cells include increased glucose intake, increased lactate generation, and using the glycolysis/TCA cycle intermediates to synthesize nucleotides. The networks inferred by PC+DCC.gamma capture these features despite of the absence of metabolites in these datasets. The networks of WP4022 also capture the key features of pyrimidine metabolism. In the networks of hsa05223, over 50% inferred interactions agree with pathway annotations. These results support network inference (Appendix 4).

### Evaluating and ensuring the reliability

Single-cell data vary in quality and sample sizes; thus, it is important to effectively evaluate and ensure the reliability of network inference. Inspired by using RNA spike-in to measure RNA-seq quality (*Jiang et al., 2011*), we developed a method to evaluate and ensure the reliability of causal discovery. This method includes three steps: extracting the data of several well-known genes and their interactions from certain dataset as the 'spike-in' data, integrating the spike-in data into the case dataset, and applying causal discovery to the integrated dataset (the latter two steps are performed automatically when a spike-in dataset is chosen or uploaded). The user can choose a spike-in dataset in the platform or design and upload a spike-in dataset. In the inferred network, a clear separation of genes and their interactions in the spike-in dataset from genes and interactions in the case dataset is an indicator of reliable inference (*Appendix 4—figure 1*). Some public databases (e.g. the STRING database, https://string-db.org/) can also be used to evaluate inferred interactions (*Appendix 4—figures 2 and 3*).

## Results

### The analysis of lung cancer cell lines and alveolar epithelial cells

Down-regulated MHC-II genes help cancer cells avoid being recognized by immune cells (*Rooney et al., 2015*); thus, identifying genes and interactions involved in MHC-II gene down-regulation is important. To assess if causal discovery helps identify the related interactions, we examined the five NSCLC cancer cell lines (A549, H1975, H2228, H838, and HCC827) and the normal alveolar epithelial cells (*Tian et al., 2019*; *Travaglini et al., 2020*). For each of the six datasets, we took the five MHC-II genes (*HLA-DPA1*, *HLA-DPB1*, *HLA-DRA*, *HLA-DRB1*, *HLA-DRB5*) as the response variables (genes of interest, hereafter also called target genes) and selected 50 feature genes (using BAHSIC, unless otherwise stated) from all genes expressed in >50% cells. Then, we applied the nine causal discovery algorithms to the 50 genes in 300 cells sampled from each of the datasets. The two DCC algorithms performed the best when processing the H2228 cells and lung alveolar epithelial cells (*Appendix 5—figures 1 and 2*).

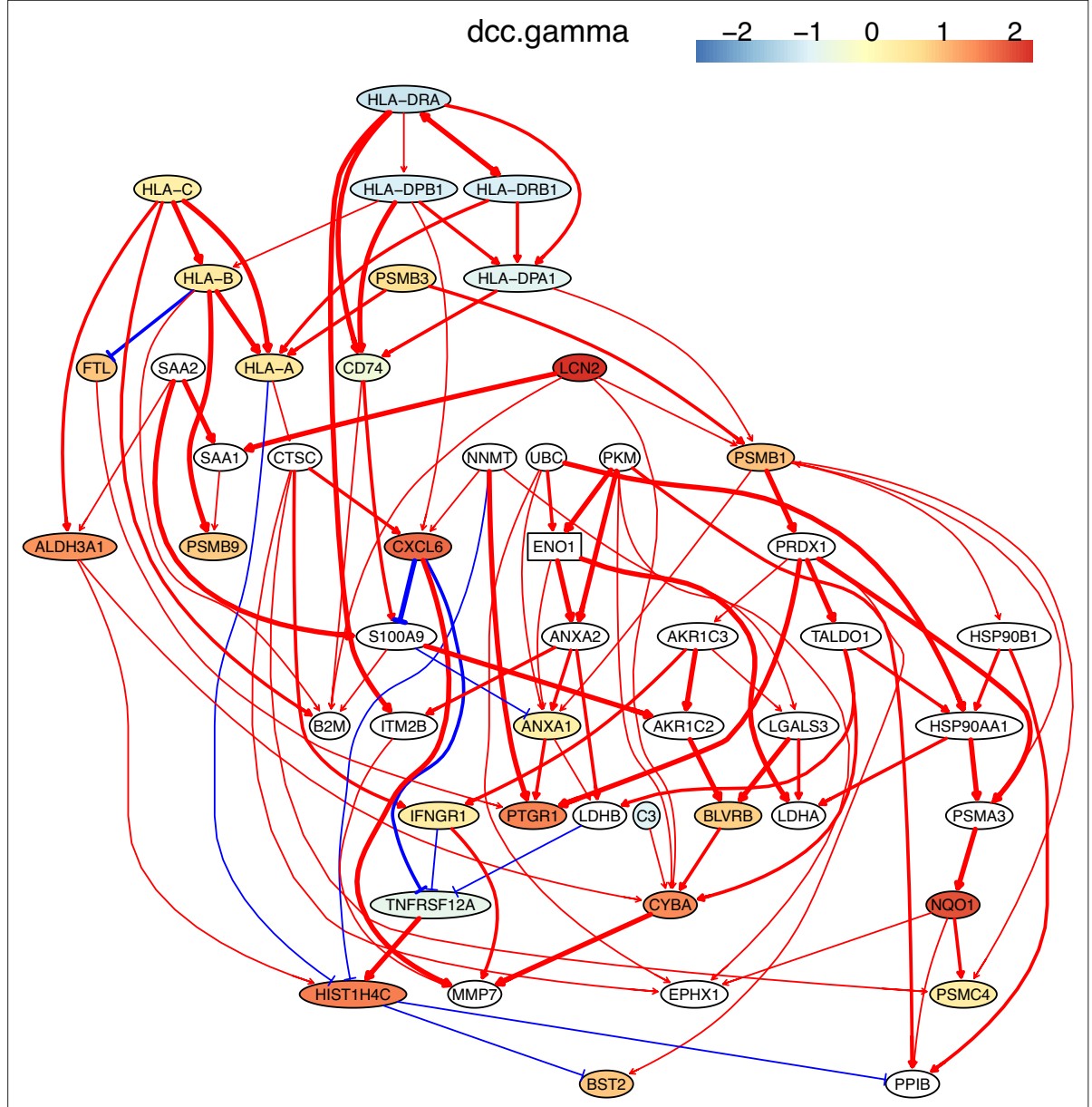

**Figure 4.** The network of the 50 genes inferred by DCC.gamma from the H2228 dataset (the alpha level for CI test was 0.1). Red → and blue -| arrows indicate activation and repression, and colors indicate fold changes of gene expression compared with genes in the alveolar epithelial cells.

The inferred networks also show that down-regulated genes weakly but up-regulated genes strongly regulate downstream targets and that activation and repression lead to up- and down-regulation of target genes. These features are biologically reasonable. Many inferred interactions, including those between MHC-II genes and CD74, between CXCL genes, and between MHC-I genes and B2M, are supported by the STRING database (http://string-db.org) and experimental findings (*Figure 4*; *Appendix 4—figure 2*; *Castro et al., 2019*; *Karakikes et al., 2012*; *Szklarczyk et al., 2021*). An interesting finding is the PRDX1→TALDO1→HSP90AA1→NQO1→PSMC4 cascade in H2228 cells. Interactions between PRDX1/TALDO1/HSP90AA1 and NQO1 were reported (*Mathew et al., 2013*; *Yin et al., 2021*), but the interaction between NQO1 and PSMC4 was not. Previous findings on *NQO1* include that it determines cellular sensitivity to the antitumor agent napabucasin in many cancer cell lines (*Guo et al., 2020*), is a potential poor prognostic biomarker, and is a promising therapeutic target for patients with lung cancers (*Cheng et al., 2018*; *Siegel et al., 2012*), and that mutations in *NQO1* are associated with susceptibility to various forms of cancer. Previous findings on

*PSMC4* include that high levels of PSMC4 (and other PSMC) transcripts were positively correlated with poor breast cancer survival (*Kao et al., 2021*). Thus, the inferred NQO1→PSMC4 probably somewhat explains the mechanism behind these experimental findings.

## The analysis of macrophages isolated from glioblastoma

Macrophages critically influence glioma formation, maintenance, and progression (*Gutmann, 2020*), and CD74 is the master regulator of macrophage functions in glioblastoma (*Alban et al., 2020*; *Quail and Joyce, 2017*; *Zeiner et al., 2015*). To examine the function of CD74 in macrophages in gliomas, we used CD74 as the target gene and selected 50 genes from genes expressed in >50% of macrophages isolated from glioblastoma patients (*Neftel et al., 2019*). In the networks of DCC algorithms (*Appendix 5—figure 3*), CD74 regulates MHC-II genes, agreeing with the finding that CD74 is an MHC-II chaperone and plays a role in the intracellular sorting of MHC class II molecules. The network includes interactions between C1QA/B/C, agreeing that they form the complement C1q complex. The identified TYROBP→TREM2→A2M→APOE→APOC1 cascade is supported by the reports that TREM2 is expressed in tumor macrophages in over 200 human cancer cases (*Molgora et al., 2020*) and that there are interactions between TREM2/A2M, TREM2/APOE, A2M/APOE, and APOE/APOC1 (*Krasemann et al., 2017*).

## The analysis of tumor-infiltrating exhausted CD8 T cells

Tumor-infiltrating exhausted CD8 T cells are highly heterogeneous yet share common differentially expressed genes (*McLane et al., 2019*; *Zhang et al., 2018*), suggesting that CD8 T cells undergo different processes to reach exhaustion. We analyzed three exhausted CD8 T datasets isolated from human liver, colorectal, and lung cancers (*Appendix 5—figure 4*; *Guo et al., 2018*; *Zhang et al., 2018*; *Zheng et al., 2017*). A key feature of CD8 T cell exhaustion identified in mice is PDCD1 up-regulation by TOX (*Khan et al., 2019*; *Scott et al., 2019*; *Seo et al., 2019*). Using TOX and PDCD1 as the target gene, we selected 50 genes expressed in >50% exhausted CD8 T cells and 50 genes expressed in >50% non-exhausted CD8 T cells, respectively. Transcriptional regulation of PDCD1 by TOX was observed in LCMV-infected mice without mentioning any role of CXCL13 (*Khan et al., 2019*). Here, indirect TOX→PDCD1 (via genes such as CXCL13) was inferred in exhausted CD8 cells, and direct TOX→PDCD1 was inferred in non-exhausted CD8 T cells (although the expression of TOX and PDCD1 is low in these cells) (*Appendix 5—figure 4*). Recently, CXCL13 was found to play a critical role in T cells for effective responses to anti-PD-L1 therapies (*Zhang et al., 2021b*). The causal discovery results help reveal differences in CD8 T cell exhaustion between humans and mice and under different pathological conditions. The PDCD1→TOX inferred in exhausted and non-exhausted CD8 T cells may indicate some feedback between TOX and PDCD1, as on the proteome level, a study reported that the binding of PD1 to TOX in the cytoplasm facilitates the endocytic recycling of PD1 (*Wang et al., 2019*).

## Identifying genes and inferring interactions that signify CD4 T cell aging

How immune cells age and whether some senescence signatures reflect the aging of all cell types draw wide attention (*Gorgoulis et al., 2019*). We analyzed gene expression in naive, TEM, rTreg, naive_Isg15, cytotoxic, and exhausted CD4 T cells from young (2–3 months, n=4) and old (22–24 months, n=4) mice (*Appendix 5—figures 5*; *Elyahu et al., 2019*). For each cell type, we compared the combined data from all four young mice with the data from each old mouse to identify differentially expressed genes. If genes were expressed in >25% cells and consistently up/down-regulated (|fold change|>0) in most of the 24 comparisons, we assumed them as aging-related (*Appendix 5—table 1*). Some of these identified genes play important roles in the aging of T cells or other cells, such as the mitochondrial genes encoding cytochrome c oxidases and the gene *Sub1* in the mTOR pathway (*Bektas et al., 2019*; *Gorgoulis et al., 2019*; *Goronzy and Weyand, 2019*; *Walters and Cox, 2021*). We directly used these genes, plus one CD4-specific biomarker (Cd28) and two reported aging biomarkers (Cdkn1b, Cdkn2d) (*Gorgoulis et al., 2019*; *Larbi and Fulop, 2014*), as feature genes to infer their interactions in different CD4 T cells in young and old mice. The inferred causal networks unveil multiple findings (*Appendix 5—figure 5*). First, B2m→H2-Q7 (a mouse MHC class I gene), Gm9843→Rps27rt (Gm9846), and the interactions between the five mitochondrial genes

(MT-ATP6, MT-CO1/2/3, and MT-Nd1) were inferred in nearly all CD4 T cells. Second, many interactions are supported by the STRING database (*Appendix 4—figure 3*). Third, some interactions agree with experimental findings, including Sub1-|Lamtor2 (*Chen et al., 2021*) and the regulations of these mitochondrial genes by Lamtor2 (*Morita et al., 2017*). Fourth, Gm9843→Rps27rt→Junb were inferred in multiple CD4 T cells (both Gm9843 and Rps27rt are mouse-specific). Since JUNB belongs to the AP-1 family TFs that are increased in all immune cells during human aging (*Zheng et al., 2020*), Gm9843→Rps27rt→Junb could highlight a counterpart regulation of JUNB in human immune cells.

## Discussion

### Single-cell causal discovery

Various methods have been proposed to infer interactions between variables from observational data. As surveyed recently (*Nguyen et al., 2021*; *Pratapa et al., 2020*), many methods assume linear relationships between variables and the Gaussian distribution of data. These assumptions enable these methods to run fast, handle many genes and even perform transcriptome-wide prediction. However, our algorithm benchmarking results suggest that networks inferred by fast methods with these assumptions should be concerned.

Causal discovery infers causal interactions directly upon observations of variables without assuming relationships between variables and the distribution of data. Because genes and molecules have varied relationships in different cells, causal discovery better satisfies inferring their interactions than other methods. Causal discovery methods have reviewed recently (*Glymour et al., 2019*; *Yuan and Shou, 2022*), but workflows and platforms integrating multiple methods for analyzing scRNA-seq data remain rare.

Our integration and benchmarking of multiple methods (note that these methods are not for inferring causal relationships from temporal data) and analysis of multiple datasets generate several conclusions. First, although kernel-based CI tests are time-consuming (*Shah and Peters, 2020*), applying them to a set of genes is feasible. A set of genes can be generated by feature selection, by gene set enrichment analysis, or by manual selection. Second, the cost of time consumption pays off in network accuracy, as the most time-consuming CI tests generate the most reliable results. Thus, trade-offs between time consumption, network size, and network accuracy should be made. Third, causal discovery can infer signaling networks or gene regulatory networks, depending on the input. If genes encoding TFs and their targets are the input, gene regulatory networks are inferred. Fourth, dropouts and noises in scRNA-seq data concern researchers and trouble correlation analysis (*Hou et al., 2020*; *Mohan and Pearl, 2018*; *Tu et al., 2019*), but can be well tolerated by PC+kernel-based CI tests if samples are sufficiently large. Finally, using 'spike-in' data can effectively evaluate the reliability of causal discovery.

### Challenges of data analysis

Single-cell causal discovery also faces several challenges. First, causal discovery assumes there are no unmeasured common causes (the causal sufficiency assumption), but in real data latent and unobserved variables are common and hard to identify. Specifically, inferring interactions between highly expressed or differentially expressed genes is a case of causal discovery with incomplete models (i.e. models with missing variables from the data-generating model). In this situation, what are inferred are indirect relationships instead of direct interactions between gene products. Second, constraint-based methods cannot differentiate networks belonging to a Markov equivalent class (the causal Markov assumption). This can be solved partly by combined use of PC and DAGMA-nonlinear (which can better determine the direction of edges). Third, the following examples indicate that the lack of relevant information makes judging inferred interactions and relationships difficult. (a) TOX is reported to activate PDCD1 in exhausted CD8 T cells in mice (*Khan et al., 2019*), but whether CXCL13 is involved in (or required for) the TOX-PDCD1 interaction in exhausted CD8 T cells in humans is unclear, until recently CXCL13 is reported to play critical roles in T cells for effective responses to anti-PD-L1 therapies (*Zhang et al., 2021b*). (b) The differences in inferred networks in exhausted CD8 T cells from different cancers are puzzling, until a recent study reports that exhausted CD8 T cells show high heterogeneity and exhaustion can follow different paths (*Zheng et al., 2021*). (c) It is difficult to explain multiple genes encoding ribosomal proteins in the inferred networks in CD4 cells from old

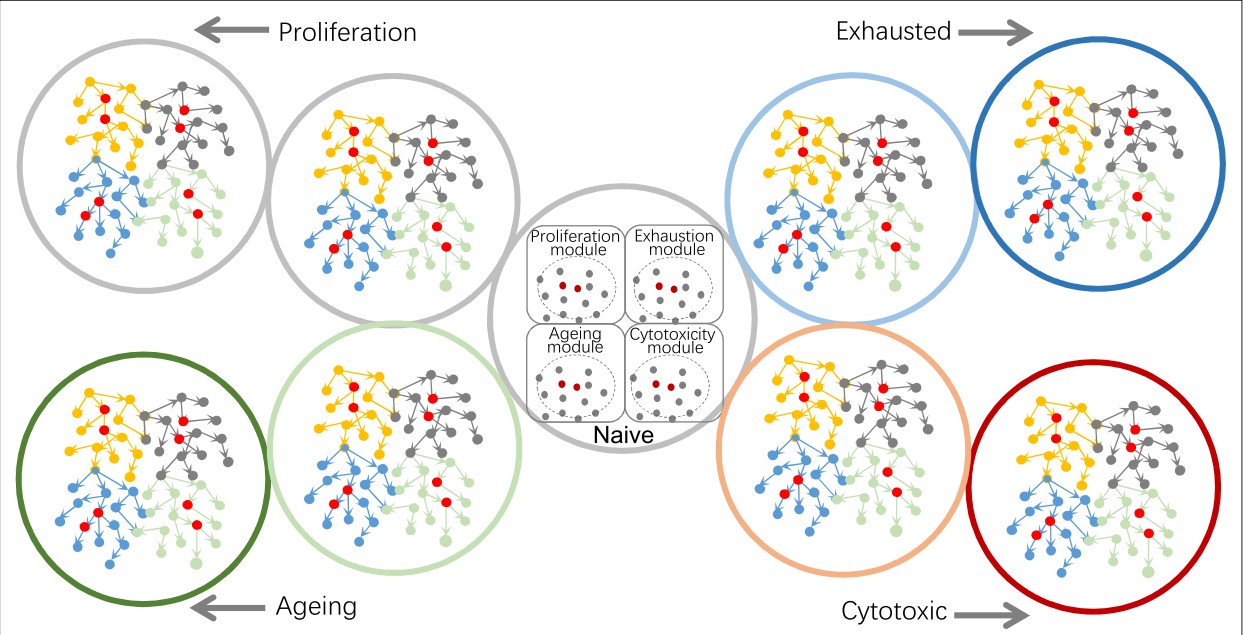

**Figure 5.** Using causal discovery to analyze different cells, cells at different stages, or different biological processes in cells. The red and gray dots within the four circles in the central cell indicate the four modules' core genes and related genes. Genes in different modules should be chosen as target genes when exploring different biological processes.

mice, until a recent study reports that aging impairs ribosomes' ability to synthesize proteins efficiently (*Stein et al., 2022*).

## Limitations of the study

The time consumption of kernel-based CI tests disallows inferring large networks, and how this challenge can be solved remains unsolved. C codes may be developed to replace the most time-consuming parts of the R functions, but this has not been done.

## Tips for best practices

First, exploring different biological modules or processes needs careful selection of genes (*Figure 5*). When it is unclear what genes are most relevant to one or several target genes, it is advisable to run multiple rounds of feature selection using different combination of target genes as response variable(s). Second, when feature genes are identified by gene set enrichment analysis or upon highly expressed genes, PC+kernel-based CI tests perform better than continuous optimization-based methods, and the inferred networks consist more likely of indirect causal relationships instead of direct causal interactions. Third, BAHSIC and SHS are the best feature selection algorithms. Since selecting feature genes from too many candidates is unreliable, filtering genes upon specific conditions (e.g. expression values, expressed cells, fold changes) is necessary. Fourth, DCC.gamma and DCC.perm are the best CI tests working with PC. When building consensus networks, it is advisable to use the results of just DCC CI tests. Fifth, trade-offs between scale, reliability, and accuracy are inevitable. When examining many genes, RCIT/RCoT may be proper, and when examining large datasets, sub-sampling is necessary. For Smart-seq2 and 10x Genomics datasets, 300 and 600 cells are recommended for analyzing 50–60 genes expressed in >50% of cells. More cells are needed if more genes are selected and/or selected genes are expressed in fewer cells (e.g. 25%). Sixth, when it is unclear if a sub-sampled dataset is large enough, repeat causal discovery several times using different sizes of sub-samples. If the inferred networks are similar, the sub-samples should be sufficient. Seventh, using "spike-in" datasets helps measure and ensure reliability. Eighth, carefully inspect the potential influence of cell heterogeneity on causal discovery, and caution is needed when interpreting the results of heterogeneous cells.

## Acknowledgements

This work was supported by the National Natural Science Foundation of China (31771456) and the Department of Science and Technology of Guangdong Province (2020A1515010803). We appreciate the help from Prof. Ruichu Cai at the Guangdong University of Technology.

## Additional information

### Funding

| Funder | Grant reference number | Author |
|---|---|---|
| National Natural Science Foundation of China | 31771456 | Hao Zhu |
| Department of Science and Technology of Guangdong Province | 2020A1515010803 | Hao Zhu |

The funders had no role in study design, data collection and interpretation, or the decision to submit the work for publication.

### Author contributions

Yujian Wen, Jielong Huang, Software, Formal analysis, Methodology; Shuhui Guo, Software; Yehezqel Elyahu, Resources, Data curation; Alon Monsonego, Resources, Data curation, Writing - review and editing; Hai Zhang, Software, Visualization; Yanqing Ding, Data curation, Supervision; Hao Zhu, Conceptualization, Formal analysis, Supervision, Funding acquisition, Writing - original draft, Project administration, Writing - review and editing

### Author ORCIDs

Hao Zhu http://orcid.org/0000-0001-7384-3840

### Decision letter and Author response

Decision letter https://doi.org/10.7554/eLife.81464.sa1
Author response https://doi.org/10.7554/eLife.81464.sa2

## Additional files

### Supplementary files

- MDAR checklist

### Data availability

Only public data were used. Links to all data are provided in the manuscript.

The following previously published datasets were used:

| Author(s) | Year | Dataset title | Dataset URL | Database and Identifier |
|---|---|---|---|---|
| Geirsdottir L | 2019 | Cross-species analysis across 450 million years of evolution reveals conservation and divergence of the microglia program (scRNA-seq) | https://www.ncbi.nlm.nih.gov/geo/query/acc.cgi?acc=GSE134705 | NCBI Gene Expression Omnibus, GSE134705 |
| Tian L | 2019 | Designing a single cell RNA sequencing benchmark dataset to compare protocols and analysis methods [5 Cell Lines 10X] | https://www.ncbi.nlm.nih.gov/geo/query/acc.cgi?acc=GSE126906 | NCBI Gene Expression Omnibus, GSE126906 |

*Continued on next page*

*Continued*

| Author(s) | Year | Dataset title | Dataset URL | Database and Identifier |
|-----------|------|---------------|-------------|-------------------------|
| Travaglini KJ | 2020 | Human Lung Cell Atlas | https://www.synapse.org/#!Synapse:syn21041850 | Synapase, syn21041850 |
| Elyahu Y | 2019 | Study: Aging promotes reorganization of the CD4 T cell landscape toward extreme regulatory and effector phenotypes | https://singlecell.broadinstitute.org/single_cell/study/SCP490/aging-promotes-reorganization-of-the-cd4-t-cell-landscape-toward-extreme-regulatory-and-effector-phenotypes | Single Cell Portal, SCP490 |
| Neftel C | 2019 | Single cell RNA-seq analysis of adult and paediatric IDH-wildtype Glioblastomas | https://www.ncbi.nlm.nih.gov/geo/query/acc.cgi?acc=GSE131928 | NCBI Gene Expression Omnibus, GSE131928 |
| Guo X | 2018 | T cell landscape of non-small cell lung cancer revealed by deep single-cell RNA sequencing | https://www.ncbi.nlm.nih.gov/geo/query/acc.cgi?acc=GSE99254 | NCBI Gene Expression Omnibus, GSE99254 |
| Zhang L | 2018 | Lineage tracking reveals dynamic relationships of T cells in colorectal cancer | https://www.ncbi.nlm.nih.gov/geo/query/acc.cgi?acc=GSE108989 | NCBI Gene Expression Omnibus, GSE108989 |
| Zheng C | 2018 | Landscape of infiltrating T cells in liver cancer revealed by single-cell sequencing | https://www.ncbi.nlm.nih.gov/geo/query/acc.cgi?acc=GSE98638 | NCBI Gene Expression Omnibus, GSE98638 |

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

## Appendix 1

## Overview of data and algorithms

### 1. Algorithms and datasets

We combined feature selection and causal discovery to infer causal interactions among a set of gene products in single cells. We used synthetic data, semi-synthetic data, real scRNA-seq data, and flow cytometry data to benchmark nine feature selection algorithms and nine causal discovery algorithms (*Appendix 1—tables 1 and 2*; *Appendix 1—figure 1*).

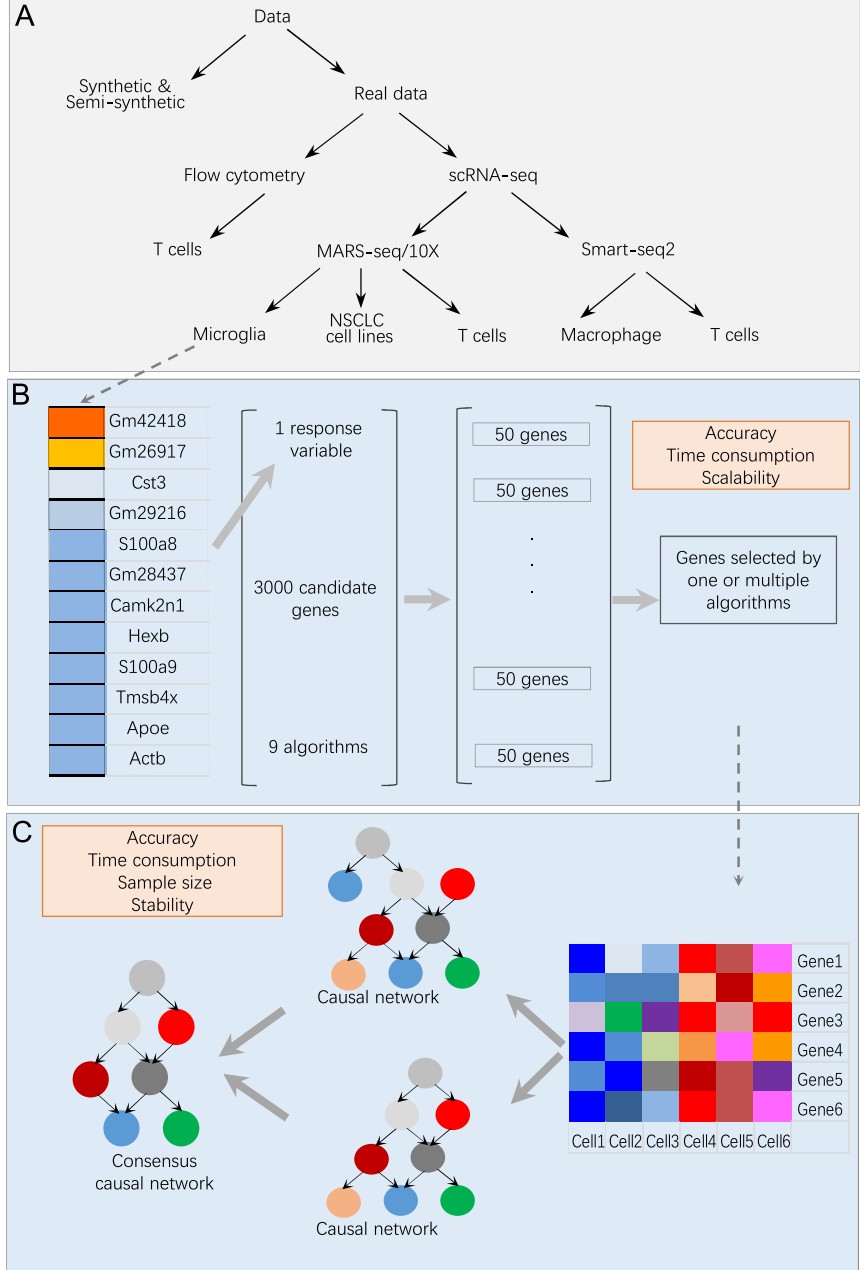

**Appendix 1—figure 1.** Overview of data and benchmarking. (**A**) The single-cell RNA-sequencing (scRNA-seq) data were generated by different protocols and from different cell types (*Appendix 1—table 1*). (**B**) Illustration of feature selection benchmarking using data of microglia from humans and mice. Steps: (i) choose a target gene from a list of microglia biomarkers; (ii) let each algorithm select 50 genes from 3000 candidates expressed in most cells; (iii) merge the nine sets of feature genes into a superset; (iv) compare each selected feature gene set with the superset. (**C**) Illustration of causal discovery benchmarking using a set of feature genes. Steps: (i) use nine
*Appendix 1—figure 1 continued on next page*

*Appendix 1—figure 1 continued*
algorithms (PC+CI tests) to generate nine causal networks; (ii) generate a type 1 consensus network upon the networks of multiple algorithms (a type 2 consensus network is generated upon running an algorithm multiple times); (iii) compare each causal network with the consensus network.

**Appendix 1—table 1.** Real single-cell RNA-sequencing (scRNA-seq) data.

| Dataset | Cell type | Species | Protocols | Dataset URL | Database and Identifier | References |
|---|---|---|---|---|---|---|
| 1 | Microglia from humans and mice | Human and mouse | MARS-seq | https://www.ncbi.nlm.nih.gov/geo/query/acc.cgi?acc=GSE134705 | NCBI Gene Expression Omnibus, GSE134705 | *Geirsdottir et al., 2019* |
| 2 | Five lung cancer cell lines (A549, H1975, H2228, H838, HCC827) from the CellBench benchmarking dataset | Human | 10x Genomics | https://www.ncbi.nlm.nih.gov/geo/query/acc.cgi?acc=GSE126906 | NCBI Gene Expression Omnibus, GSE126906 | *Tian et al., 2019* |
| 3 | Lung alveolar epithelial cells | Human | 10x Genomics | https://www.synapse.org/#!Synapse:syn21041850 | Synapase, syn21041850 | *Travaglini et al., 2020* |
| 4 | Six types of CD4 T cells (naïve, TEM, rTregs, naïve_Isg15, cytotoxic, exhausted) from young and old mice | Mouse | 10x Genomics | https://singlecell.broadinstitute.org/single_cell/study/SCP490/aging-promotes-reorganization-of-the-cd4-t-cell-landscape-toward-extreme-regulatory-and-effector-phenotypes | Single Cell Portal, SCP490 | *Elyahu et al., 2019* |
| 5 | Macrophages isolated from glioblastomas | Human | Smart-seq2 | https://www.ncbi.nlm.nih.gov/geo/query/acc.cgi?acc=GSE131928 | NCBI Gene Expression Omnibus, GSE131928 | *Neftel et al., 2019* |
| 6 | Exhausted CD8 T cells isolated from liver cancer, lung cancer, and CRC | Human | Smart-seq2 | https://www.ncbi.nlm.nih.gov/geo/query/acc.cgi?acc=GSE99254. https://www.ncbi.nlm.nih.gov/geo/query/acc.cgi?acc=GSE108989. https://www.ncbi.nlm.nih.gov/geo/query/acc.cgi?acc=GSE98638. | NCBI Gene Expression Omnibus, GSE99254. NCBI Gene Expression Omnibus, GSE108989. NCBI Gene Expression Omnibus, GSE98638. | *Guo et al., 2018; Zhang et al., 2018; Zheng et al., 2017* |
| 7 | Non-exhausted CD8 T cells isolated from the normal liver, lung, and colorectal tissues | Human | Smart-seq2 | https://www.ncbi.nlm.nih.gov/geo/query/acc.cgi?acc=GSE99254. https://www.ncbi.nlm.nih.gov/geo/query/acc.cgi?acc=GSE108989. https://www.ncbi.nlm.nih.gov/geo/query/acc.cgi?acc=GSE98638. | NCBI Gene Expression Omnibus, GSE99254. NCBI Gene Expression Omnibus, GSE108989. NCBI Gene Expression Omnibus, GSE98638. | *Guo et al., 2018; Zhang et al., 2018; Zheng et al., 2017* |
| 8 | CD4 T cell | Human | Flow cytometry | https://www.science.org/doi/10.1126/science.1105809 | Science Supplementary Materials, doi: 10.1126/science.1105809 | *Sachs et al., 2005* |

**Appendix 1—table 2.** Feature selection and causal discovery algorithms.

| Feature selection | Category | Causal discovery | Category |
|---|---|---|---|
| Random forests | | GaussCItest | Test for CI between Gaussian random variables upon partial correlation |
| Extremely randomized trees | | DCC.perm | Test for CI using a distance covariance-based kernel |
| XGBoost | Ensemble learning-based | DCC.gamma | |
| SHS | | HSIC.perm | Test for CI using a HSIC-based kernel |
| BAHSIC | | HSIC.gamma | |
| Block HSIC Lasso | HSIC-based | HSIC.clust | |

*Appendix 1—table 2 Continued on next page*

*Appendix 1—table 2 Continued*

| Feature selection | Category | Causal discovery | Category |
|---|---|---|---|
| Lasso | | RCIT | Test for CI using an approximate KCIT kernel |
| Ridge regression | | RCoT | |
| Elastic net | Regularization-based | CMIknn | Test for CI based on conditional mutual information |

## 2. Synthetic data for feature selection

### Fully synthetic dataset

*N* variables (indicating candidate genes) without a specific pattern were generated randomly from a (0, 2) uniform distribution, from which *n* variables (indicating feature genes) were selected randomly to synthesize a response variable. Each feature influences the response variable depending randomly on one of the five functions: $y=x^2$, $y=sin(x)$, $y=cos(x)$, $y=tanh(x)$, and $y=e^x$.

By combining different numbers of feature genes and candidate genes (4–50, 8–100, 8–200, 20–500, 20–1000, and 50–2000) and generating samples of different sizes (100, 200, 500, 1000, and 2000), we generated 30 schemes. For each algorithm, we ran each scheme 10 times, and each time the true positive rate (TPR) was calculated by:

$$TPR = \frac{selected\ features \cap true\ features}{true\ features}.$$

A TPR of 1.0 means that the feature selection algorithm completely correctly selects the features; a small TPR indicates poor performance.

### Semi-synthetic dataset

First, we extracted genes from a benchmark scRNA-seq dataset (https://support.10xgenomics.com/single-cell-gene-expression/datasets/4.0.0/Parent_NGSC3_DI_PBMC), sorted these genes based on the cells in which they were expressed (expression level >0), and obtained the top 5000 genes expressed in most cells. Then, we obtained the top 5000 cells that contained the most expressed genes. These genes and cells formed a 5000*5000 matrix. Different candidate gene sets were sampled from this matrix, and different feature gene sets were selected randomly from each candidate gene set. Next, the response variables (target genes) were synthesized using feature genes.

## 3. Synthetic data for causal discovery

We used the *randomDAG* function in the *pcalg* package (https://cran.r-project.org/web/packages/pcalg/index.html) to generate DAGs with random topologies. Values of nodes (i.e. genes) in these DAGs were randomly generated using the following 10 functions that determined relationships between nodes:

$$y = x^2, \quad y = \sqrt{abs(x)}, \quad y = sin(x) * sin(x), \quad y = sin(x), \quad y = cos(x),$$

$$y = cos(x) * cos(x), \quad y = tanh(x), \quad y = e^{log_2(abs(x))}, \quad y = log(|x| + 1), \quad y = tanh(x) * tanh(x)$$

With the variable number ranging from 20 to 80 (step = 20), and the sample size ranging from 500 to 1000 (step = 500), we generated eight datasets with known networks.

## 4. Real single-cell data for feature selection and causal discovery

Single-cell datasets in *Appendix 1—table 1* were used for benchmarking.

## Appendix 2

### Feature selection algorithms and benchmarking

1. Ensemble learning-based algorithms

Random forests

We used the *RandomForestRegressor* function (with default parameters) in the *sklearn* package (https://scikit-learn.org/stable/) to build random forest models (**Breiman, 2001**). Each model contained 200 decision trees. After regression based on the response variable(s), genes were sorted based on *Gini importance,* and the top genes were selected as feature genes.

Extremely randomized trees

We used the *ExtraTreesRegressor* function (with default parameters) in the *sklearn* package (https://scikit-learn.org/stable/) to generate extremely randomized trees (**Geurts et al., 2006**). Each tree model contained 200 decision trees. After regression based on the response variable(s), genes were sorted based on *Gini importance,* and the top genes were selected as feature genes.

XGBoost

We used the *XGBRegressor* function (with default parameters) in the *Scikit-Learn API* (https://xgboost.readthedocs.io/en/latest/python/python_api.html) to build the XGBoost models (**Chen and Guestrin, 2016**). Each XGBoost model contained 200 decision trees. After regression based on the response variable(s), genes were sorted based on *Gini importance,* and the top genes were selected as feature genes.

2. Regularization-based algorithms

Lasso

We used the *Lasso* function (with default parameters) in the *sklearn* package (https://scikit-learn.org/stable/) to produce the regression models. In the Lasso (least absolute shrinkage and selection operator) regression equation (**Tibshirani, 1997**):

$$\beta^{lasso} = \underset{\beta}{\mathrm{argmin}} \left\{ \frac{1}{2N} \sum_{i=1}^{N} \left( y_i - \beta_0 - \sum_{j=1}^{p} x_{ij}\beta_j \right)^2 + \lambda \sum_{j=1}^{p} \left| \beta_j \right| \right\},$$

$N$ is the number of samples, $p$ is the number of features, $\beta_j$ is the coefficient of the $j$th feature, and $\lambda$ (by default $\lambda=0.5$) is a penalty coefficient controlling the shrinkage. Feature genes were selected based on the value of $|\beta_j|$, which indicates the importance of the $j$ th feature for the response variable(s).

Ridge regression

We used the *Ridge* function (with default parameters) in the *sklearn* package (https://scikit-learn.org/stable/) to build Ridge repression models. The equation of Ridge regression is similar to that of Lasso (**Hoerl and Kennard, 2000**):

$$\beta^{ridge} = \underset{\beta}{\mathrm{argmin}} \left\{ \frac{1}{2N} \sum_{i=1}^{N} \left( y_i - \beta_0 - \sum_{j=1}^{p} x_{ij}\beta_j \right)^2 + \lambda \sum_{j=1}^{p} \beta_j^2 \right\},$$

but the L2 penalty term is $\sum_{j=1}^{p} \beta_j^2$ . Feature genes were selected based on the value of $|\beta_j|$, which indicates the importance of the $j$ th feature for the response variable(s).

Elastic net

We used the *ElasticNet* function (with default parameters) in the *sklearn* package (https://scikit-learn.org/stable/) to build elastic net models. Elastic net linearly combines the L1 and L2 penalties of the Lasso and Ridge methods using the following equation (by default $\lambda=1$ and $\alpha=0.5$) (**Zou and Hastie, 2005**). In the equation:

$$\beta^{elastic\ net} = \underset{\beta}{\operatorname{argmin}} \left\{ \frac{1}{2N} \sum_{i=1}^{N} \left( y_i - \beta_0 - \sum_{j=1}^{P} x_{ij}\beta_j \right)^2 + \lambda \left( \frac{1-\alpha}{2} \sum_{j=1}^{P} \beta_j^2 + \alpha \sum_{j=1}^{P} |\beta_j| \right) \right\},$$

feature genes are selected upon the value of $|\beta_j|$, which indicates the importance of the $j$ th feature for the response variable(s).

## 3. HSIC-based algorithms

### BAHSIC

Hilbert-Schmidt independence criterion (HSIC) is a measure of dependency between two variables (*Gretton et al., 2005*). After obtaining a measure between a response variable and a feature, a backward elimination process is used to extract a subset of features that are most relevant to the response variable (*Song et al., 2007*). We used the *BAHSIC* program (https://www.cc.gatech.edu/~lsong/code.html), together with the nonlinear radial basis function kernel, to evaluate the dependency between feature genes and response variable(s), and set the parameter *flg3* = $1 - float \left( \frac{desired\ feature\ number}{total\ feature\ number} \right)$ to accelerate computation.

### SHS

Sparse HSIC (SHS), which combines HSIC with fast sparse decomposition of matrices, is an HSIC-based feature selection algorithm without the backward elimination process to identify a sparse projection of all features (*Gangeh et al., 2017*). We translated the SHS program encoded in MATLAB (https://uwaterloo.ca/data-science/sites/ca.data-science/files/uploads/files/shs.zip) into a Python program and used eigenvalue decomposition as the matrix halving procedure. The parameters $\gamma$=1.1 (a penalty parameter that controls the sparsity of the solution) and $\rho = 0.1$.

### Block HSIC Lasso

HSIC Lasso is a variant of the minimum redundancy maximum relevance feature selection algorithm and is suitable for high-dimensional small sample data. We used the *pyHSICLasso* program (https://github.com/riken-aip/pyHSICLasso; *Yamada et al., 2014*), which is an approximation of HSIC Lasso but reduces memory usage dramatically while retaining the properties of HSIC Lasso (*Climente-González et al., 2019*). We used the function *get_index_score*() to compute feature importance and the function *get_features*() to return top feature genes.

## 4. Benchmarking results

We evaluated the time consumption, accuracy, and scalability of nine feature selection algorithms in three categories (*Appendix 1—tables 1 and 2*). First, tested using synthesized data, all algorithms showed moderate time consumption, which increased insignificantly when the sample size increased (*Appendix 2—figure 1*). Second, using synthesized data, multiple algorithms selected all features correctly if schemes were simple (e.g. selecting 4 features from 50 candidates). If features and/or candidates increased (e.g. selecting 50 features from 2000 candidates), BAHSIC showed the best performance, with accuracy decreasing more slowly than others (*Appendix 2—figure 2*). Third, using well-known microglial biomarkers in humans and mice (*Butovsky et al., 2014*; *Patir et al., 2019*) and using scRNA-seq data of microglia from the human and mouse brain (*Geirsdottir et al., 2019*), we further evaluated feature selection algorithms' accuracy. We merged feature genes generated by the nine algorithms into a superset (*Appendix 1—figure 1B*), identified a subset generated by the majority of algorithms, and examined how many feature genes of each algorithm overlap with the subset. When selecting 50 genes from 3000 candidates upon a target gene (e.g. Hexb), BAHSIC and SHS were the best and second-best algorithms, and they also selected most microglia biomarkers (*Appendix 2—figures 3 and 4*). Fourth, we used real scRNA-seq data in applications to evaluate algorithms' accuracy and found that BAHSIC also performs well. Finally, to evaluate algorithms' scalability, we let algorithms select feature genes from different numbers of candidate genes. When the number of candidate genes is large (>10,000), the accuracy of feature selection is somewhat decreased.

BAHSIC's performance was examined further using macrophages isolated from human glioblastoma by checking whether the selected feature genes accurately characterize macrophages (*Neftel et al., 2019*). We used six macrophage biomarkers (CD14, AIF1, FCER1G, FCGR3A, TYROBP, and CSF1) exclusively expressed in these macrophages as the target genes (response variables) and used BAHSIC to select 50 feature genes from 3000 candidate genes expressed in >50% macrophages

(*Appendix 2—figure 5*). Nearly all feature genes were expressed exclusively in the macrophages (*Appendix 2—figure 6*). Experimental findings support many feature genes. C1QA/B/C and C3 are supported by the finding that C1Q is produced and the complement cascade is up-regulated in cancer-infiltrated macrophages (*Yang et al., 2021*). CD74 and MHC-II genes are supported by the finding that CD74 is correlated with malignancies and the immune microenvironment in gliomas (*Xu et al., 2021*). TREM2 and APOE are supported by the finding that highly expressed TREM2 and APOC2 in macrophages contribute to immune checkpoint therapy resistance (*Xiong et al., 2020*). MS4A4A and MS4A6A are supported by the finding that APOE and TREM2 are up-regulated by MS4A (*Deming et al., 2019*). In contrast, feature genes selected by RidgeRegression upon the same six biomarkers were expressed in diverse cells (*Appendix 2—figure 7*). These confirm that BAHSIC can quite reliably select feature genes upon target genes.

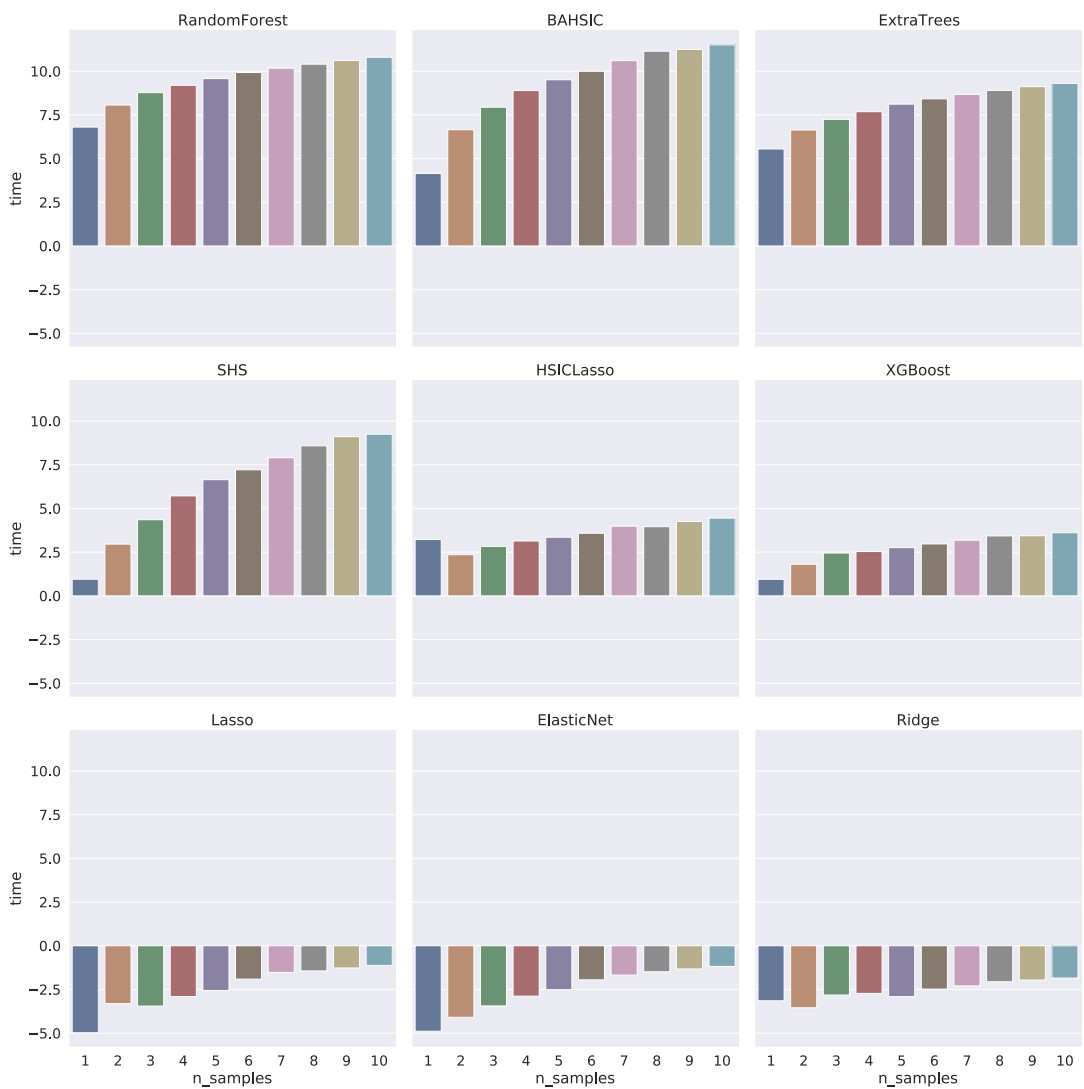

**Appendix 2—figure 1.** Time consumption of feature selection algorithms increased mildly when the sample size increased from 1000 to 10,000 (the *X*-axis indicates the sample size; 1–10 indicate 1000–10,000, respectively). The *Y*-axis indicates time (s) in the log2 form. The log2 form makes the time consumption of Lasso, ElasticNet, and Ridge have negative values (<1 s).

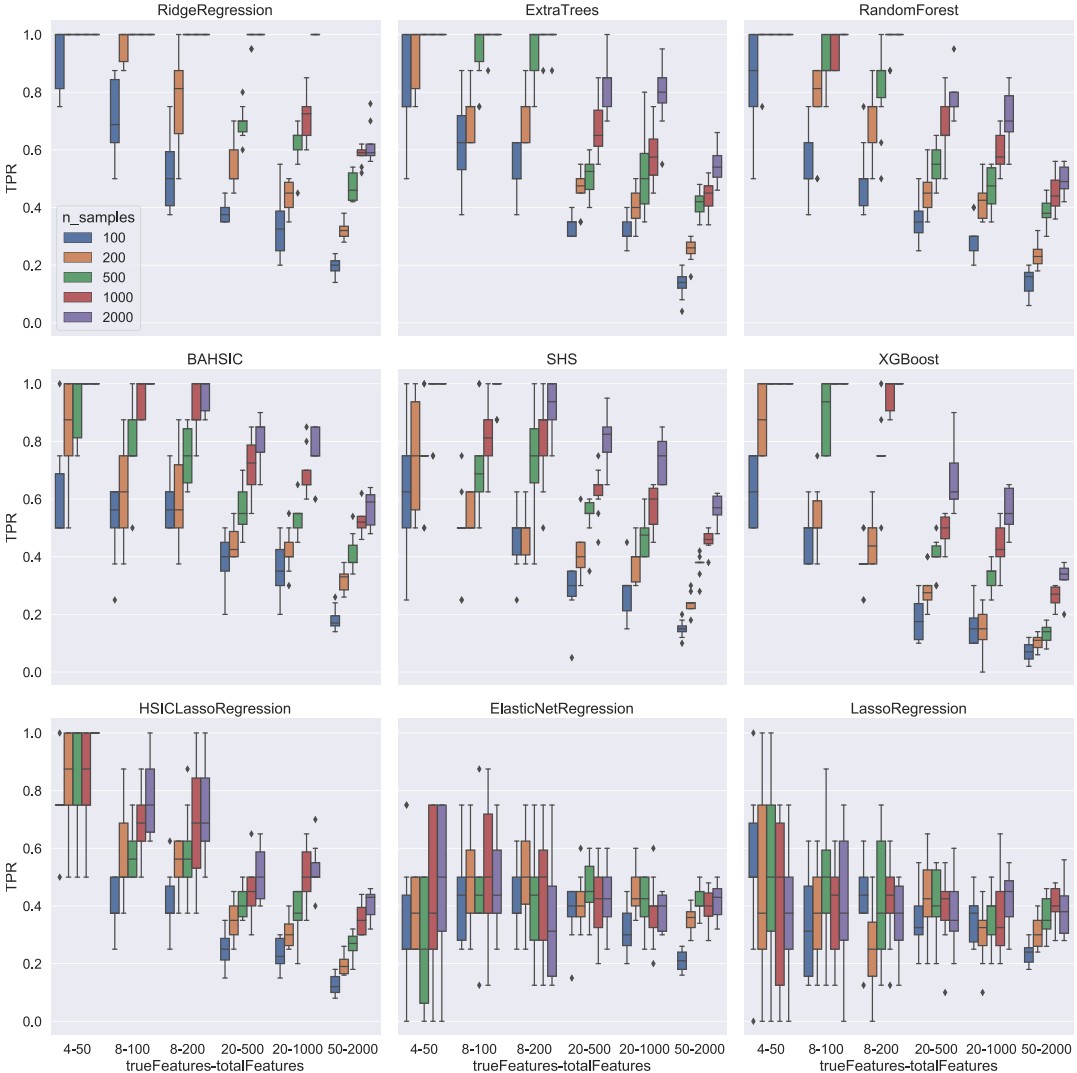

**Appendix 2—figure 2.** Feature selection results using synthetic data. Colors indicate different sample sizes (see the inset in the top-left panel). The *X*-axis (depicted under the bottom panels) indicates different schemes. For example, 4–50 means that there are 50 candidates (i.e. total features), 4 of which are chosen randomly to generate the response variable, and feature selection should select the 4 features (i.e. true features) from the 50 candidates upon the response variable. The *Y*-axis indicates true positive rate (TPR). For simple schemes (e.g. 4–50), some algorithms reached a TPR of 1.0. For complex schemes (e.g. 50–2000, selecting 50 features from 2000 candidates upon the response variable), some algorithms (especially BAHSIC) reached a TPR of 0.6.

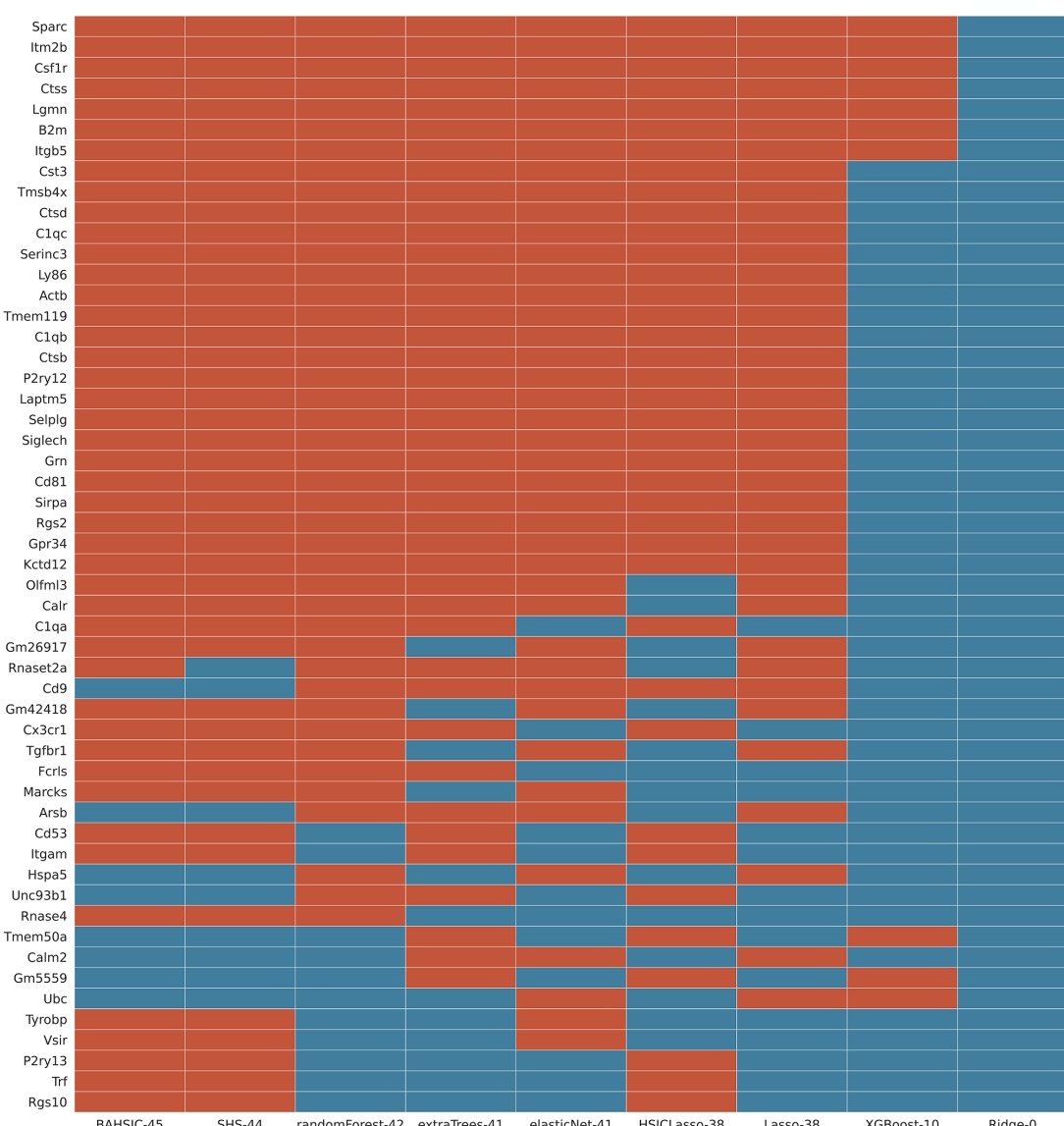

**Appendix 2—figure 3.** This screenshot showed a feature selection result (the superset of feature genes selected by ≥3 algorithms from the mouse microglia dataset) when all 9 algorithms were used. The *Hexb* gene was the target gene, and each algorithm selected 50 feature genes from the 3000 candidate genes expressed in most cells. BAHSIC, SHS, RandomForest, ExtraTrees, ElasticNet selected highly overlapping feature genes, many of which are microglia biomarkers in mice (*Appendix 2—figure 4*). The numbers right side of algorithm names indicate genes overlapping with the superset.

| Human (CD74) (1) | Human (ACTB) (33) | Mouse (ACTB) (7) | Mouse (Gm42418) (1) | Mouse (Hexb) (4) |
|---|---|---|---|---|
| A2M | A2M | C1qa | Abce1 | Actb |
| AC011481.3 | ACTBP11 | C1qb | AC124744.2 | Arsb |
| AC025857.2 | ACTG1P21 | C1qc | Actb | B2m |
| ACTB | ATP5J2 | Calm1 | Aldoa | C1qa |
| ACTG1 | B2M | Calr | Apoe | C1qb |
| APOE | BECN1 | Cd81 | Cnpy2 | C1qc |
| B2M | C1QB | Cfl1 | Coro1a | Calm2 |
| BOD1L1 | C3 | Csf1r | Cryl1 | Calr |
| C1QA | CALM1 | Cst3 | Csf1r | Cd53 |
| C1QB | CALM2 | Ctsb | Ctsd | Cd81 |
| C3 | CCL3 | Ctsd | Cx3cr1 | Cd9 |
| C3AR1 | CD74 | Ctss | Cyth4 | Csf1r |
| CCL2 | CD81 | Cx3cr1 | Fcgr3 | Cst3 |
| CCL4 | CSF1R | Eef1a1 | Gm13339 | Ctsb |
| CD14 | CST3 | Epb41l2 | Gm15725 | Ctsd |
| CD81 | CX3CR1 | Fam102b | Gm21198 | Ctss |
| CH25H | CYBB | Fcer1g | Gm22213 | Cx3cr1 |
| CSF1R | CYFIP1 | Glul | Gm26917 | Fcer1g |
| CSGALNACT1 | DUSP7 | Gm11942 | Gm29047 | Fcrls |
| CST3 | FBXL18 | Gm15725 | Hbb-bs | Gm15344 |
| DENND3 | FCER1G | Gm26917 | Hexb | Gm26917 |
| FTL | FCGR3A | Gm42418 | Kpnb1 | Gm29216 |
| FYB1 | FTH1 | Golm1 | Laptm5 | Gm42418 |
| GLIPR1 | FTL | Hexb | Lpin2 | Gpr34 |
| HECTD1 | HLA-DRA | Itgb5 | Lrrc3 | Grn |
| HLA-B | IFNAR1 | Itm2b | Malat1 | Hspa5 |
| HLA-C | ITM2B | Laptm5 | Marcks | Itgam |
| HLA-DMB | MARCKS | Lcp1 | Mcrip1 | Itgb5 |
| HLA-DPA1 | MIR3064 | Ly86 | Mir6236 | Itm2b |
| HLA-DPB1 | PFN1 | Man2b1 | Nek7 | Kctd12 |
| HLA-DQA1 | PRDX4 | Marcks | Olfml3 | Laptm5 |
| HLA-DQB1 | SNORA70 | Olfml3 | P2ry12 | Lgmn |
| HLA-DRA | SPP1 | P2ry12 | Pmepa1 | Ly86 |
| HLA-DRB1 | SRGN | Rassf2 | Reep5 | Marcks |
| HLA-DRB6 | SSB | Rgs10 | Selenop | Olfml3 |
| HLA-E | TMSB4X | Rnase4 | Serinc3 | P2ry12 |
| IL1B | TMSB4XP8 | Selenop | Sipa1 | P2ry13 |
| ITM2B | TUBA1B | Selplg | Ski | Rgs10 |
| LAPTM5 | TYROBP | Serinc3 | Sparc | Rgs2 |
| MALAT1 | VSIG4 | Siglech | Stat6 | Rnase4 |
| NDUFS5 | ZFP36L1 | Slco2b1 | Syngr2 | Rnaset2a |
| NEAT1 | ZSWIM6 | Sparc | Tmem119 | Selplg |
| PLD4 | | Tgfbr1 | Tmem19 | Serinc3 |
| PSAP | | Tmem119 | Ubc | Siglech |
| PTGS1 | | Tmsb4x | Vsir | Sirpa |
| RNASE6 | | Tyrobp | Wdr1 | Sparc |
| RNASET2 | | Ubl3 | Zfhx3 | Tgfbr1 |
| SLC15A2 | | | | Tmem119 |
| SNORA31 | | | | Tmsb4x |
| SORL1 | | | | Tyrobp |
| SPP1 | | | | Unc93b1 |
| TBC1D12 | | | | Vsir |
| TMSB4X | | | | |
| TREM2 | | | | |
| VSIR | | | | |

**Appendix 2—figure 4.** Feature genes selected by ≥3 algorithms in microglia from humans and mice using *CD74, ACTB, ACTB, Gm42418*, with *Hexb* as the target gene. The number right side of each target gene indicates the rank of its transcript's variance in the 3000 candidate genes. A large variance indicates that the gene may be important in the examined cells. Many feature genes are microglia biomarkers, indicating that feature genes

*Appendix 2—figure 4 continued*
selected by ≥3 algorithms are biologically rational. Annotated microglia biomarkers in humans and mice are marked in red (*Patir et al., 2019*).

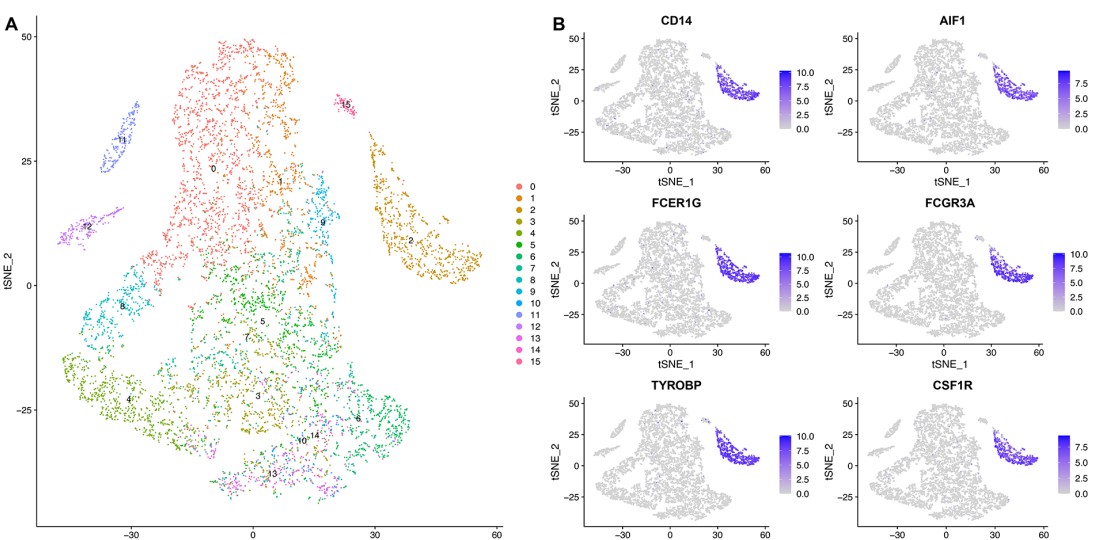

**Appendix 2—figure 5.** Cell types and cells that express macrophage biomarker genes. (**A**) The tSNE plot shows all cells isolated from the human glioblastoma (*Neftel et al., 2019*). Region 2 (the blue area in (**B**)) is macrophages. (**B**) The six macrophage biomarker genes were exclusively expressed in macrophages (the blue area).

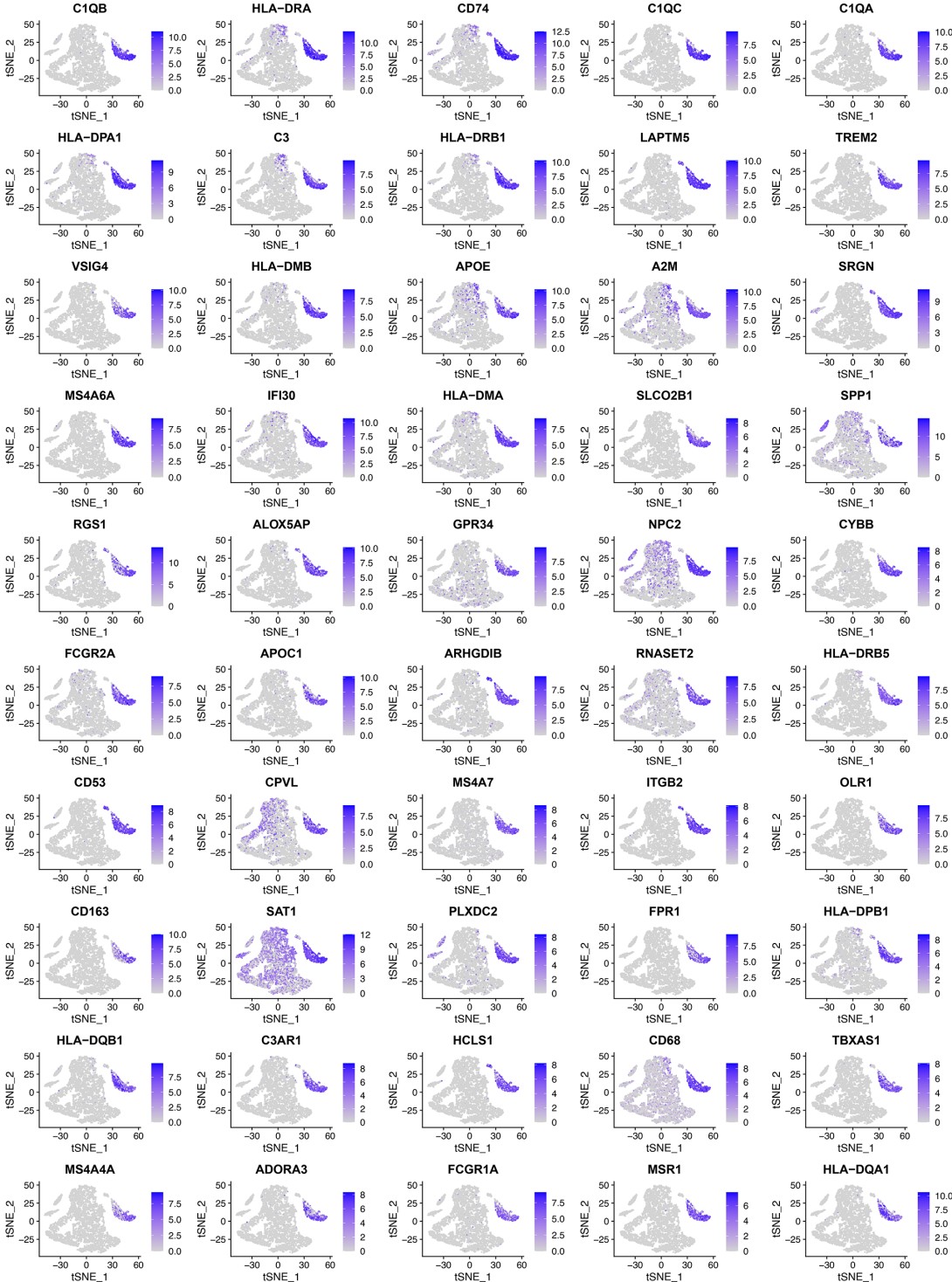

**Appendix 2—figure 6.** These tSNE plots show that nearly all of the 50 feature genes were exclusively expressed in macrophages. These feature genes were selected by BAHSIC using six macrophage biomarkers (CD14, AIF1, FCER1G, FCGR3A, TYROBP, and CSF1R) as the target genes. They include genes involved in macrophage activation (e.g. C1QA, CD74, TREM2) and multiple class II major histocompatibility complex (MHC) genes (e.g. HLA-DMA, HLA-DPA1). The interactions between CD74 and MHC-II genes (CD74 is an MHC class II chaperone) probably contribute to the co-selection of these genes.

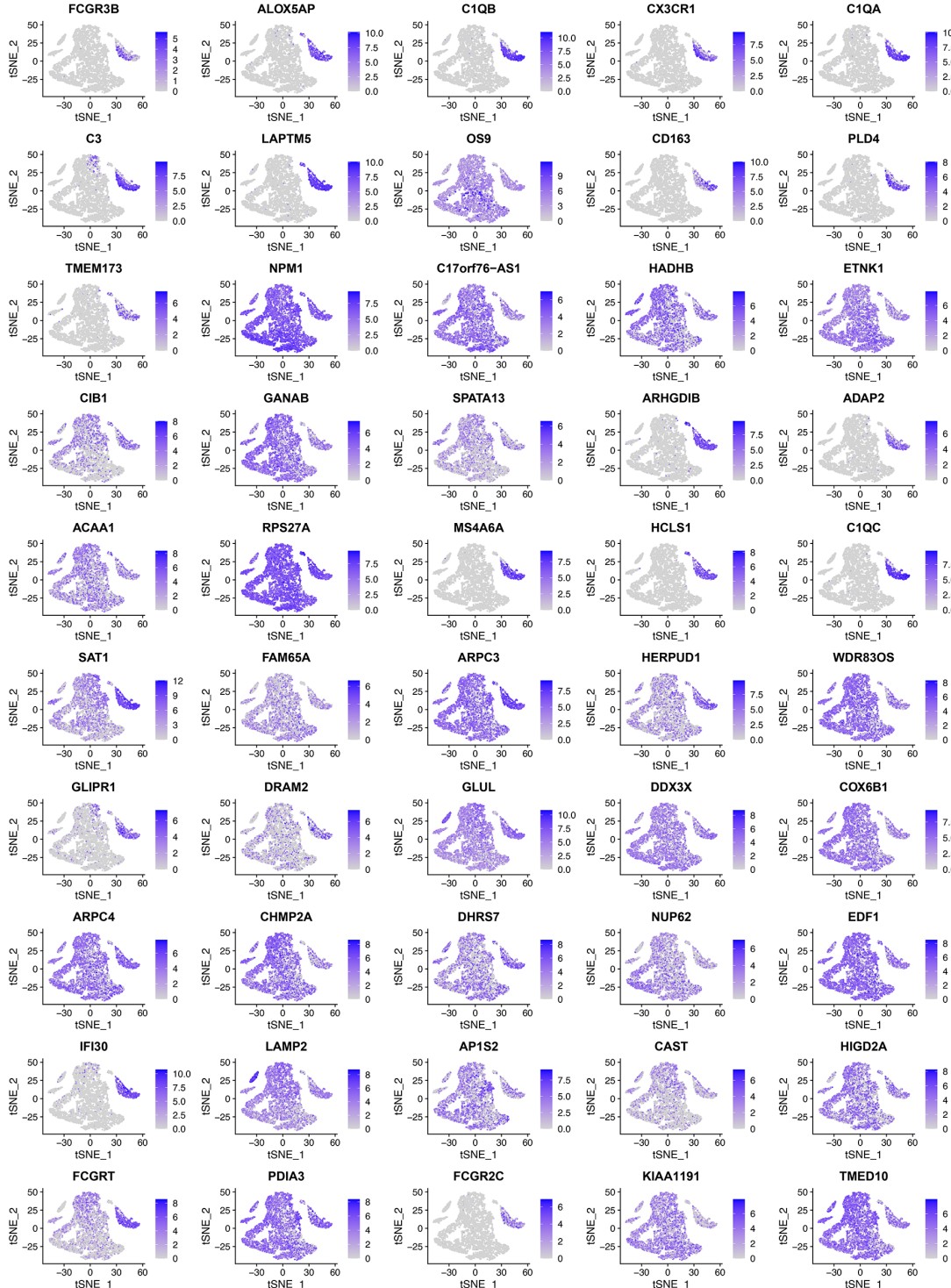

**Appendix 2—figure 7.** When RidgeRegression was used to select 50 feature genes using the same six macrophage markers (CD14, AIF1, FCER1G, FCGR3A, TYROBP, and CSF1R) as target genes, as these tSNE plots show, many feature genes were expressed in diverse cells instead of macrophages.

## Appendix 3

## Causal discovery algorithms and benchmarking

### 1. Causal discovery methods

CausalCell integrates four causal discovery methods – PC, GES, GSP, and DAGMA-nonliear – which are representative constraint-based, score-based, hybrid, and continuous optimization-based methods. Constraint-based methods identify causal interactions in a set of variables in two steps: skeleton estimation and orientation. Score-based methods assign a score function (e.g. the Bayesian information criterion) to each potential causal network and optimize the score via greedy approaches. Hybrid methods combine score-based methods and CI tests. Continuous optimization-based methods recast the combinatoric graph search problem as a continuous optimization problem. The PC and GSP algorithms can be combined with different CI tests.

To benchmark the performance of different CI tests, we combined 10 CI tests with the parallel version of the PC algorithm (i.e. the *pc* function in the R package *pcalg*, with the default setting *skel. method*="stable") (*Le et al., 2019*). The results show that kernel-based CI tests (especially the two DCC CI tests) outperform other CI tests (*Appendix 3—table 1*; *Appendix 3—figures 1 and 2*). To evaluate the score-based and hybrid methods GES (https://cran.r-project.org/web/packages/pcalg/index.html) and GSP (https://github.com/uhlerlab/causaldag; *Chickering, 2003*; *Solus et al., 2021*; *Squires, 2018*), we compared PC+DCC.gamma, GES, and GSP+DCC.gamma. The results show that PC+DCC.gamma and GSP+DCC.gamma have comparable network accuracy and time consumption, and both are more accurate but more time-consuming than GES (*Appendix 3—figures 3–6*).

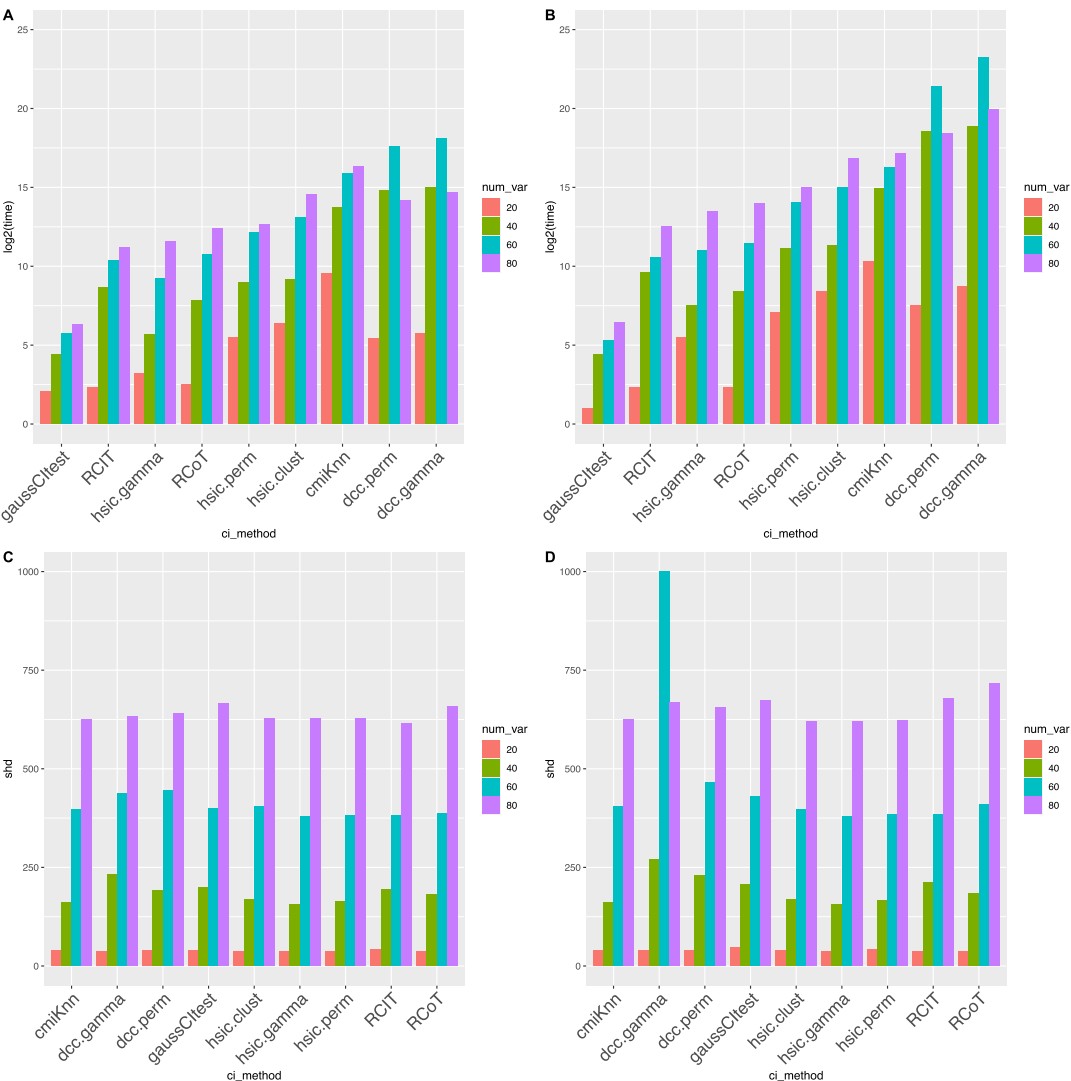

**Appendix 3—figure 1.** Causal discovery performance that was tested using synthetic data. DCC.perm and HSIC. gamma are the most time-consuming algorithms, but all algorithms have similar accuracy. The sample size is 500 (AB) or 1000 (CD), and the variable number ranges from 20 to 80. (AC) show the time consumption (in second) of different algorithms, and (BD) show structural Hamming distance (SHD) values. '*' in (CD) indicates that the algorithm did not finish running in 6 weeks. These two cases, and the two cases in (**A**) where DCC.perm and DCC. gamma took more time when there were 60 variables than when there were 80 variables, were anomalies caused by synthetic data. When testing using real scRNA-seq data, no such anomalies occurred.

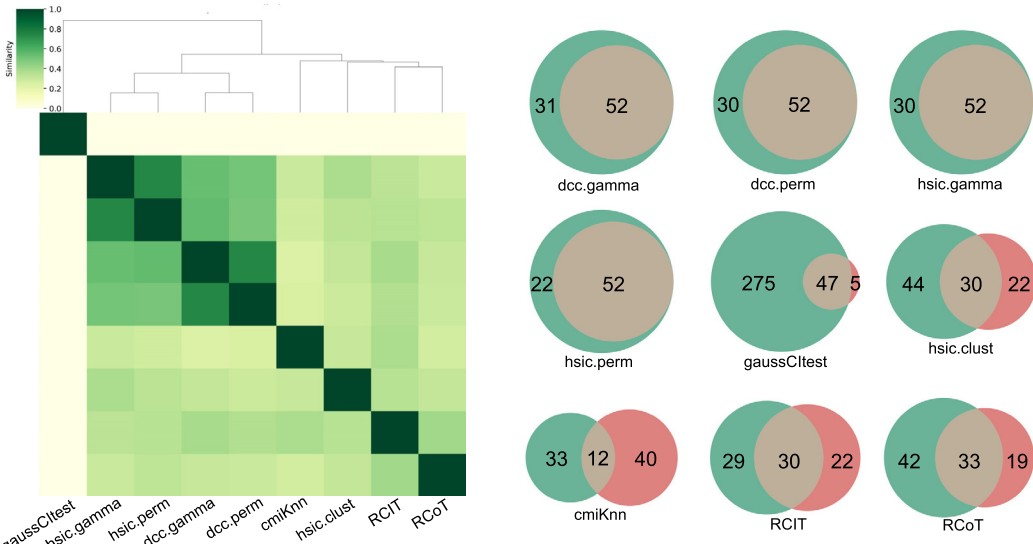

**Appendix 3—figure 2.** Consistency between each causal network and the consensus network. (**A**) In the cluster map of different CI tests, darker colors indicate a higher similarity of networks. The networks of HSIC.gamma, HSIC.perm, HSIC.gamma, and HSIC.perm have the highest similarity values, thus sharing the most similar structures. We used the four networks to build a consensus network, which was assumed most close to the ground truth. (**B**) For interactions inferred by each algorithm (green circle), we checked how many interactions overlap the interactions in the consensus network (pink circle). The true positive rate (TPR) of DCC.gamma, DCC.perm, HSIC.gamma, HSIC.perm, GaussCItest, HSIC.clust, cmiKnn, RCIT, and RCoT were 62.7%, 63.4%, 63.4%, 70.3%, 14.6%, 40.5%, 26.7%, 50.8%, and 44.0%, respectively, confirming that Hilbert-Schmidt independence criterion (HSIC) and distance covariance criteria (DCC) are better than others and that it is reasonable to use the consensus network generated upon the four algorithms' networks to evaluate algorithms' performance.

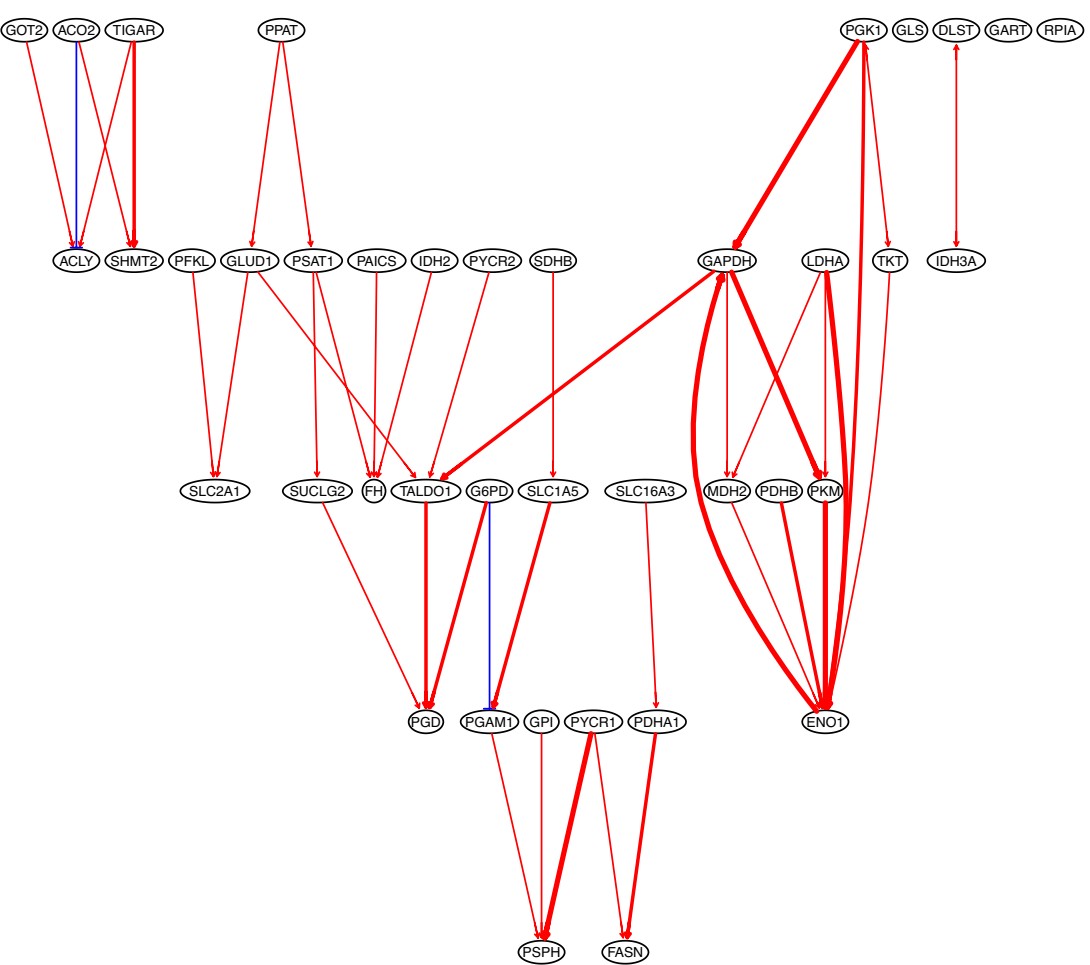

**Appendix 3—figure 3.** The causal relationships between genes in the WP4290 pathway in the cell line H838 inferred using 200 cells (alpha = 0.1). The color bar indicates fold changes of genes in the case dataset compared with in the control dataset.

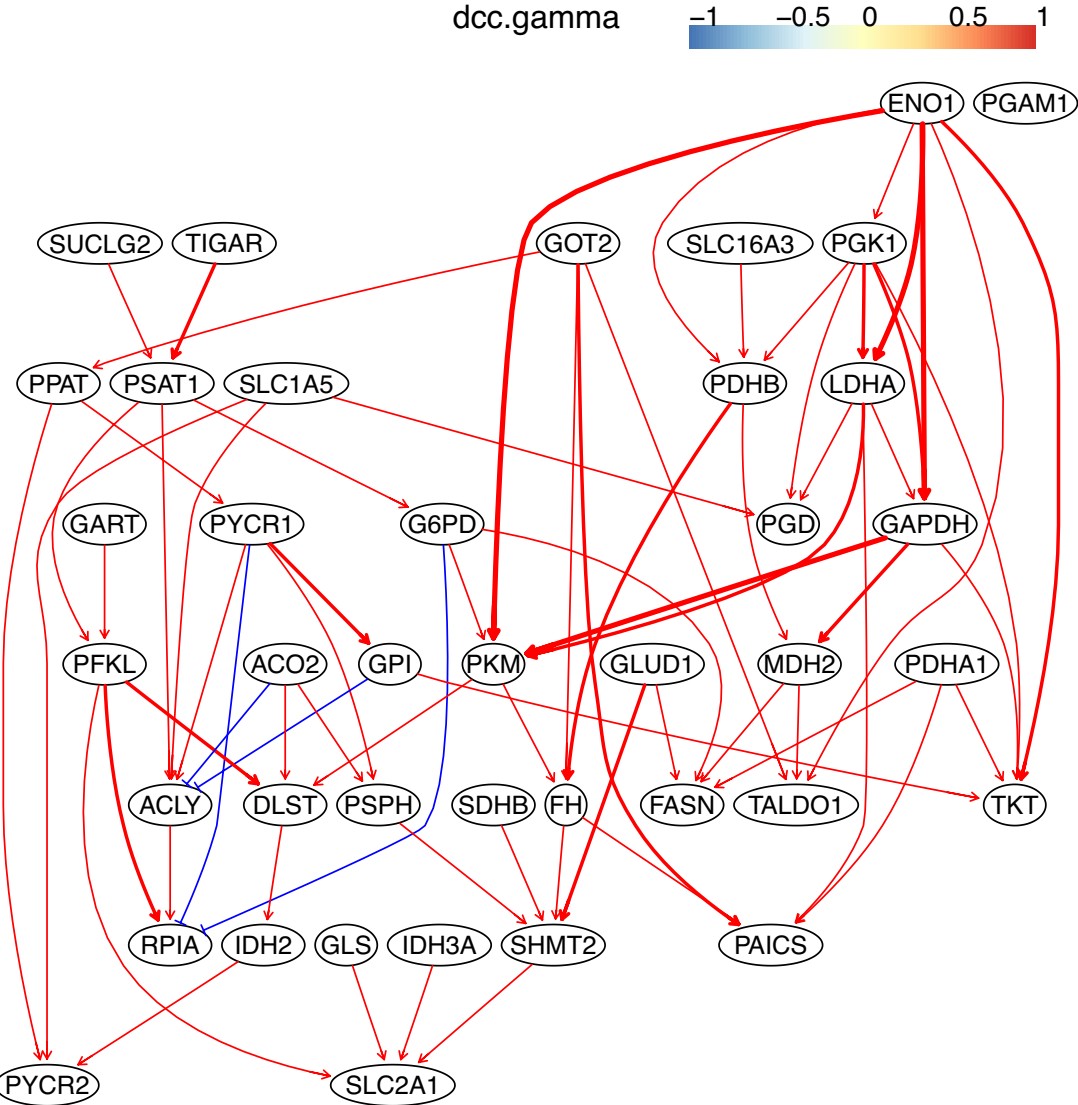

**Appendix 3—figure 4.** The causal relationships between genes in the WP4290 pathway in the cell line H838 inferred using 400 cells (alpha = 0.1).

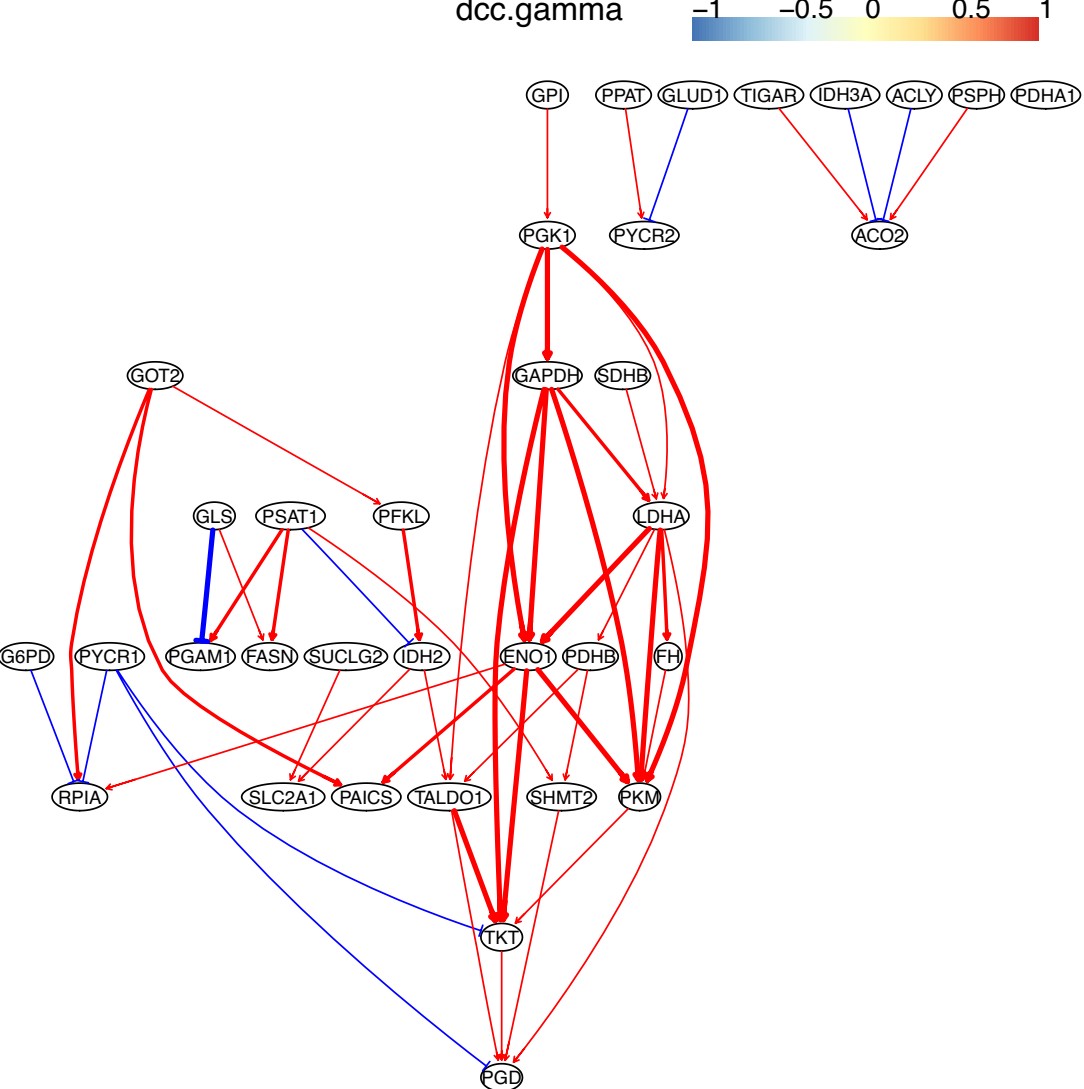

**Appendix 3—figure 5.** The causal relationships between genes in the WP4290 pathway in the cell line H838 inferred using 600 cells (alpha = 0.1).

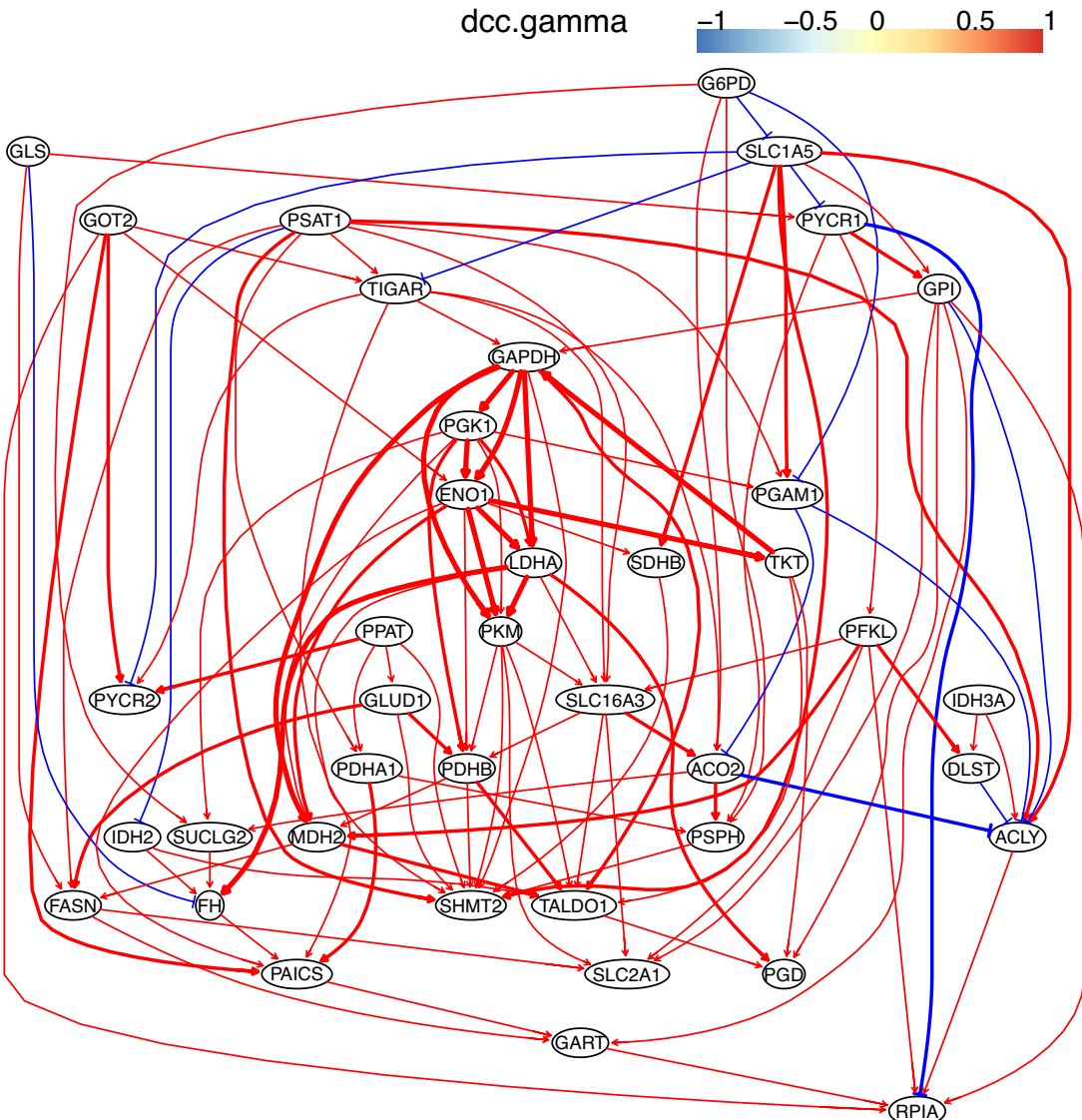

**Appendix 3—figure 6.** The causal relationships between genes in the WP4290 pathway in the cell line H838 inferred using 800 cells (alpha = 0.1). More cells make more relationships be inferred, but relationships with high significance (with thick arrows) are stable.

Further, we benchmarked six continuous optimization-based methods (NOTEARS-linear, NOTEARS-nonlinear, DAGMA-linear, DAGMA-nonlinear, GOLEM, and DAG_GNN) (*Bello et al., 2022a*; *Zheng et al., 2018*), and two linear non-Gaussian acyclic model methods (ICLiGNAM and DirectLiGNAM). We compared the performance of these methods with PC+DCC.gamma and PC+GaussCItest. Continuous optimization-based methods, especially DAGMA-nonlinear (https://github.com/kevinsbello/dagma; *Bello et al., 2022b*), perform well when relationships between variables are free of missing variables and missing values, otherwise they perform poorly and underperform PC+DCC.gamma. All benchmarking used both simulated data and multiple scRNA-seq datasets, especially the five lung cancer cell lines (A549, H1975, H2228, H838, HCC827) from the CellBench benchmarking dataset (*Tian et al., 2019*). Genes differentially expressed in these cell lines were determined upon gene expression in the lung alveolar cells (*Travaglini et al., 2020*).

## 2. Partial correlation-based CI test
### GaussCItest
Gauss CI test examines CI using partial correlation, assuming that all variables are multivariate Gaussian. The partial correlation coefficient $\rho_{XYZ}$ is zero if and only if $X$ is conditionally independent

of $Y$ given $Z$ (**Kunihiro et al., 2004**). $H_0$ is $\hat{\rho}_{XYZ} = 0$, $H_1$ is $\hat{\rho}_{XYZ} \neq 0$, and a hypothesis test (p<0.05) decides whether two variables are conditionally independent given $Z$. We used the *gaussCItest* function in the R package *pcalg* with default parameters (https://cran.r-project.org/web/packages/pcalg/index.html).

## 3. HSIC-based CI test

HSIC is a measure of dependency between two variables; HSIC $(X, Y) = 0$ if $X$ and $Y$ are unconditionally independent. Performing two extra transformations can determine if $X$ and $Y$ are conditionally independent given the conditioning set $Z$: first, performing nonlinear regressions for $X$ and $Z$ and for $Y$ and $Z$, respectively, to generate the residuals $X_{resid}$ and $Y_{resid}$ based on $Z$; then, calculating HSIC $(X_{resid}, Y_{resid})$ that indicates whether $X$ and $Y$ are conditionally independent given the conditioning set $Z$ $(X \perp\!\!\!\perp Y|Z)$ (**Verbyla et al., 2017**). We used the *gam*() function in the R package *mgcv* to build the nonlinear regression model and used the three HSIC-based functions (with default parameters unless otherwise specified) in the R package *kpcalg* (https://cran.r-project.org/web/packages/kpcalg/index.html) to perform CI test.

### hsic.perm

In practice, HSIC $(X, Y)$ may be slightly larger than 0.0 when $X$ and $Y$ are independent, making it hard to judge whether $X$ and $Y$ are independent. *hsic.perm* uses a permutation test to solve this problem by assuming that permuting $Y$ removes any dependency between $X$ and $Y$. We used the *hsic.perm* function to permute $Y$ 100 times to calculate HSIC $(X, Y_{perm})$, then we compared them with HSIC $(X, Y)$. The p-value was the fraction of times HSIC $(X, Y_{perm})$ was smaller than the HSIC $(X, Y)$.

### hsic.gamma

We used the *hsic.gamma* function to fit a gamma distribution: Gamma($\alpha$, $\theta$) of the HSIC under the null hypothesis. The shape parameter $\alpha$ and the scale parameter $\theta$ were calculated using the equation:

$$\alpha = \frac{E\left[\hat{H}_{X,Y}\right]^2}{Var\left(\hat{H}_{X,Y}\right)}, \; \theta = \frac{Var(\hat{H}_{X,Y})}{E[\hat{H}_{X,Y}]}.$$

A p-value was obtained as an upper-tail quantile of HSIC $(X, Y)$.

### hsic.clust

First, samples were clustered using the R function *kmeans*() by calculating the Euclidean distance between the $Z$ coordinates of samples; then, $Y$ was permutated based on the clustered $Z$. Within each $Z$, cluster $Y_{perm}$ was generated, ensuring that the permuted samples break the dependency between $X$ and $Y$ but retain the dependency between $X$ and $Y$ on $Z$. After permutation, a p-value was calculated to make a statistical decision.

## 4. Distance covariance-based CI test

Distance covariance is an alternative to HSIC for measuring independence (**Székely and Rizzo, 2009**; **Székely et al., 2007**). We used two DCC-based functions *dcc.perm* and *dcc.gamma* (with default parameters) in the R package *kpcalg* (https://cran.r-project.org/web/packages/kpcalg/index.html) to perform CI test. Similar to HSIC-based algorithms, the two functions directly calculate DCC $(X, Y)$ for an UI test, then, the nonlinear regression is performed, next, DCC $(X_{resid}, Y_{resid})$ is calculated for a CI test and a statistical decision (**Verbyla et al., 2017**).

### dcc.perm

This program is similar to *hsic.perm* and uses a permutation test to estimate a p-value. The DCC statistic is calculated in each permutation, and finally, a statistical decision is made based on the p-value. We used the *dcov.test* function (with default parameters) in the R package *energy* to calculate the statistic DCC in the permutation test. The p-value was the fraction of times that DCC($X, Y_{perm}$) was smaller than DCC($X, Y$).

### dcc.gamma

Similar to *hsic.gamma*, *dcc.gamma* uses the gamma distribution Gamma($\alpha$, $\theta$) of the DCC under the null hypothesis. The two parameters were estimated by

$$\alpha = \frac{E\left[\hat{D}_{X,\,Y}\right]^2}{Var\left(\hat{D}_{X,\,Y}\right)}, \theta = \frac{Var(\hat{D}_{X,\,Y})}{E[\hat{D}_{X,\,Y}]}$$

We used the dcov.gamma function (with default parameters) in the R package kpcalg to calculate the p-value. The p-value was obtained as an upper-tail quantile of DCC($X$, $Y$).

## 5. Approximation of KCIT

The KCIT is another powerful CI test (**Zhang and Peters, 2011**), but it is time-consuming for large datasets. Based on random Fourier features (**Rahimi and Recht, 2007**), two approximation methods (randomized conditional independence test, RCIT, and randomized conditional correlation test, RCoT) were proposed (**Strobl et al., 2019**). RCIT and RCoT approximate KCIT by sampling Fourier features, return p-values orders of magnitude faster than KCIT when the sample size is large, and may also estimate the null distribution more accurately than KCIT.

### RCIT

We used the *RCIT* function in the R package RCIT (with default parameters) (https://github.com/ericstrobl/RCIT; **Strobl, 2019**) to implement the randomized CI test.

### RCoT

RCoT often outperforms RCIT, especially when the size of the conditioning set is greater than or equal to 4. We used the *RCoT* function in the R package *RCIT* (with default parameters) (https://github.com/ericstrobl/RCIT; **Strobl, 2019**) to implement the RCoT.

## 6. Conditional mutual information-based CI test

Mutual information is used to measure mutual dependence between two variables. Conditional mutual information (CMI) is a measure based on mutual information, which is zero if and only if $X \perp\!\!\!\perp Y|Z$.

### CMIknn

*CMIknn* is a program that combines CMI with a local permutation scheme determined by the nearest-neighbor approach (**Runge, 2018**). We used the Python package *tigramite* (with default parameters) (http://github.com/jakobrunge/tigramite; **Runge, 2020**) to perform the CI test.

## 7. CI test based on generalized covariance measure

### GCM

GCM (https://cran.r-project.org/web/packages/GeneralisedCovarianceMeasure/index.html; **Peters and Shah, 2022**) is a CI test based on generalized covariance measure. It is also classified as a regression-based CI test because it is based on a suitably normalized version of the empirical covariance between the residual vectors from the regressions (**Shah and Peters, 2020**).

## 8. Benchmarking results

The time consumption, accuracy, sample requirement, and stability of the PC+ nine CI tests were evaluated (**Appendix 3—table 1**). First, we simulated eight datasets with known causal networks, whose variable numbers and sample sizes ranged from 20 to 80 (step = 20) and 500 to 1000 (step = 500), respectively, to evaluate causal discovery algorithms' time consumption, scalability, and accuracy. Algorithms based on the DCC kernel were more time-consuming than others (**Appendix 3—figure 1A,C**). Algorithms' accuracy was assessed based on the structural Hamming distance (SHD) between the inferred and the true networks (SHD = 0 indicates no difference). The networks of all algorithms showed similar SHD when the sample size was 500 (**Appendix 3—figure 1B**); the close performance was probably because synthetic data were generated using a few simple functions. When the sample size was increased from 500 to 1000, time consumption increased (but was not doubled), but SHD did not decrease (i.e. algorithms' performance did not increase) significantly, indicating that 500 cells may be adequate for causal discovery (**Appendix 3—figure 1D**).

Second, to further evaluate algorithms' accuracy, for each feature gene set, we merged causal networks generated by multiple good algorithms into a consensus network (multi-algorithm-based

consensus network), then compared the network of each algorithm with the consensus network (Main text-*Figure 2*; *Appendix 3—figure 2*). We used the SHD to define the difference between two networks, and the network with the shortest SHD with the consensus network is assumed to be the most accurate.

Third, to evaluate the impact of sample size on algorithms' performance, we ran the nine algorithms using 200 (instead of 300) H2228 cells. The results of 200 cells were poorer than the results of 300 cells (compared with the consensus network in Main text-*Figure 2* and *Appendix 3—figure 2*). Still, the two DCC algorithms performed the best and were less sensitive to the decreased sample size than the two HSIC algorithms. We also inferred interactions between genes in the 'Metabolic reprogramming in colon cancer' (WP4290) pathway using 200, 400, 600, and 800 cells in the H838 (*Appendix 3—figures 3–6*). We find that more cells make more interactions be inferred, but the interactions with high significance are quite stable.

Fourth, to evaluate algorithms' stability, we used the H2228 dataset to run the nine algorithms five times and estimated each algorithm's stability by computing the mean relative SHD of the five networks. The networks of gaussCItest have the smallest mean relative SHD and the networks of HSIC.perm, HSIC.clust, and DCC.perm have the largest mean relative SHD (*Appendix 3—table 1*). As DCC.perm and DCC.gamma are the most accurate algorithms, we examined whether their stability impairs their accuracy by checking the distribution of interactions in the five networks. DCC. gamma and DCC.perm inferred 127 and 143 interactions, 78% and 64.3% occurred stably in ≥4 networks, and many inconsistent interactions occurred in just one network (Main text-*Figure 3*), indicating that most interactions were stably inferred in multiple running. The networks of multiple running can be merged into a consensus network (multi-running-based consensus network), which can be used to examine which algorithm generated the most consistent networks.

Fifth, we compared the accuracy of PC+DCC.gamma, GES, and GSP+DCC.gamma using genes in the WikiPathways 'Metabolic reprogramming in colon cancer' (WP4290) and 600 cells in the A549, H2228, and H838 datasets. GSP+DCC.gamma (the significance level alpha = 0.01) inferred much more interactions than PC+DCC.gamma (alpha = 0.1) and GES (alpha = 0.1). The results indicate that PC+DCC.gamma (alpha = 0.1) and GSP+DCC.gamma (alpha = 0.05) have comparable accuracy and time consumption, and both are more accurate but time-consuming than GES (alpha = 0.1) (*Appendix 3—figures 7–12*).

**Appendix 3—table 1.** Performance of the nine causal discovery algorithms ('+++' and '+' indicate the best and worst, respectively).

| Algorithm | Time complexity* | Time consumption† | Accuracy ‡ | Sample size | Stability (mean of rSHD) § |
|---|---|---|---|---|---|
| GaussCItest | $O(q^3)$ | +++ | + | +++ | 0.075 (+++) |
| CMIknn | $O(n^2)$ | + | ++ | + | 0.25 (+) |
| RCIT | $O(d^2 \times n)$ | ++ | ++ | ++ | 0.12 (++) |
| RCoT | $O(d^2 \times n)$ | ++ | ++ | ++ | 0.12 (++) |
| HSIC.clust | $O\left(\sum_{k=1}^{K} m_k^3\right)$ | + | +++ | ++ | 0.26 (+) |
| HSIC.gamma | $O(n^3)$ | + | +++ | ++ | 0.12 (++) |
| HSIC.perm | $O(r \times n^3)$ | + | +++ | ++ | 0.28 (+) |
| DCC.gamma | $O(n^3)$ | + | +++ | +++ | 0.14 (++) |
| DCC.perm | $O(r \times n^3)$ | + | +++ | +++ | 0.24 (+) |
| GCM | Depending on regression methods | + | ++ | ++ | |

*Appendix 3—table 1 Continued on next page*

*Appendix 3—table 1 Continued*

| Algorithm | Time complexity* | Time consumption† | Accuracy ‡ | Sample size | Stability (mean of rSHD) § |
|---|---|---|---|---|---|

*(a) Assuming a dataset has $n$ samples and the total dimension of $X,Y,Z$ is $q$. Generally, $q<<n$. (b) $r$ is the time of permutation. (c) $d$ is the number of random Fourier features, generally $d<n$. (d) The time complexity of the PC algorithm is $\frac{N^2(N-1)^{deg-1}}{(deg-1)!}$ where $N$ is the number of nodes and $deg$ is the maximal degree.

†Time-consuming levels are estimated upon simulated data (**Appendix 3—figure 1**).

‡Accuracy is estimated upon the lung cancer cell lines (Main text-**Figure 2**; **Appendix 3—figure 2**). We performed causal discovery using the nine algorithms five times for the H2228 cell line and obtained 9*5=45 causal networks.

§We estimated the stability of each algorithm's performance by computing the mean relative SHD for the five causal networks the algorithm generated using the equation: $\frac{2}{NG(NG-1)}\sum_{i=1}^{NG-1}\sum_{j=i+1}^{NG}\frac{SHD\left(G_i,G_j\right)}{\#edges\left(G_i\right)}$ In this equation, SHD(Gi, Gj) is the structural Hamming distance between causal network Gi and Gj, #edges(Gi) is the number of edges in Gi, and NG = 5 because each algorithm generates five causal networks.

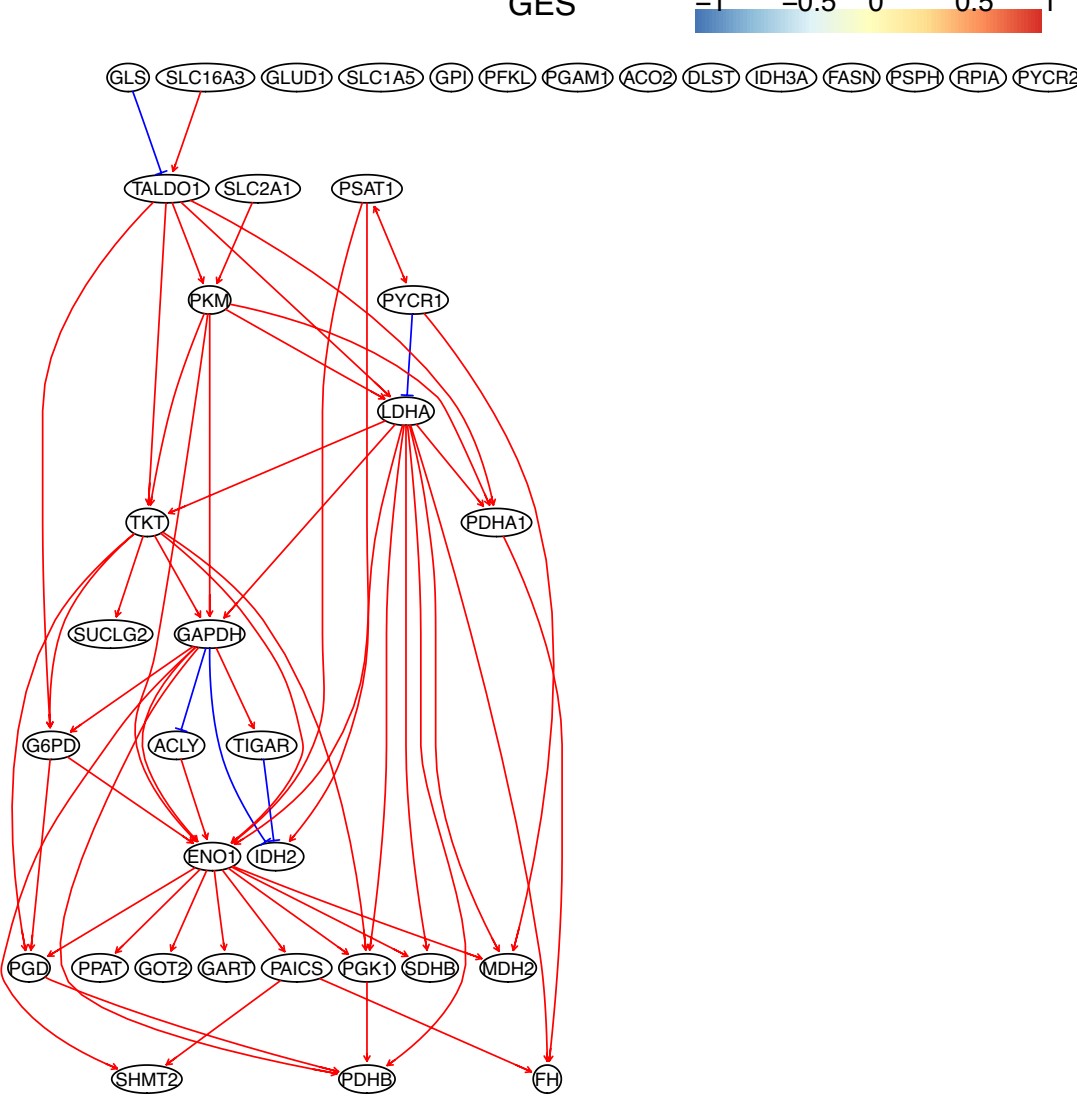

**Appendix 3—figure 7.** The causal relationships between genes in the WP4290 pathway in the cell line A549 inferred using the GES method and 600 cells (alpha = 0.1). Compared with the networks in these cells inferred using PC+DCC.gamma, here there are more isolated nodes.

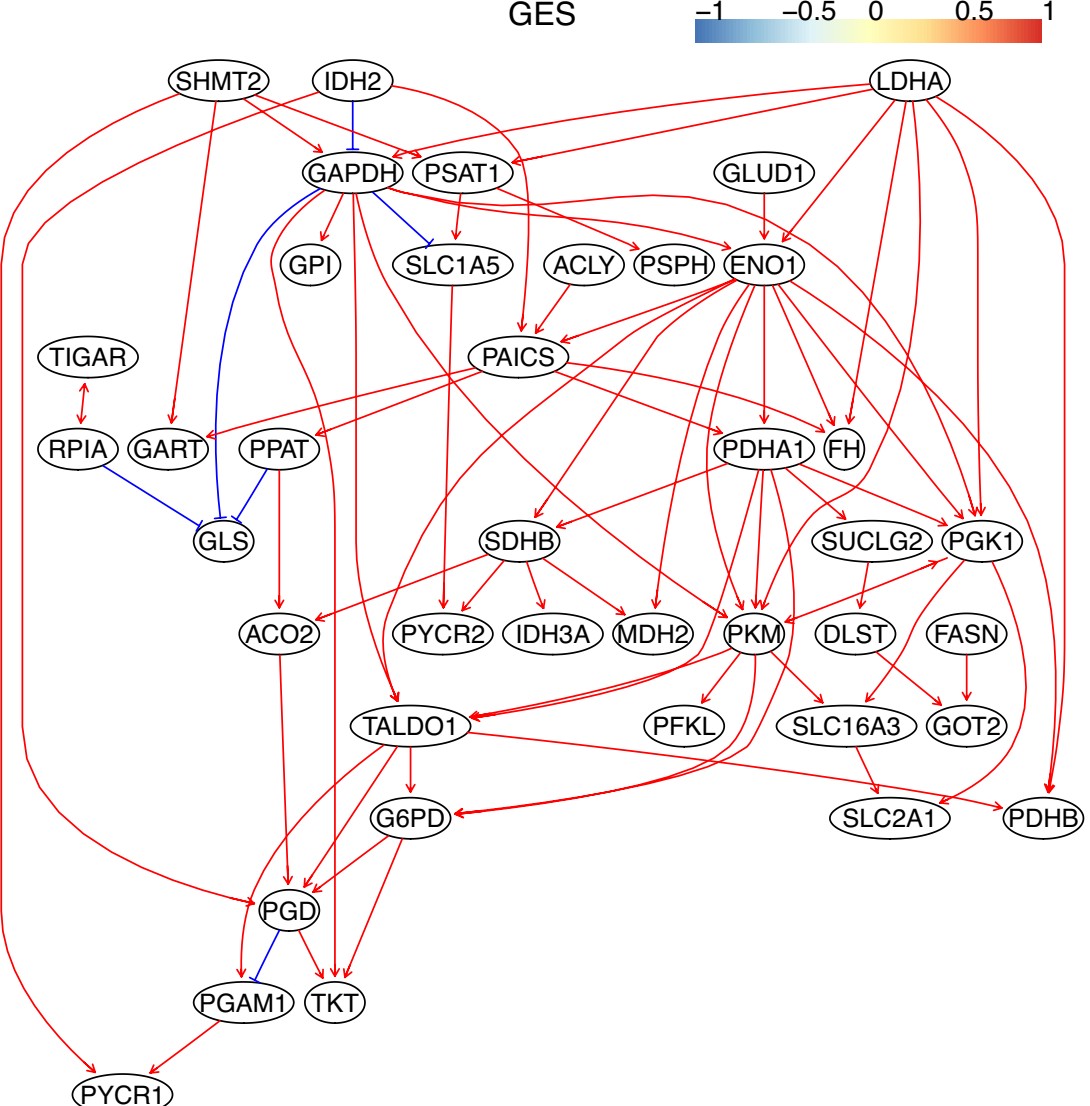

**Appendix 3—figure 8.** The causal relationships between genes in the WP4290 pathway in the cell line H2228 inferred using the GES method and 600 cells (alpha = 0.1).

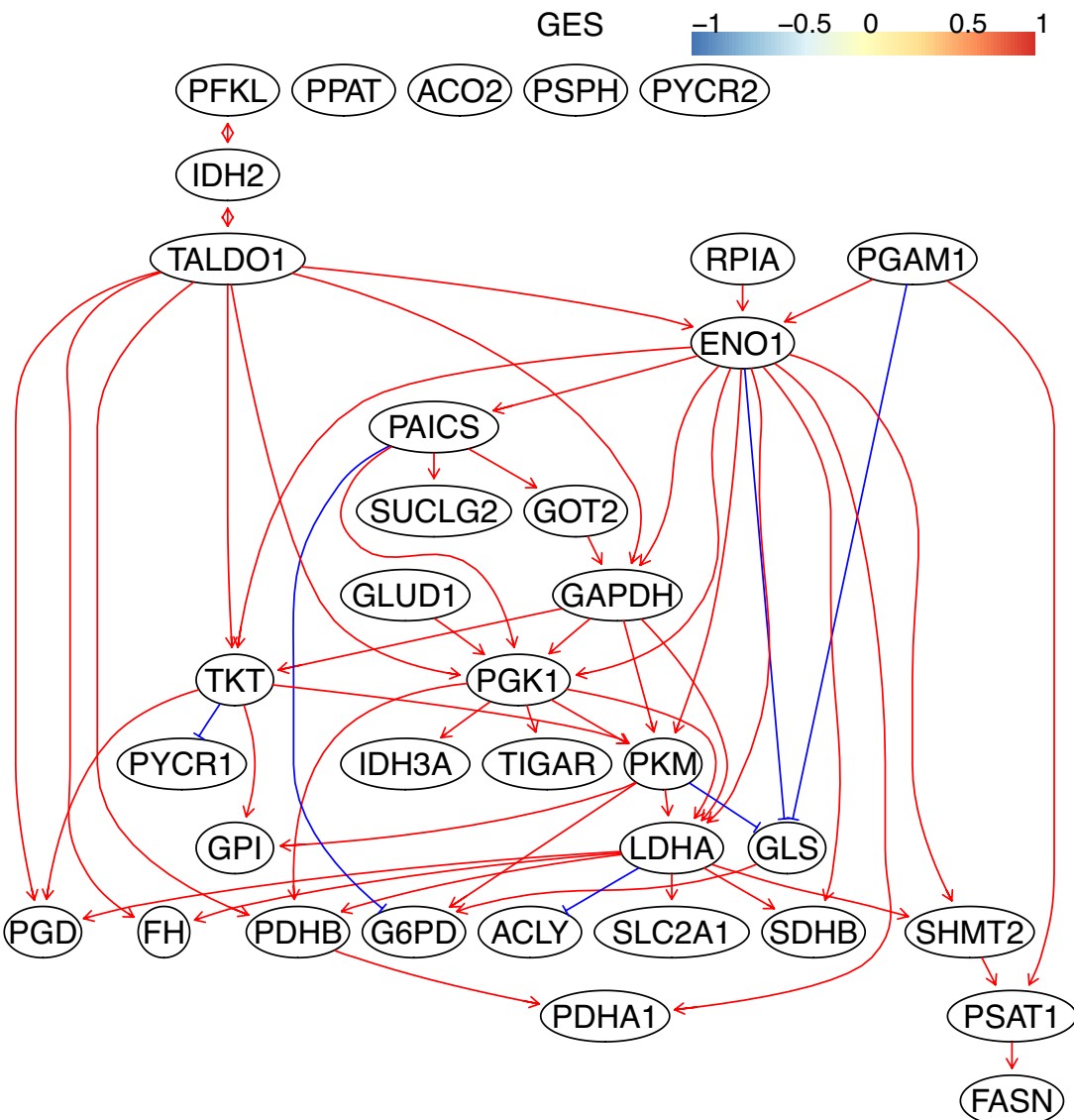

**Appendix 3—figure 9.** The causal relationships between genes in the WP4290 pathway in the cell line H838 inferred using the GES method and 600 cells (alpha = 0.1).

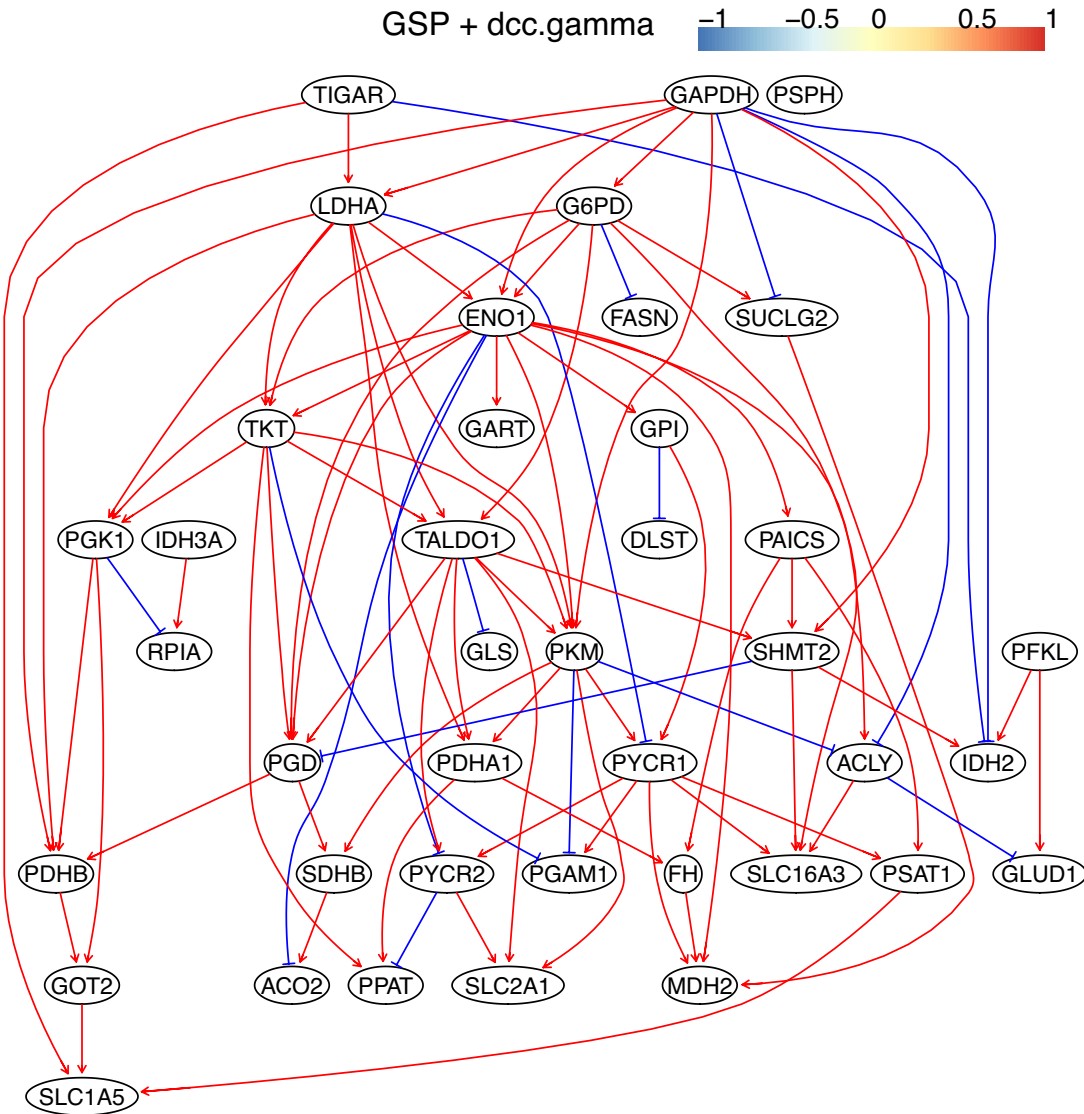

**Appendix 3—figure 10.** The causal relationships between genes in the WP4290 pathway in the cell line A549 inferred using the GSP+DCC.gamma and 600 cells (alpha = 0.05). Compared with the networks in these cells inferred using PC+DCC.gamma (*Appendix 4—figures 6–8*), more relationships are inferred even if the significance level is 0.05. The key features of the reprogrammed glucose metabolism (as indicated in the inferred networks of PC+DCC.gamma, see Appendix 4) also occur in the network.

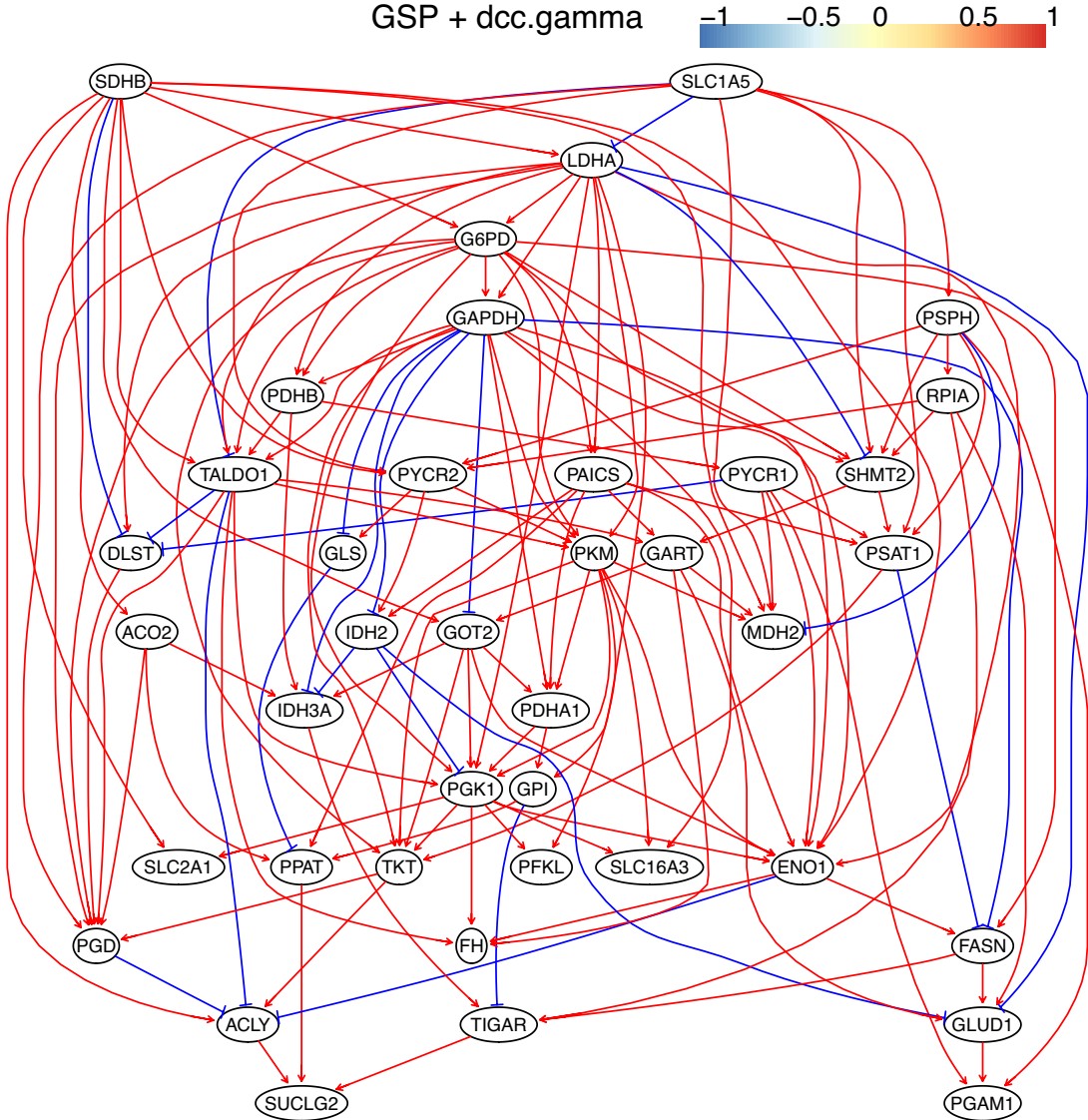

**Appendix 3—figure 11.** The causal relationships between genes in the WP4290 pathway in the cell line H2228 inferred using the GSP+DCC.gamma and 600 cells (alpha = 0.05). Compared with the networks in these cells inferred using PC+DCC.gamma (*Appendix 4—figures 6–8*), more relationships are inferred even if the significance level is 0.05. The key features of the reprogrammed glucose metabolism (as indicated in the inferred networks of PC+DCC.gamma, see Appendix 4) also occur in the network.

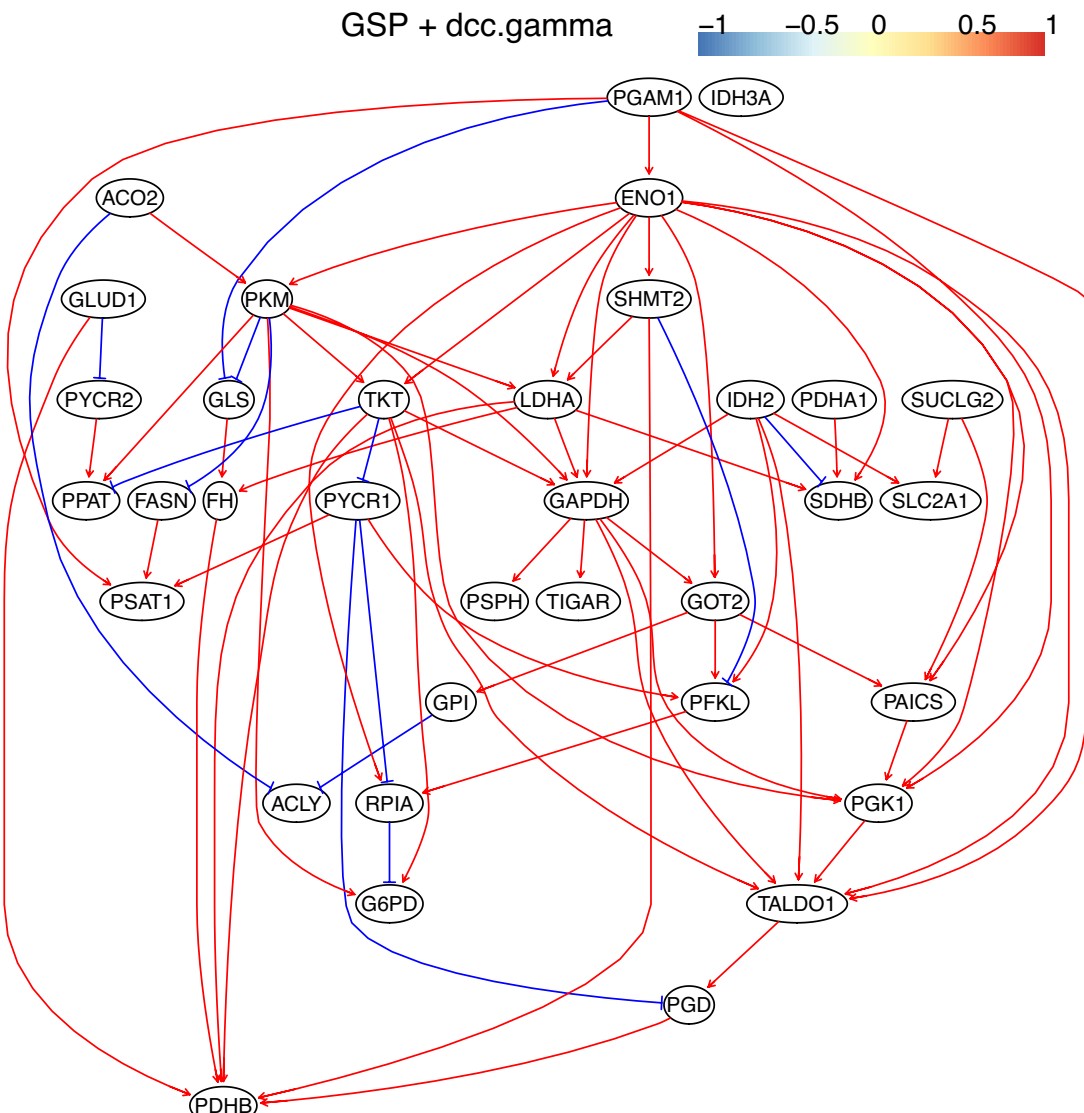

**Appendix 3—figure 12.** The causal relationships between genes in the WP4290 pathway in the cell line H838 inferred using the GSP+DCC.gamma and 600 cells (alpha = 0.05). Compared with the networks in these cells inferred using PC+DCC.gamma (*Appendix 4—figures 6–8*), more relationships are inferred even if the significance level is 0.05. The key features of the reprogrammed glucose metabolism (as indicated in the inferred networks of PC+DCC.gamma, see Appendix 4) also occur in the network.

## Appendix 4

### Evaluating the reliability and verifying causal discovery results

We evaluated the reliability of causal discovery by examining whether algorithms can differentiate interactions between genes in different cells. Inspired by using RNA spike-in to measure RNA-seq quality, we extracted the data of six MHC-II-related genes (HLA-DRB1, HLA-DMA, HLA-DRA, HLA-DPA, CD74, C3, which have the suffix _si to mark them) from the macrophage dataset (generated by Smart-seq2 sequencing) and the alveolar epithelial cell dataset (generated by 10x Genomics) to form two spike-in datasets. We mixed the spike-in dataset with the dataset of exhausted CD8 T cells and examined if the causal discovery was able to separate MHC-II genes and their interactions in the spike-in dataset from feature genes and their interactions in the exhausted CD8 T dataset. When the datasets contain sufficient cells (usually >300), the two DCC algorithms can discriminate genes and interactions in the two datasets quite well (*Appendix 4—figures 1 and 2*), indicating the power of causal discovery based on kernel-based CI tests. The inferred causal interactions can be verified using annotated protein interactions in the STRING database (https://string-db.org/). The results of our application cases indicate that many inferred interactions are supported by annotated protein interactions in the STRING database (*Appendix 4—figures 3 and 4*).

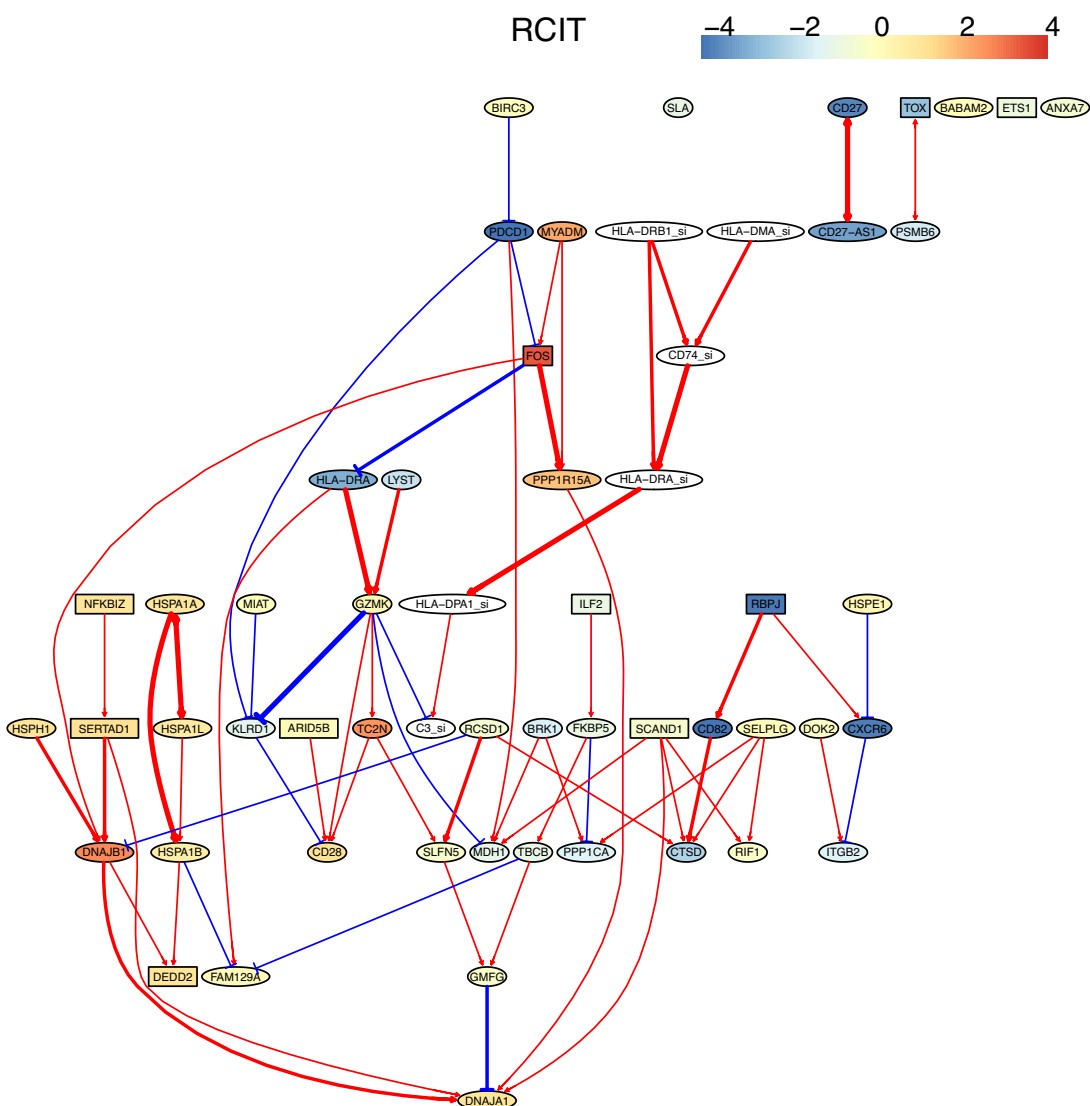

**Appendix 4—figure 1.** Multiple algorithms can identify genes and their relationships in the spike-in dataset from genes and their relationships in the primary dataset. The six genes and their relationships were identified in the network of RCIT, but the network has multiple orphan nodes.

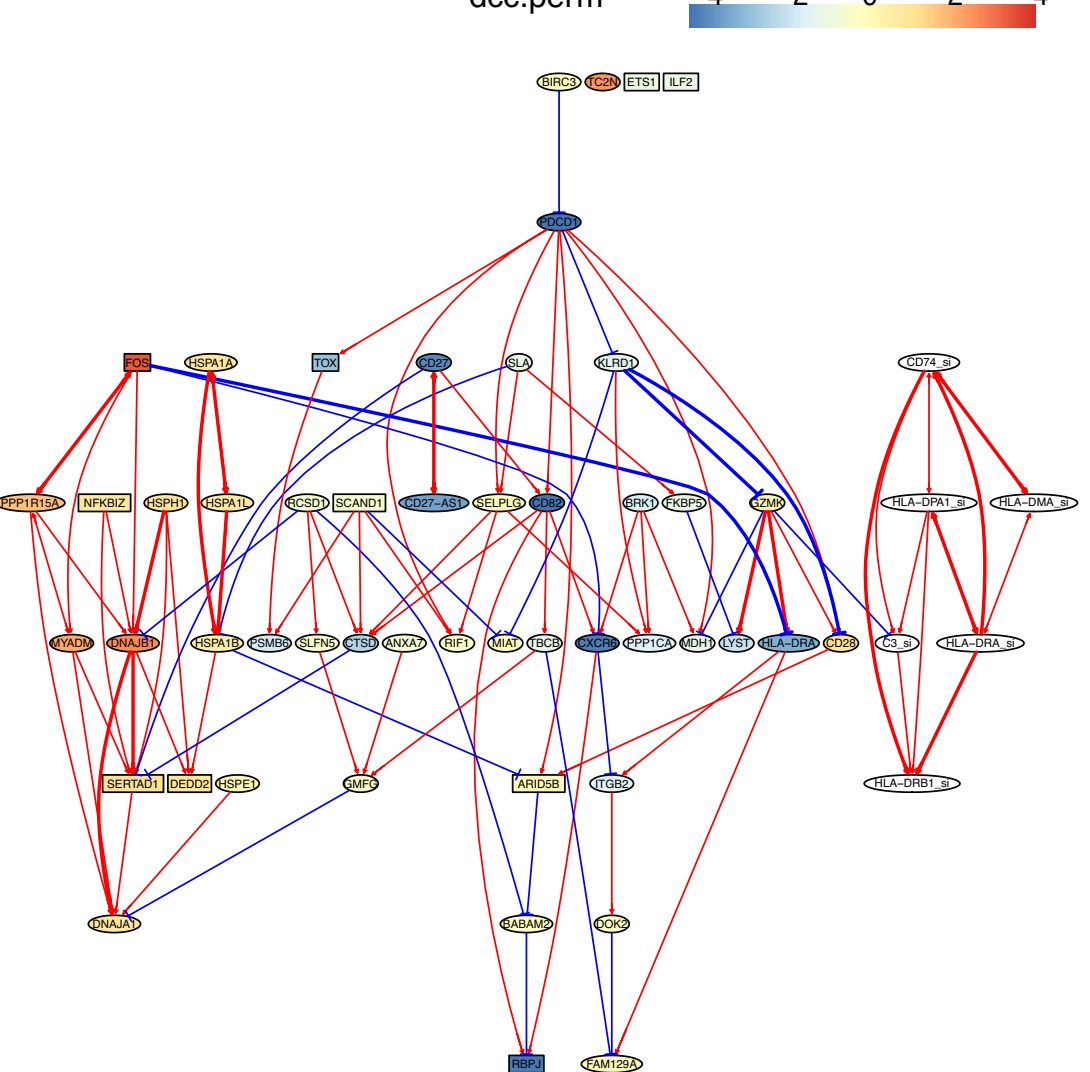

**Appendix 4—figure 2.** Multiple algorithms can identify genes and their relationships in the spike-in dataset from genes and their relationships in the primary dataset. The six genes and their relationships were identified in the network of RCIT, but the network has multiple orphan nodes. GZMK-|C3_si is a wrong interaction in the networks of RCIT and DCC.perm.

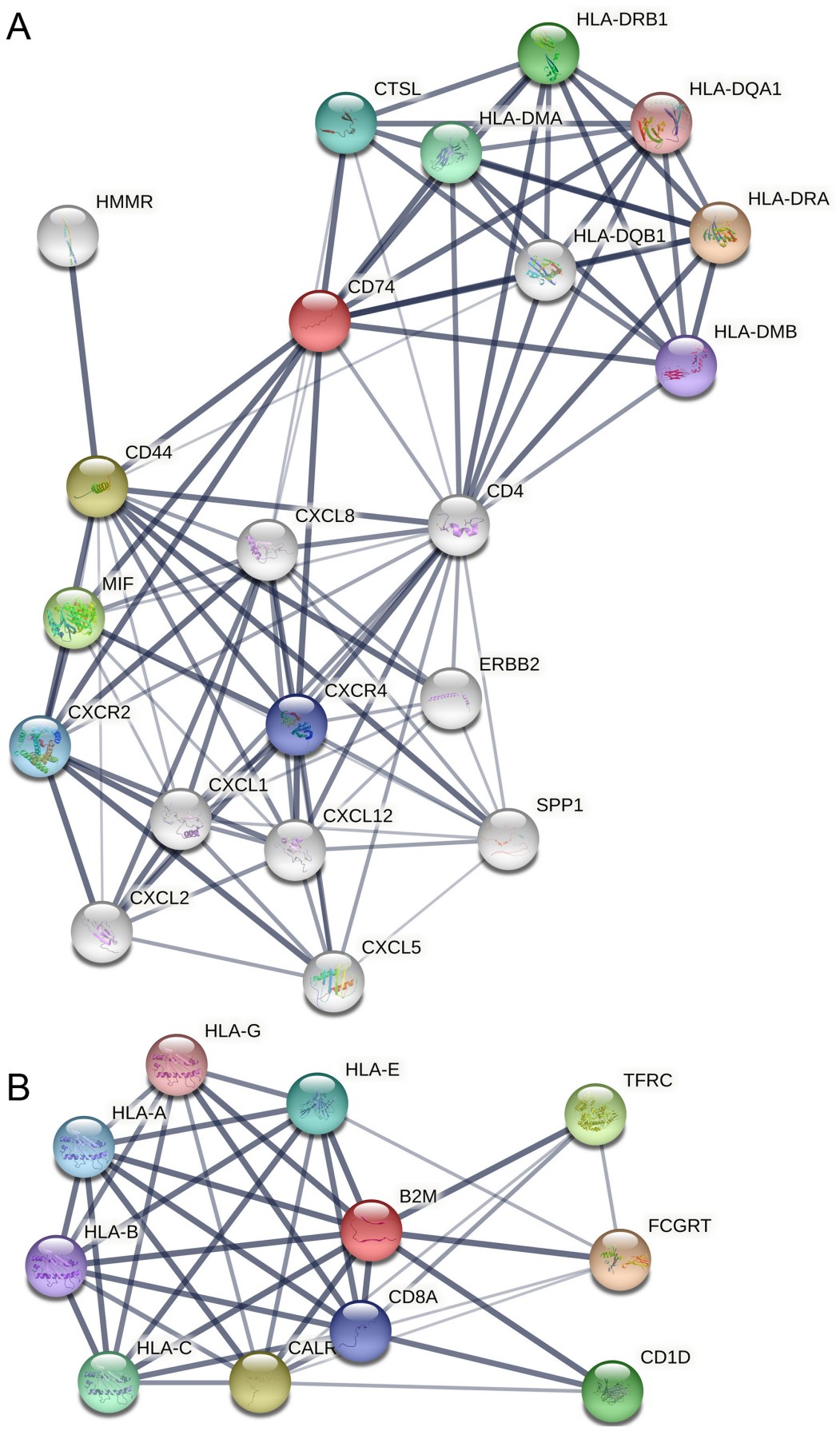

**Appendix 4—figure 3.** Protein interactions in the STRING database (parameter settings: *network type* = full STRING network, *meaning of network edges* = confidence, *active interaction sources* = all, *minimum required interaction score* = medium confidence, 0.4). (**A**) Interactions among MHC-II genes and CD74 and among CXCL and CXCR genes. (**B**) Interactions among MHC-I genes and B2M.

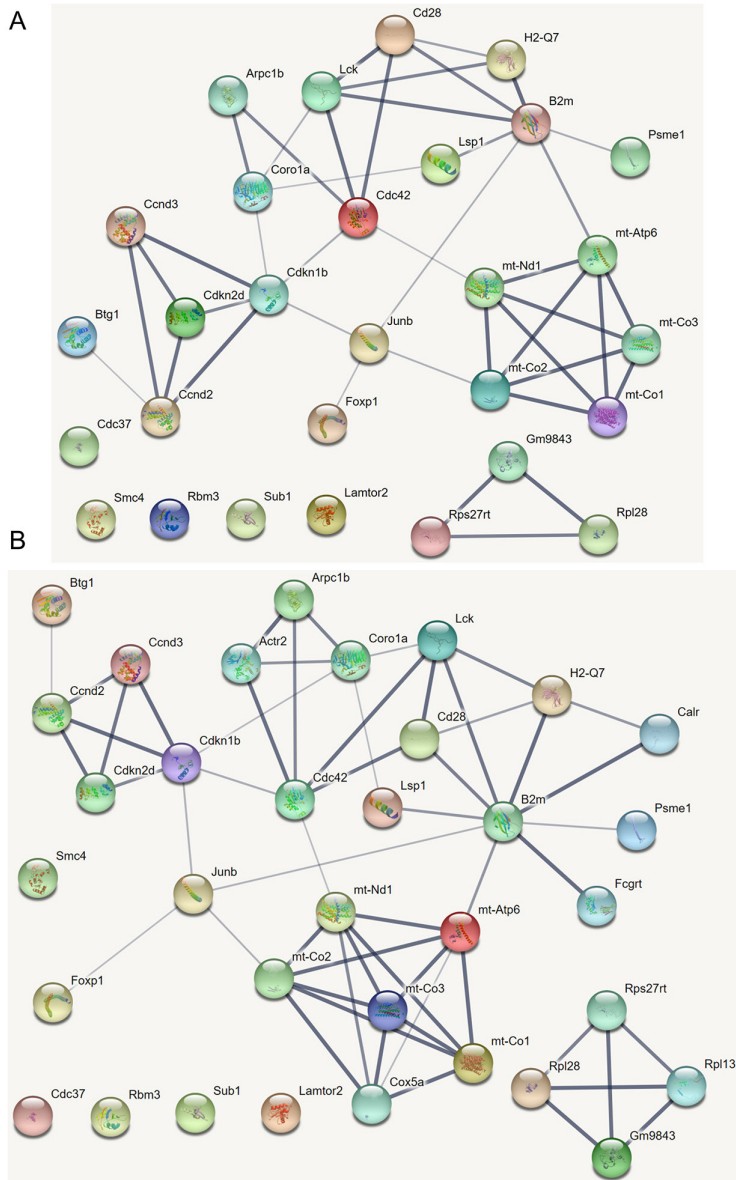

**Appendix 4—figure 4.** Interactions among the differentially expressed genes in CD4 T cells in the STRING database (https://string-db.org/) (parameter settings: network type = full STRING network, meaning of network edges = confidence, active interaction sources = all, minimum required interaction score = medium confidence, 0.4). (**A**) The interactions. (**B**) The extended interactions ('add more nodes to current network' is chosen).

We have taken a systematic approach to validate causal discovery using the five lung cancer cell lines and lung alveolar cells. First, upon (a) gene expression value >0.1, (b) gene expression >50% cells, (c) fold change >0.3, we identified differentially expressed genes in each cell line against the alveolar cells. Second, we applied GO analysis to the differentially expressed genes in each cancer dataset using g:Profiler (https://biit.cs.ut.ee/gprofiler/gost) (parameters: Significance threshold = Benjamini-Hochberg FDR, User threshold = 0.05, Data sources = KEGG and WikiPathways). The WikiPathways and KEGG pathways 'Metabolic reprogramming in colon cancer' (WP4290), 'Pyrimidine metabolism' (WP4022), and 'Nucleotide metabolism' (hsa01232) are commonly enriched in all cancer cell lines

(*Appendix 4—figure 5*). We also performed GO analysis using the GSEA package, which identified the KEGG pathway 'Non-small cell lung cancer' (hsa05223) as an enriched pathway in cancer cell lines (note that these lung cancer cell lines were derived from NSCLC). We used the PC+DCC. gamma to infer interactions among genes in the three pathways in the five cancer cell lines and the alveolar cells.

| Source | Term_name | Term_id | Adjusted_p_value | Term_size | Intersection_size |
|---|---|---|---|---|---|
| KEGG | Ribosome | KEGG:03010 | 2.25E-34 | 127 | 72 |
| KEGG | Parkinson disease | KEGG:05012 | 4.45E-33 | 238 | 99 |
| KEGG | Huntington disease | KEGG:05016 | 5.62E-29 | 272 | 101 |
| KEGG | Prion disease | KEGG:05020 | 2.25E-26 | 243 | 91 |
| KEGG | Amyotrophic lateral sclerosis | KEGG:05014 | 6.00E-25 | 327 | 106 |
| KEGG | Alzheimer disease | KEGG:05010 | 5.01E-22 | 353 | 106 |
| KEGG | Oxidative phosphorylation | KEGG:00190 | 5.01E-22 | 118 | 56 |
| KEGG | Chemical carcinogenesis - reactive oxygen species | KEGG:05208 | 1.54E-19 | 206 | 73 |
| KEGG | Pathways of neurodegeneration - multiple diseases | KEGG:05022 | 1.76E-19 | 442 | 117 |
| KEGG | Diabetic cardiomyopathy | KEGG:05415 | 5.83E-19 | 187 | 68 |
| KEGG | Thermogenesis | KEGG:04714 | 5.02E-15 | 207 | 66 |
| KEGG | Proteasome | KEGG:03050 | 7.21E-14 | 39 | 25 |
| KEGG | Non-alcoholic fatty liver disease | KEGG:04932 | 5.62E-12 | 142 | 48 |
| KEGG | Spliceosome | KEGG:03040 | 2.14E-09 | 123 | 40 |
| KEGG | Coronavirus disease - COVID-19 | KEGG:05171 | 4.04E-08 | 207 | 53 |
| KEGG | Metabolic pathways | KEGG:01100 | 4.27E-06 | 1457 | 217 |
| KEGG | Carbon metabolism | KEGG:01200 | 1.42E-05 | 112 | 31 |
| KEGG | Spinocerebellar ataxia | KEGG:05017 | 7.13E-05 | 132 | 33 |
| **KEGG** | **Nucleotide metabolism** | **KEGG:01232** | **9.90E-05** | **83** | **24** |
| KEGG | Protein processing in endoplasmic reticulum | KEGG:04141 | 3.19E-04 | 154 | 35 |
| KEGG | Citrate cycle (TCA cycle) | KEGG:00020 | 4.02E-04 | 29 | 12 |
| KEGG | Cardiac muscle contraction | KEGG:04260 | 8.53E-04 | 77 | 21 |
| KEGG | Glutathione metabolism | KEGG:00480 | 9.14E-04 | 56 | 17 |
| KEGG | Biosynthesis of unsaturated fatty acids | KEGG:01040 | 9.14E-04 | 27 | 11 |
| KEGG | Protein export | KEGG:03060 | 1.63E-03 | 20 | 9 |
| KEGG | Glycolysis / Gluconeogenesis | KEGG:00010 | 4.90E-03 | 64 | 17 |
| KEGG | Base excision repair | KEGG:03410 | 4.90E-03 | 32 | 11 |
| KEGG | Retrograde endocannabinoid signaling | KEGG:04723 | 7.56E-03 | 135 | 28 |
| KEGG | DNA replication | KEGG:03030 | 1.06E-02 | 35 | 11 |
| KEGG | Pyruvate metabolism | KEGG:00620 | 1.11E-02 | 46 | 13 |
| KEGG | Fatty acid elongation | KEGG:00062 | 1.81E-02 | 27 | 9 |
| KEGG | Salmonella infection | KEGG:05132 | 1.95E-02 | 236 | 41 |
| KEGG | Pyrimidine metabolism | KEGG:00240 | 2.39E-02 | 56 | 14 |
| KEGG | Cell cycle | KEGG:04110 | 2.47E-02 | 120 | 24 |
| KEGG | Folate biosynthesis | KEGG:00790 | 2.88E-02 | 24 | 8 |
| KEGG | Shigellosis | KEGG:05131 | 3.80E-02 | 239 | 40 |
| KEGG | Bacterial invasion of epithelial cells | KEGG:05100 | 3.80E-02 | 72 | 16 |
| KEGG | p53 signaling pathway | KEGG:04115 | 3.80E-02 | 72 | 16 |
| KEGG | Pathogenic Escherichia coli infection | KEGG:05130 | 4.36E-02 | 183 | 32 |
| KEGG | Cysteine and methionine metabolism | KEGG:00270 | 4.72E-02 | 49 | 12 |
| WP | Cytoplasmic ribosomal proteins | WP:WP477 | 1.60E-25 | 69 | 47 |
| WP | Electron transport chain: OXPHOS system in mitochondria | WP:WP111 | 5.24E-24 | 92 | 53 |
| WP | Oxidative phosphorylation | WP:WP623 | 2.28E-11 | 55 | 29 |
| **WP** | **Metabolic reprogramming in colon cancer** | **WP:WP4290** | **2.23E-10** | **42** | **24** |
| WP | Nonalcoholic fatty liver disease | WP:WP4396 | 2.25E-10 | 146 | 49 |
| WP | Mitochondrial complex I assembly model OXPHOS system | WP:WP4324 | 2.69E-09 | 50 | 25 |
| WP | Proteasome degradation | WP:WP183 | 3.15E-08 | 55 | 25 |
| WP | mRNA processing | WP:WP411 | 1.01E-05 | 114 | 34 |
| WP | VEGFA-VEGFR2 signaling pathway | WP:WP3888 | 2.34E-05 | 415 | 83 |
| WP | Alzheimer's disease | WP:WP5124 | 2.34E-05 | 250 | 57 |
| WP | Mitochondrial complex IV assembly | WP:WP4922 | 6.41E-05 | 29 | 14 |
| WP | Cori cycle | WP:WP1946 | 2.06E-04 | 17 | 10 |
| WP | Parkin-ubiquitin proteasomal system pathway | WP:WP2359 | 3.76E-04 | 60 | 20 |
| WP | Aerobic glycolysis | WP:WP4629 | 4.24E-04 | 12 | 8 |
| WP | Retinoblastoma gene in cancer | WP:WP2446 | 4.24E-04 | 86 | 25 |
| WP | Pathogenic Escherichia coli infection | WP:WP2272 | 4.24E-04 | 47 | 17 |
| WP | Glycolysis and gluconeogenesis | WP:WP534 | 2.88E-03 | 44 | 15 |
| WP | Pentose phosphate metabolism | WP:WP134 | 3.89E-03 | 6 | 5 |
| WP | TCA cycle (aka Krebs or citric acid cycle) | WP:WP78 | 9.88E-03 | 17 | 8 |
| WP | Base excision repair | WP:WP4752 | 1.04E-02 | 30 | 11 |
| WP | Alzheimer's disease and miRNA effects | WP:WP2059 | 1.04E-02 | 319 | 58 |
| WP | nsp1 from SARS-CoV-2 inhibits translation initiation in the h| WP:WP5027 | 1.28E-02 | 14 | 7 |
| **WP** | **Pyrimidine metabolism** | **WP:WP4022** | **1.28E-02** | **78** | **20** |
| WP | Eukaryotic transcription initiation | WP:WP405 | 1.32E-02 | 36 | 12 |

**Appendix 4—figure 5.** The enriched KEGG and WikiPathways pathways of differentially expressed genes in the A549 cell line. KEGG:01232, WP4290, and WP4022 are enriched in all of the five lung cancer cell lines.

First, we examined the 'Metabolic reprogramming in colon cancer' (WP4290) pathway (*Appendix 4—figures 6–9*). Numerous studies report that glucose metabolism is reprogrammed and nucleotides synthesis is increased in cancer cells. Thus, we first examined and compared the WP4290 pathway in the five lung cancer cell lines and lung alveolar cells. The key features of the reprogrammed glucose metabolism are that (a) glucose intake is increased, (b) the glycolysis/TCA cycle intermediates are used for synthesizing nucleotide, (c) lactate generation is increased. The inferred networks capture these features. (a) multiple activations of SLC2A1 (which encodes a major glucose transporter and controls glucose intake), PGD (which promotes glucose metabolism into the pentose phosphate shunt), PSAT1 (which encodes a phosphoserine aminotransferase that catalyzes the reversible conversion of 3-phosphohydroxypyruvate to phosphoserine), and LDHA (whose protein catalyzes the conversion of pyruvate to lactate) are inferred in all cancer cell lines but not in alveolar cells. (b) Many activations of genes by downstream genes are inferred, and this sort of feedback regulations is an intrinsic feature of metabolism. Especially, the controlling factor SLC2A1 is activated by multiple genes. (c) In contrast, none of these features occur in the alveolar cells (partly due to key genes such as SLC2A1 is not expressed). These inferred results are literature-supported and biologically reasonable, despite that the causal inference is flawed by the absence of metabolites in the data.

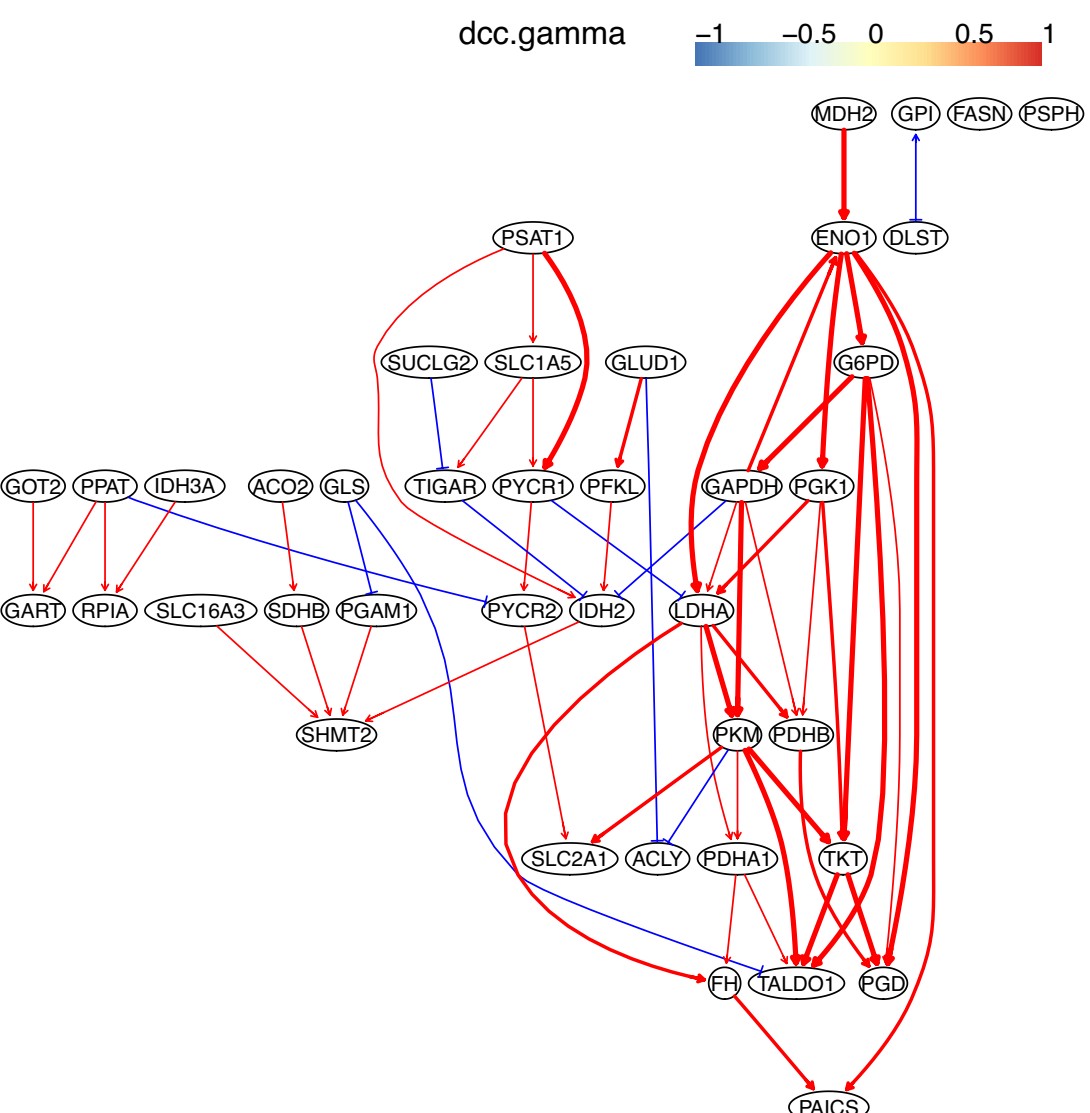

**Appendix 4—figure 6.** The causal relationships between genes in the WP4290 pathway in the cell line A549 inferred using PC+DCC.gamma. The inference is flawed because the true network contains both gene

*Appendix 4—figure 6 continued on next page*

*Appendix 4—figure 6 continued*

products and metabolites but single-cell RNA-sequencing (scRNA-seq) data do not contain metabolites. Nevertheless, *Appendix 4—figures 6–8* show that multiple inferred interactions reasonably reveal key features of reprogrammed glucose metabolism. First, shared interactions in ≥2 datasets are 21.12%, 58.82%, 50.51%, 40.82%, 53.09%, and 50.0% in alveolar cells, H838 cells, H2228 cells, HCC827 cells, H1975 cells, and A549 cells, indicating that causal inference differentiates glucose metabolism in cancer cells from in alveolar cells. Second, the inferred networks reflect key features of reprogrammed glucose metabolism, especially the activation of SLC2A1 (which encodes a major glucose transporter and controls glucose intake), PGD (which promotes glucose metabolism toward nucleotide synthesis), PSAT1 (which promotes glucose metabolism toward nucleotide synthesis), and LDHA (which promotes glucose metabolism toward lactate generation) in cancer cell lines.

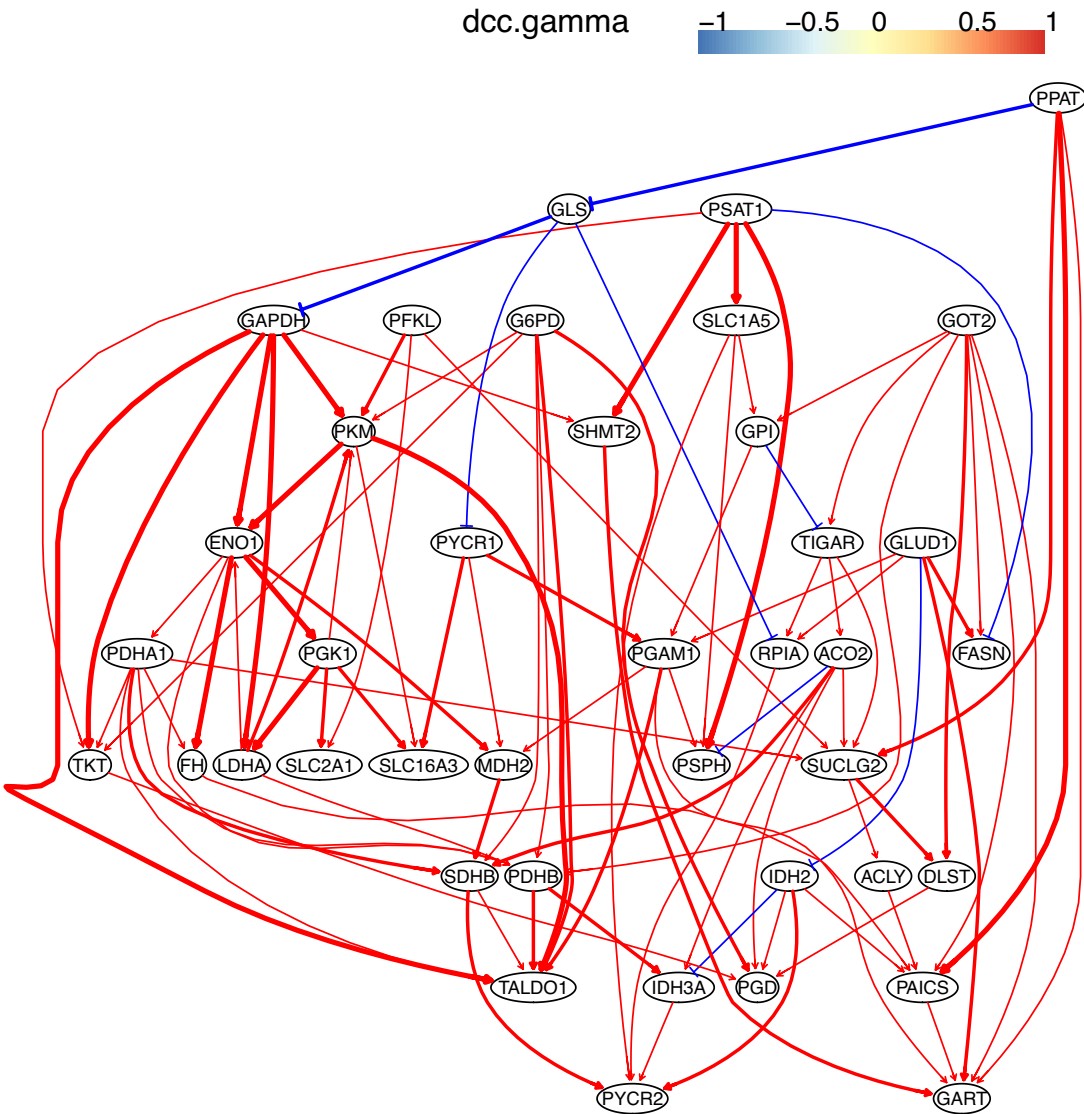

**Appendix 4—figure 7.** The causal relationships between genes in the WP4290 pathway in the cell line H2228 inferred using PC+DCC.gamma.

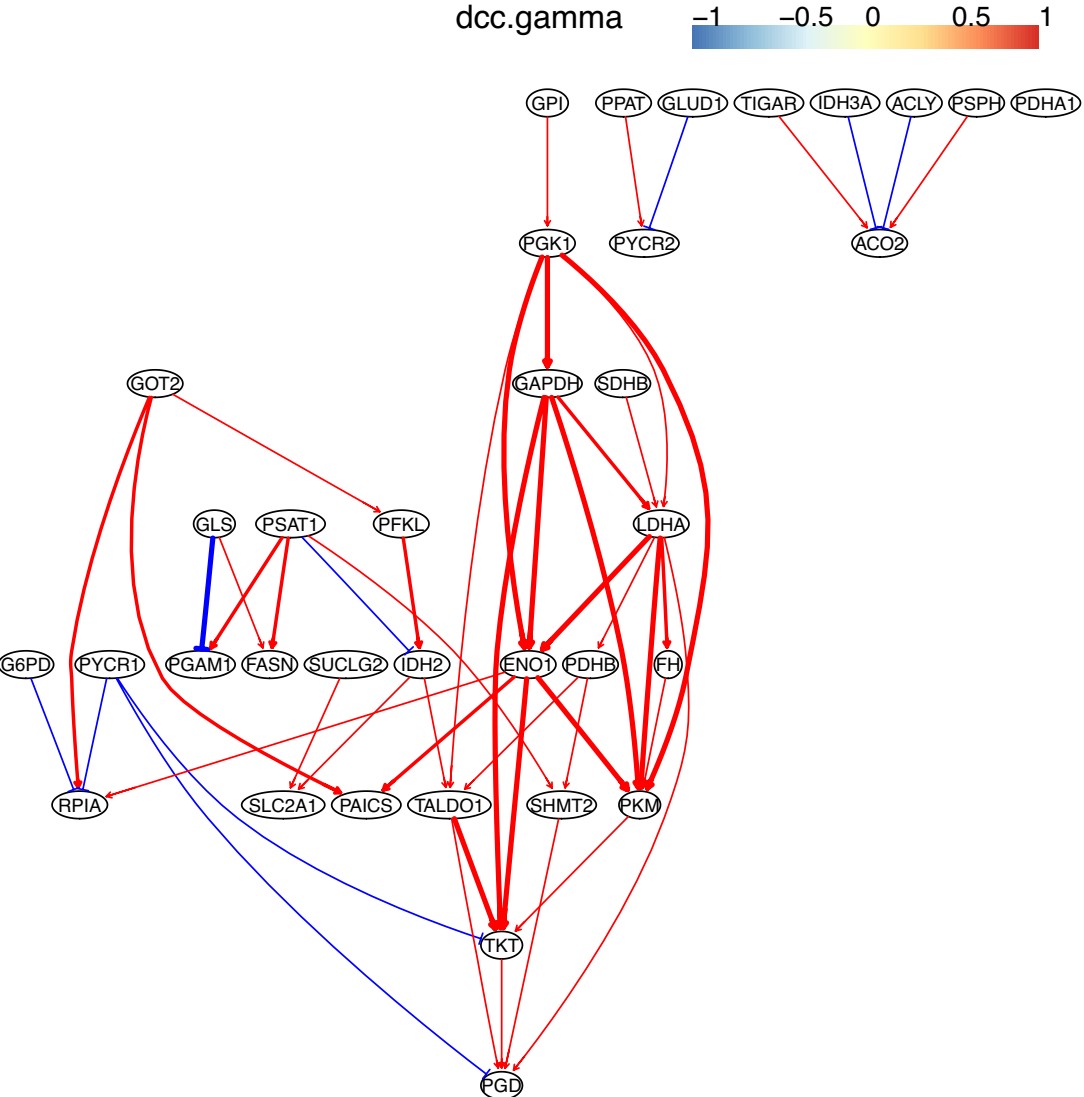

**Appendix 4—figure 8.** The causal relationships between genes in the WP4290 pathway in the cell line H838 inferred using PC+DCC.gamma.

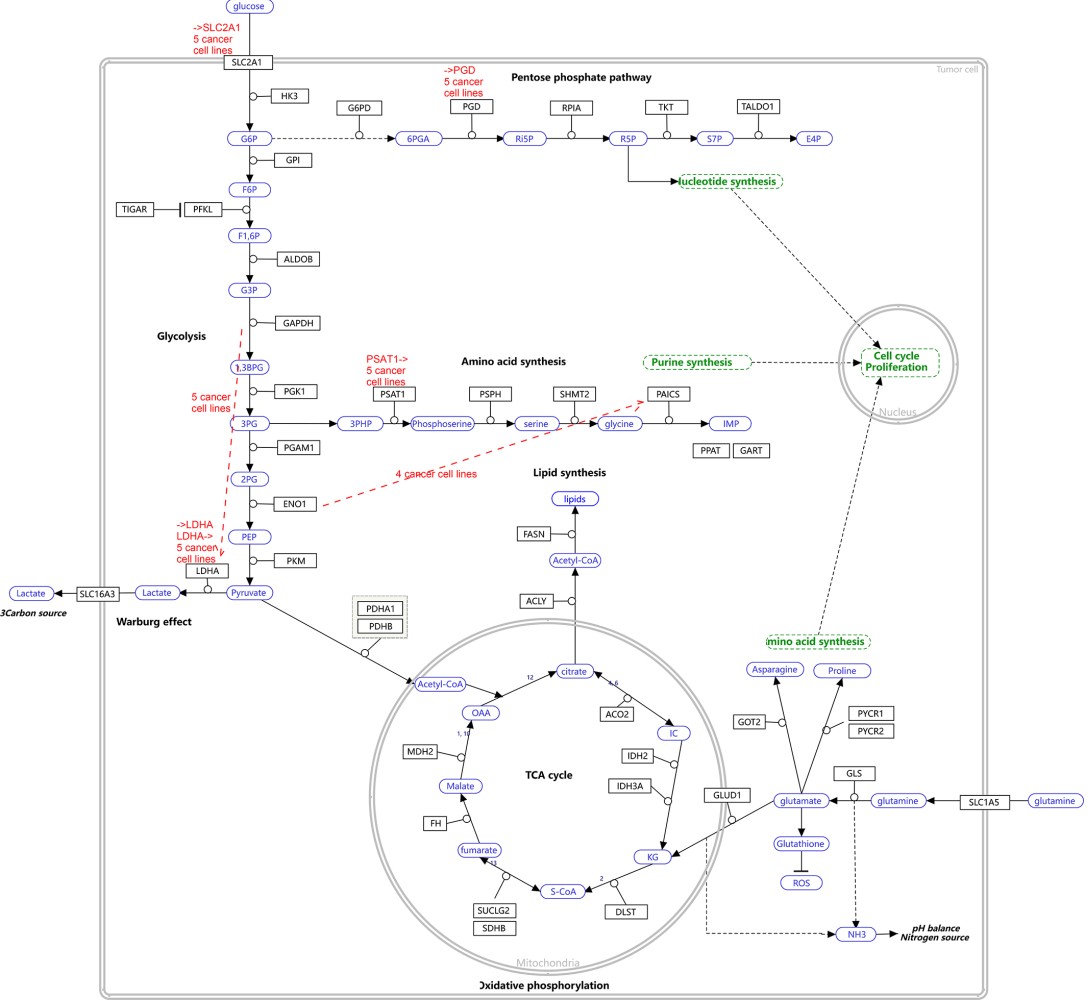

**Appendix 4—figure 9.** The WP4290 pathway and the key features of inferred interactions in lung cancer cell lines. Glucose intake is greatly increased in cancer cells. The increased glucose consumption is used as a carbon source for anabolic processes and this excess carbon is also used for the de novo generation of nucleotides, lipids, and proteins. This so-called Warburg effect is proposed to be an adaptation mechanism to support these biosynthesis processes for uncontrolled proliferation of cancer cells. SLC2A1, PGD, PSAT1, and LDHA are critical genes controlling glucose intake and the generation of nucleotides and lactate. The added red notes indicate key inferred interactions. '→XXX' and 'XXX→' indicate the activation of the gene XXX by others and the activation of others by the gene XXX, respectively. In the alveolar cells, SLC2A1, PGD, PSAT1, ENO1, and LDHA are not expressed and none of these interactions are inferred.

Second, we examined the 'Pyrimidine metabolism' (WP4022) pathway (**Appendix 4—figures 10–13**). We used genes in the 'Pyrimidine metabolism' (hsa00240) to perform the inference (because WP4022 contains too many POLR gene families) and used the more readable WP4022 pathway to illustrate the results. Compared with glucose metabolism, pyrimidine metabolism has many reversable reactions, making interactions vary greatly in cells and the differences between cancer and alveolar cells opaque. The following genes and reactions are notable. (a) TYMS catalyzes dUMP->dTMP unidirectionally toward DNA synthesis. (b) Tk1/2 catalyze thymidine->dTMP and deoxyuridine->dUMP toward DNA synthesis (while NT5C/E/M do the opposite). (c) DUT catalyzes dUTP->dUMP (and dUMP is the substrate for TYMS). (d) TYMP catalyzes thymidine->thymine unidirectionally away from DNA synthesis. (e) ENTPD1/3 catalyze dTTP->dTDP->dTMP, UTP->UDP->UMP, and CTP->CDP->CMP away from DNA and RNA synthesis (but AK9/NME reverse these reactions). (f) NT5C/E/M catalyze dCMP->deoxycytidine, dUMP->deoxyuridine, and dTMP->thymidine away from DNA synthesis. Accordingly, the following interactions were inferred from cancer cell lines. (a) TYMS (the most critical gene promoting DNA synthesis) is activated in all cancer cell lines but not in alveolar

cells, and it is not repressed by any gene in cancer cell lines. (b) Tk1/2 are activated in cancer cells and alveolar cells. (c) DUT is activated in all cancer cell lines but is not expressed in alveolar cells. (d) activations of TYMP (the critical gene making reactions away from DNA synthesis) by multiple others are inferred in alveolar cells. (e) ENTPD1/3 (genes making reactions away from DNA synthesis) are activated only in alveolar cells. (f) NT5C/E/M are repressed in all cancer cell lines but are not expressed in alveolar cells. The most notable may be DUT->Tk1 and DUT->TYMS in all cancer cell lines, indicating feedforward or coordinated regulations that promote DNA synthesis. These features are literature-supported and biologically reasonable, despite that the causal inference is flawed by the absence of metabolites in the data.

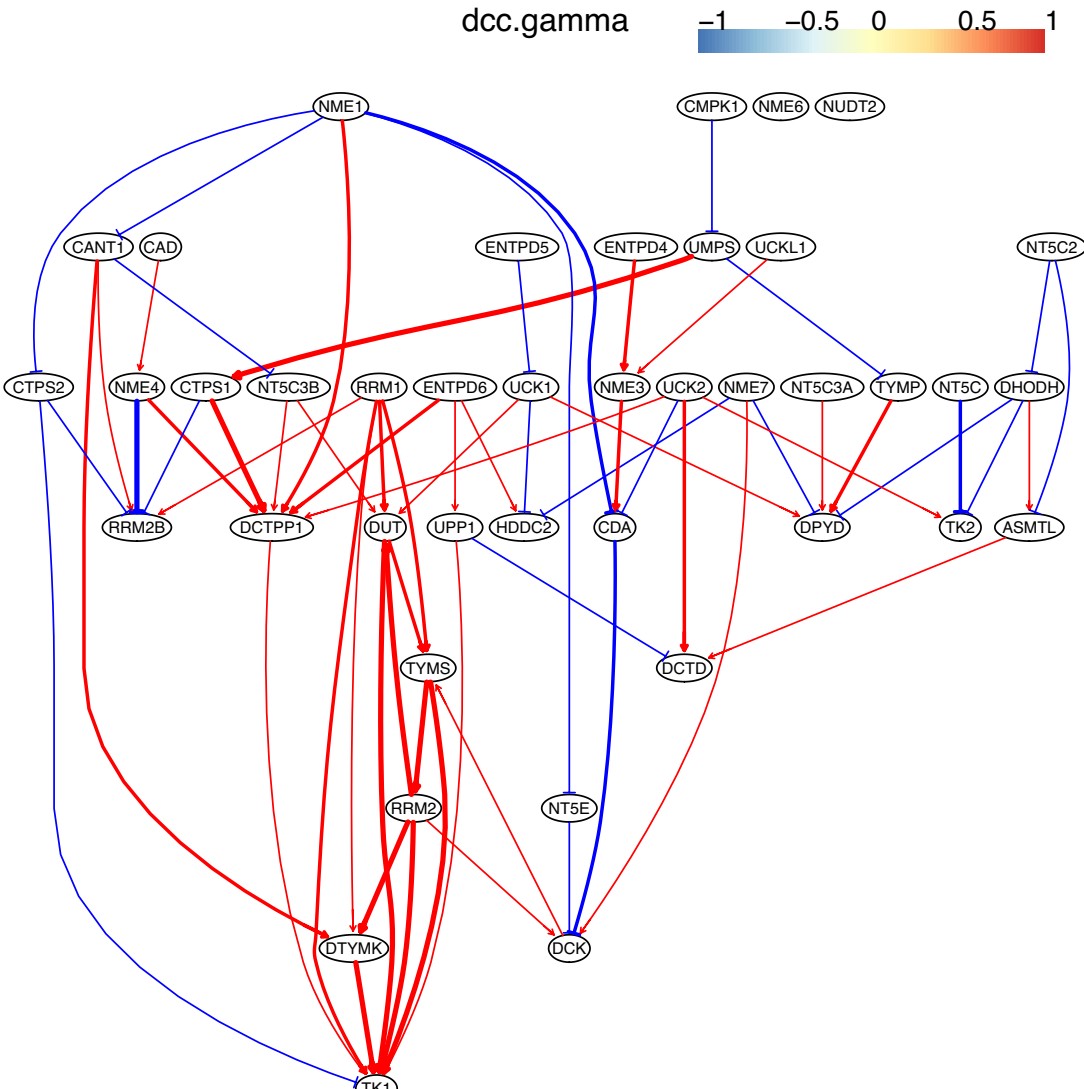

**Appendix 4—figure 10.** The causal interactions between genes in the hsa00240 pathway in the A549 cells. Since pyrimidine metabolism consists of many reversible reactions (*Appendix 4—figure 13*), inferred interactions are more varied than those of glucose metabolism. *Appendix 4—figures 10–12* show that causal inference reasonably reveals the critical differences between cancer cells and alveolar cells, which include that percentages of interactions shared by ≥2 cell lines are 19.74%, 41.38%, 42.42%, 30.86%, 36.67%, and 32.99% in alveolar, H838, H2228, HCC827, H1975, and A549 cells. The regulations of important genes are notable (*Appendix 4—figure 13*). (1) TYMS is activated in five cancer cell lines but not in alveolar cells, and is not repressed in cancer cell lines. (2) Tk1/2 are activated in five cancer cells and alveolar cells. (3) DUT is not expressed in alveolar cells and is activated in the five cancer cell lines. (4) Multiple TYMP activations are inferred in alveolar cells. (5) ENTPD1/3 are activated only in alveolar. (6) NT5C/E/M are repressed in five cancer cell lines but are not expressed in alveolar cells. (7) There are many cases where downstream enzymes activate upstream enzymes, such as ENTPD3->CTPS2. Of note, DUT->Tk1 and DUT->TYMS in all five cancer cell lines indicate well-coordinated causal interactions for DNA synthesis in cancer cells.

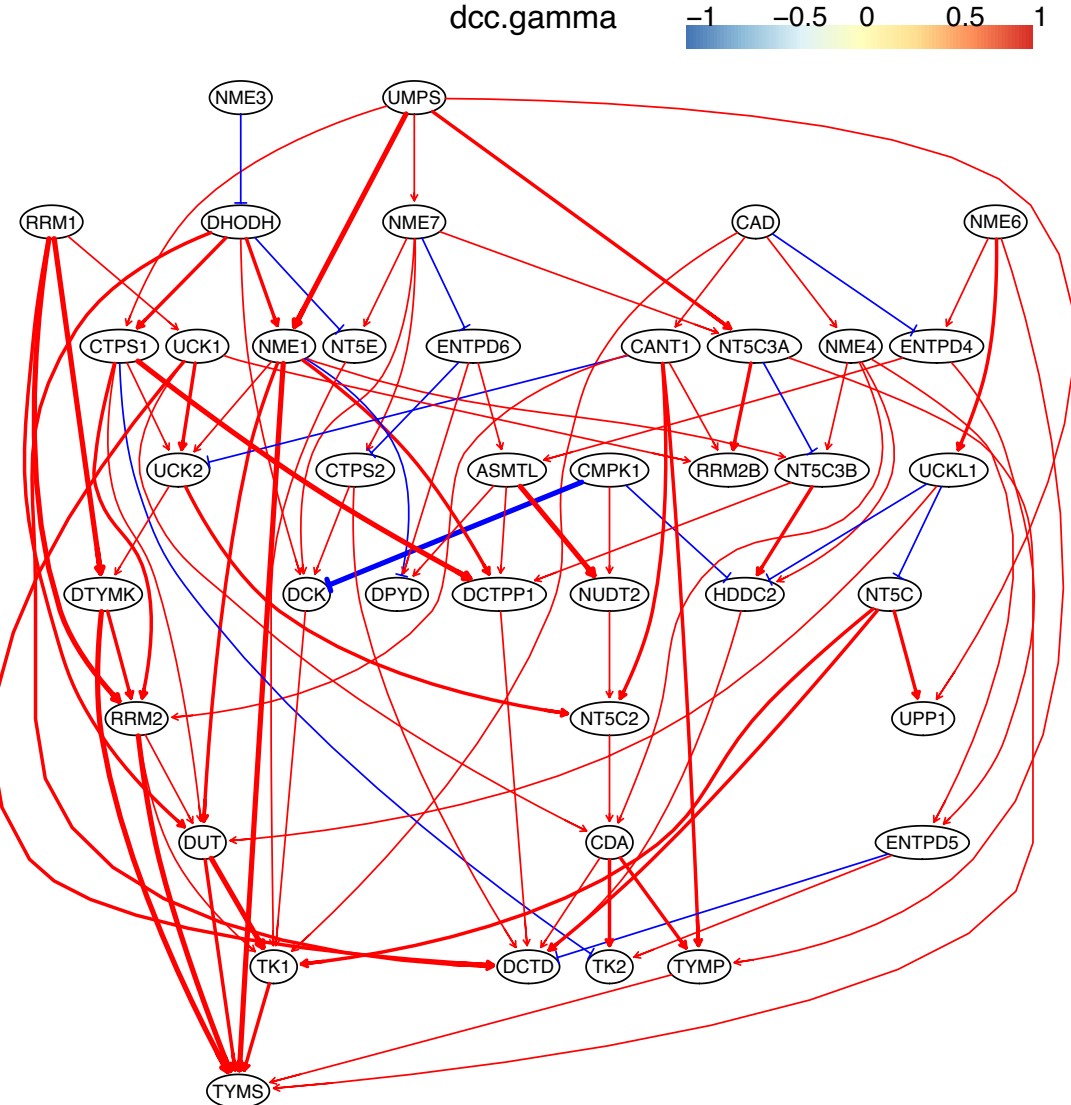

**Appendix 4—figure 11.** The causal interactions between genes in the hsa00240 pathway in the HCC827 cells.

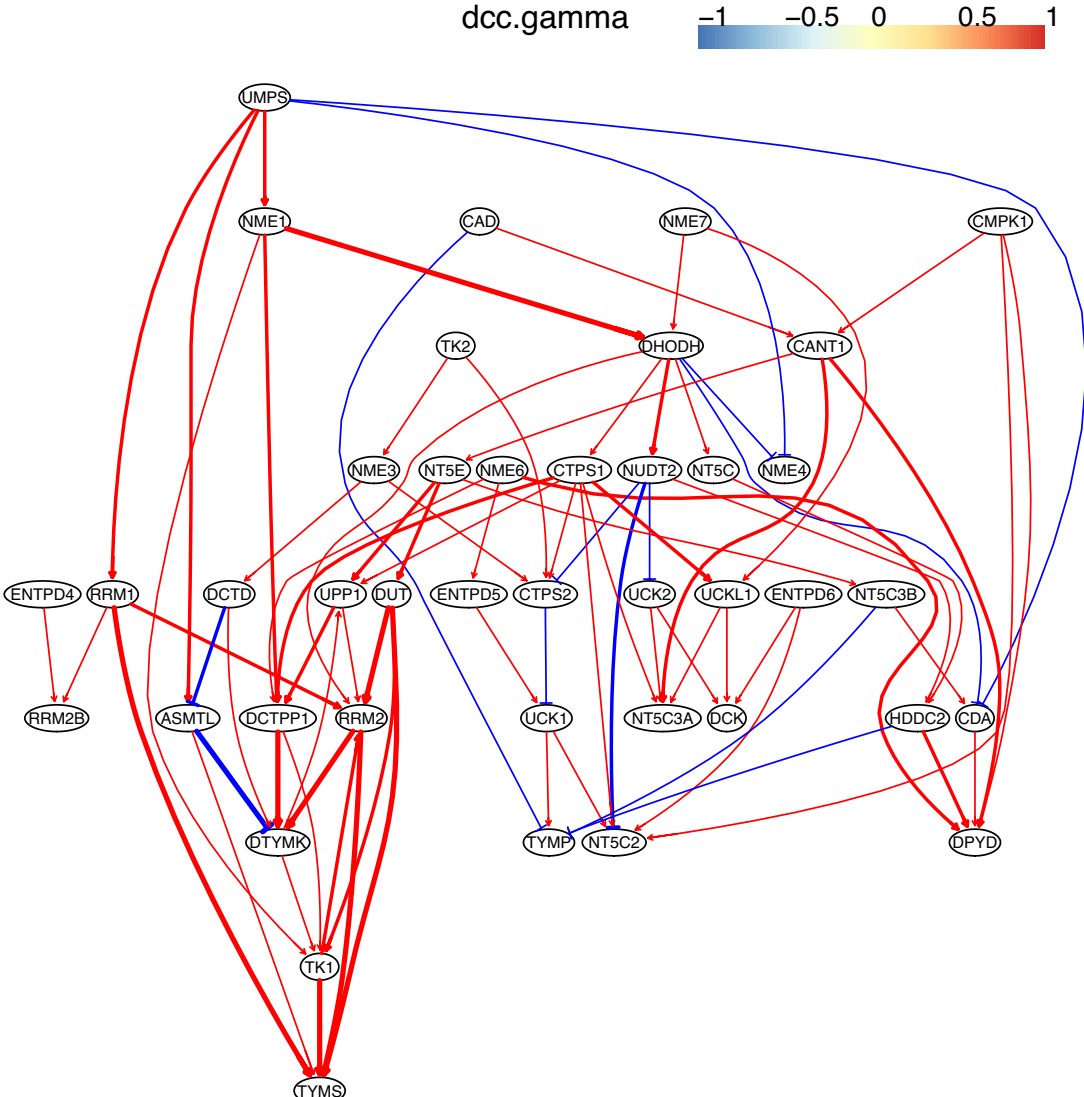

**Appendix 4—figure 12.** The causal interactions between genes in the hsa00240 pathway in the H1975 cells.

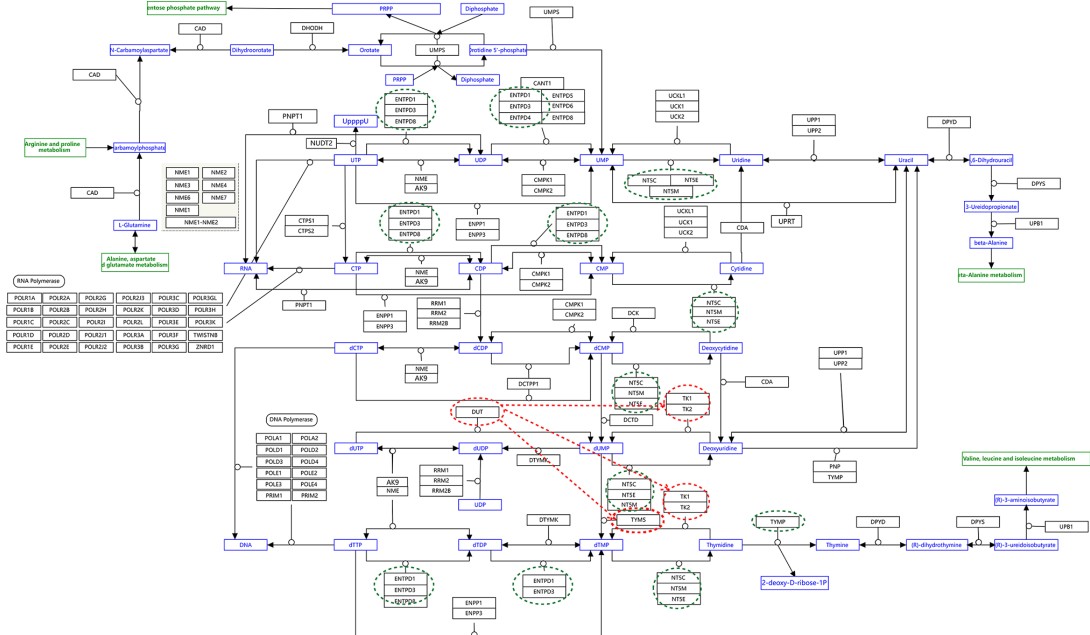

**Appendix 4—figure 13.** The pyrimidine metabolism pathway and key features of inferred causal networks in lung cancer cell lines and alveolar cells. Genes in the KEGG 'Pyrimidine metabolism' (hsa00240) pathway were used to perform causal inference (because WP4022 contains too many POLR gene families) and the figure of WP4022 was used to illustrate the results (this figure is more readable). This figure indicates that pyrimidine metabolism has many reversible reactions, and these reactions somewhat blur the key features in cancer cell lines and alveolar cells. The following genes and reactions are notable. (1) TYMS turns dUMP->dTMP unidirectionally toward DNA synthesis. (2) Tk1/2 turn thymidine->dTMP and deoxyuridine->dUMP toward DNA synthesis (while NT5C/E/M do the opposite). (3) DUT turns dUTP->dUMP, and dUMP is the substrate for TYMS. (4) TYMP turns thymidine->thymine unidirectionally away from DNA synthesis. (5) ENTPD1/3 turn dTTP->dTDP->dTMP, UTP->UDP->UMP, and CTP->CDP->CMP away from DNA synthesis and RNA synthesis (but these reactions can be reversed by AK9/NME). (6) NT5C/E/M turn dCMP->deoxycytidine, dUMP->deoxyuridine, and dTMP->thymidine away from DNA synthesis. Red and green ellipses mark genes that promote DNA synthesis and genes that do not promote DNA synthesis. In the inferred causal networks, accordingly, there are following interactions. (1) TYMS is activated in five cancer cell lines but not in alveolar cells, and is not repressed in cancer cell lines. (2) Tk1/2 are activated in five cancer cells and alveolar cells. (3) DUT is not expressed in alveolar cells and is activated in the five cancer cell lines. (4) Multiple TYMP activations are inferred in alveolar cells. (5) ENTPD1/3 are activated only in alveolar. (6) NT5C/E/M are repressed in five cancer cell lines but are not expressed in alveolar cells. (7) There are many cases where downstream enzymes activate upstream enzymes, such as ENTPD3->CTPS2. Of note, there are DUT->Tk1 and DUT->TYMS in all five cancer cell lines, indicating coordinated molecular interaction and gene regulation for DNA synthesis in cancer cells.

Third, we examined the 'Non-small cell lung cancer' (hsa05223) pathway (*Appendix 4—table 1*; *Appendix 4—figure 14*). We used the 'graphite' R package to turn hsa05223 into an adjacency matrix and mapped inferred interactions to the matrix. If an interaction can be mapped to an edge or a path with any directions (forward, inverse, or undirected) in hsa05223, it was assumed mapped to the pathway. hsa05223 contains sub-pathways such as p53 signaling pathway and PI3K-AKT pathway, therefore there are considerable epistatic interactions that are not annotated in hsa05223. Also, synergistic interactions (e.g. CDKN1A->BAX and EGFR->MET, see *Dong et al., 2019*; *Wang et al., 2014*), and many of which are literature-supported but not annotated. We additionally examined hsa05223 and sub-pathways wherein manually and found that many inferred interactions can be mapped to epistatic and synergistic interactions. Taken together, in each cell line, about 50% of inferred interactions can be mapped to the pathway. Note that this is the result without considering feedback regulations by TFs. For example, many EGF1-related interactions were inferred (e.g. E2F1->EGFR and RB1->ERBB2), but these interactions were not accounted because they are not annotated in the KEGG database. Two extra notes here. First, unlike reprogrammed glucose metabolism, common interactions between genes in different cell lines are not impressive, probably because these cell lines are generated with different genetic basis despite being derived

from NSCLC. Second, the annotation of hsa05223 has defects, because it is not in the list of enriched pathways identified by g:Profiler.

**Appendix 4—table 1.** The percentages of mapped edges between inferred networks and the hsa05223 pathway.

| Cell lines | Inferred interactions | Num of 'forward' | Num of 'reverse' | Num of 'undirected' | Num of 'epistatic' and 'synergistic' | All |
|---|---|---|---|---|---|---|
| A549 | 82 | 10 (12.2%) | 12 (14.63%) | 2 (2.44%) | 17 (20.73%) | 50% |
| H838 | 77 | 9 (11.69%) | 10 (12.99%) | 1 (1.3%) | 21 (27.27%) | 53.25% |
| H1975 | 102 | 13 (12.75%) | 16 (15.69%) | 1 (0.98%) | 17 (16.67%) | 46.09% |
| H2228 | 106 | 13 (12.26%) | 20 (18.87%) | 2 (1.89%) | 25 (23.58%) | 56.60% |
| HCC827 | 122 | 13 (10.66%) | 26 (21.31%) | NA (NA%) | 28 (22.95%) | 54.92% |

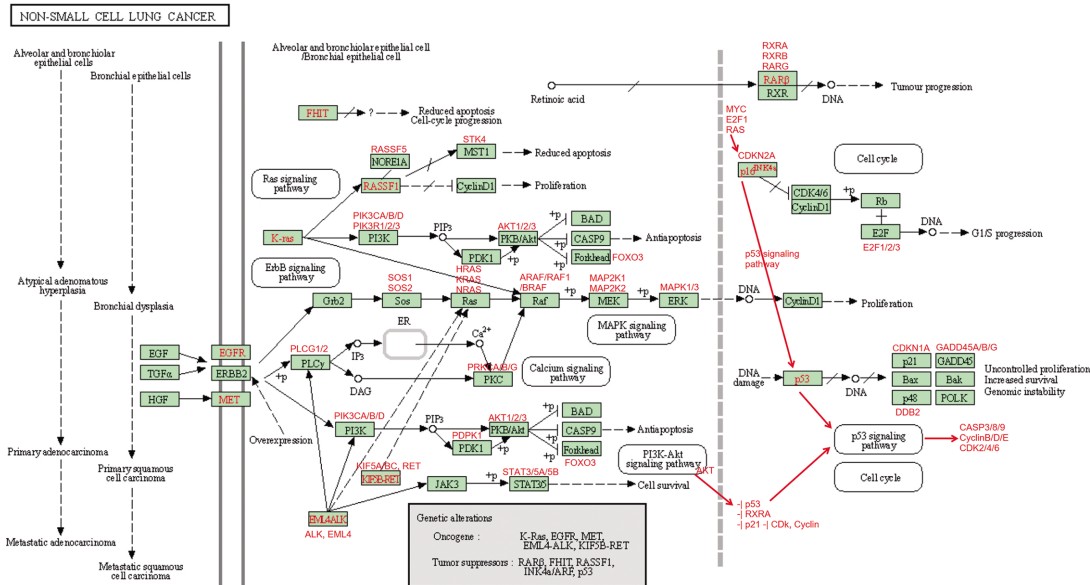

**Appendix 4—figure 14.** The 'Non-small cell lung cancer' (hsa05223) pathway. Annotated sub-pathways, genes, and interactions are marked in red.

## Appendix 5

### Additional results of applications

This appendix file describes the additional results of five applications, including the analysis of lung cancer cell lines and alveolar epithelial cells, the analysis of macrophages isolated from glioblastoma, the analysis of tumor-infiltrating exhausted CD8 T cells, identifying genes and inferring interactions that signify CD4 T cell aging, and the analysis of a flow cytometry dataset. These examples were used to examine the applicability of causal discovery to varied cell types and sequencing protocols. To same running time and also examine algorithms' power, varied sample sizes were used. All of these data were analyzed using the PC+CI method. The results indicate that causal discovery can be applied flexibly to varied cells. The appendix text (including appendix tables and figures) is brief and divided into five subsections, with the first four corresponding to the four subsections in the Results section in the main text, following appendix figures that are ordered accordingly.

### 1. The analysis of lung cancer cell lines and lung alveolar epithelial cells

As expected, feature genes and causal networks in H2228 and lung alveolar epithelial cells are distinctly different (Main text-*Figure 4*; *Appendix 5—figures 1–8*). (a) HLA Class II genes and CD74 are down-regulated in H2228 cells but up-regulated in lung alveolar epithelial cells. (b) LCN2 is up-regulated in H2228 cells but down-regulated in lung alveolar epithelial cells. (c) Algorithms inferred multiple interactions between PRDX1, TALDO1, HSP90AA1, NQO1, and PSMC4 in H2228 cells, but none of them were inferred in lung alveolar epithelial cells. (d) HLA Class I genes are feature genes in H2228 cells but not in the lung alveolar epithelial cells. HLA genes make proteins called human leukocyte antigens (HLA), which take bits and pieces of proteins from inside the cell and display them on the cell's surface. If the cell is cancerous or infected, the HLA proteins display abnormal fragments that trigger immune cells to destroy that cell. Down-regulated HLA genes may help cancer cells escape from immune cells. Annotating the networks upon related experimental findings suggest that DCC algorithms are the best and cmiKnn and GaussCItest are the poorest.

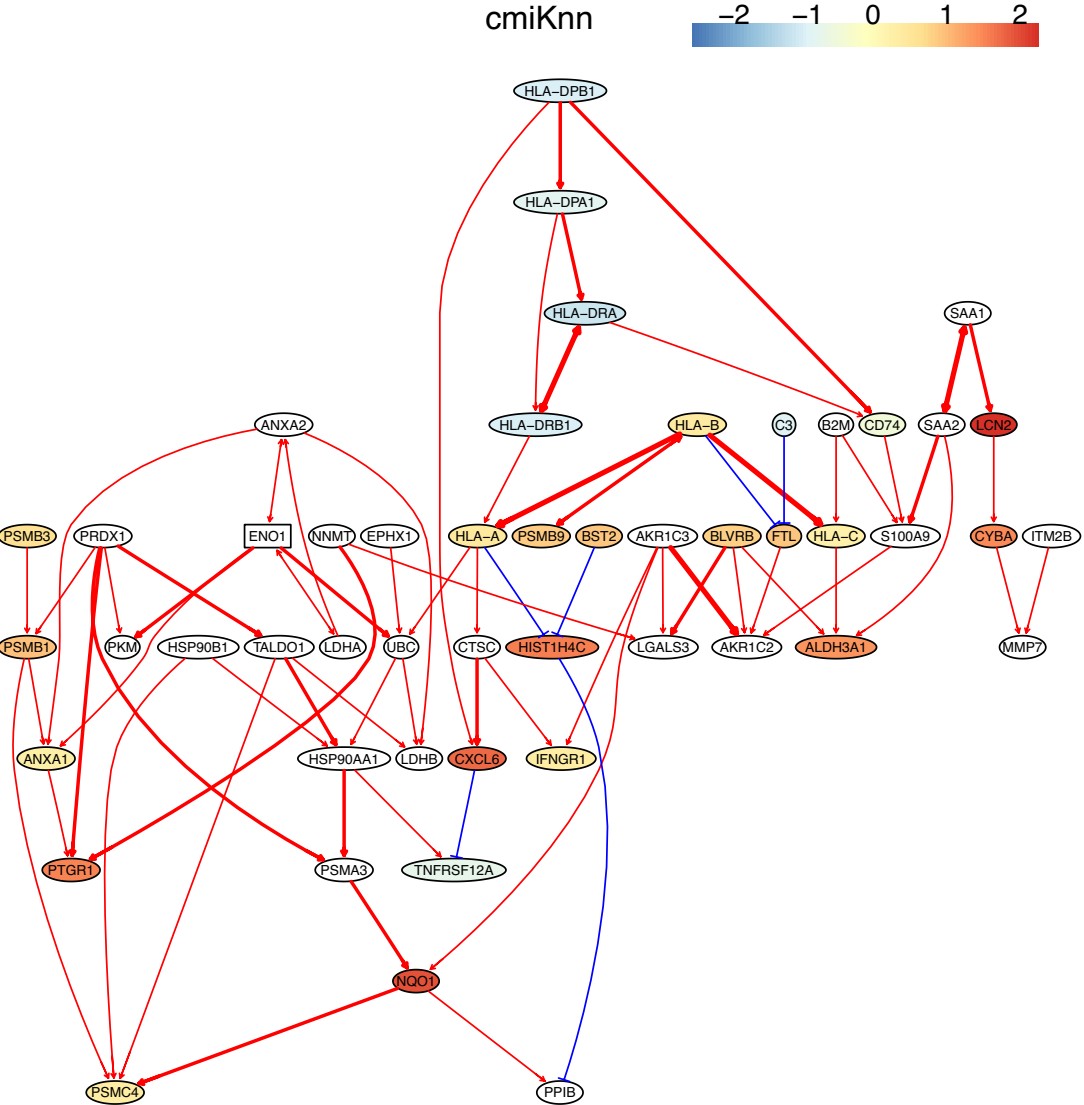

**Appendix 5—figure 1.** The causal network of the 50 feature genes inferred by PC+cmiKnn from the H2228 dataset (settings: feature genes were expressed in >50% cells, the alpha level for CI test was 0.1, and the 300 cells with more feature genes expressed in cells were used). In these and the following figures, red and blue arrows indicate activation and inhibition, double arrows indicate undermined direction, arrows' thickness indicates the statistical significance of CI test, and node colors indicate fold changes of gene expression. MHC-II genes were significantly down-regulated compared with the control (the lung alveolar epithelial cells).

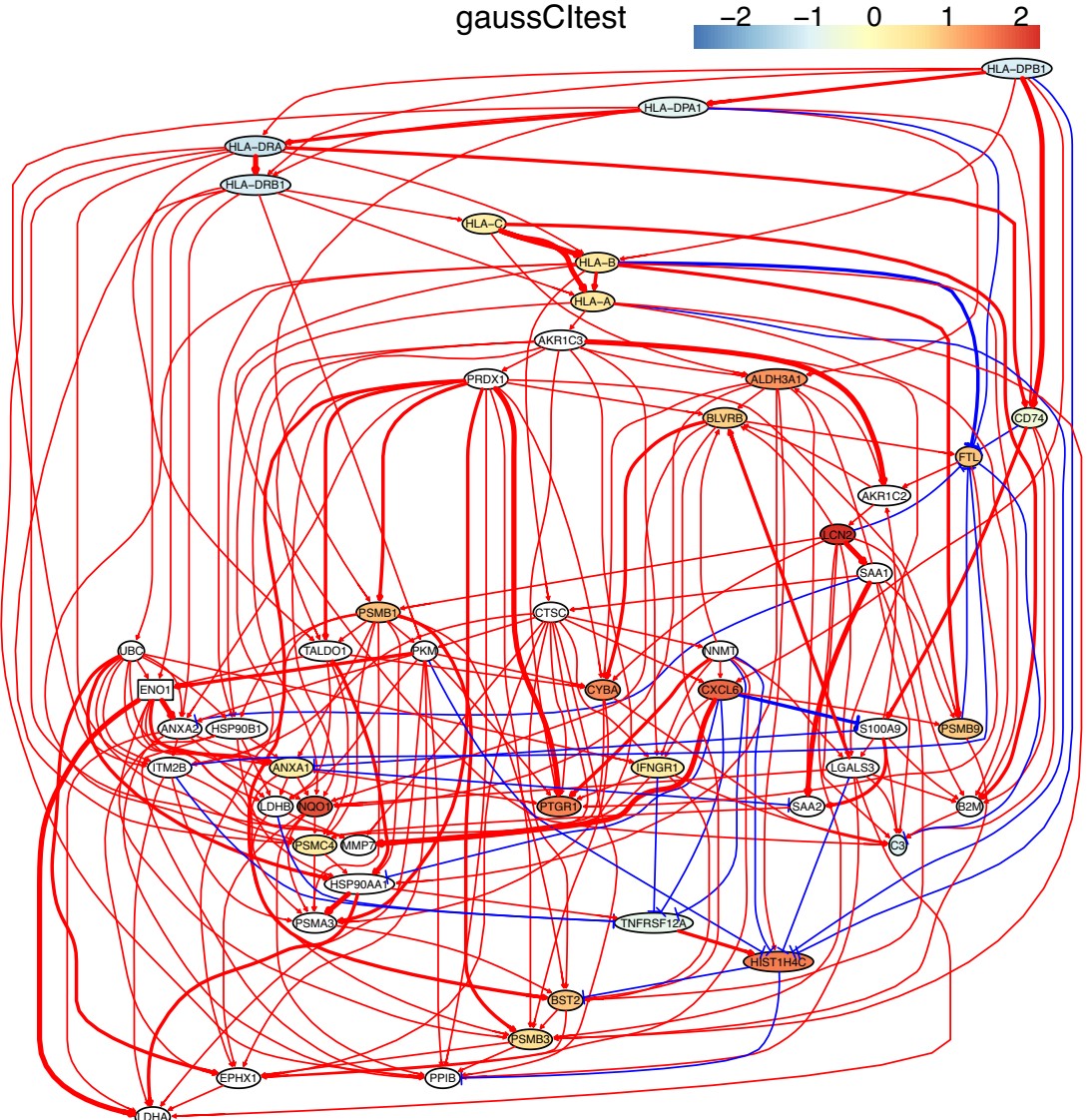

**Appendix 5—figure 2.** The causal network of the 50 feature genes inferred by PC+GaussCItest from the H2228 dataset (settings: feature genes were expressed in >50% cells, the alpha level for CI test was 0.1, and the 300 cells with more feature genes expressed in cells were used). MHC-II genes were significantly down-regulated compared with the control (the lung alveolar epithelial cells).

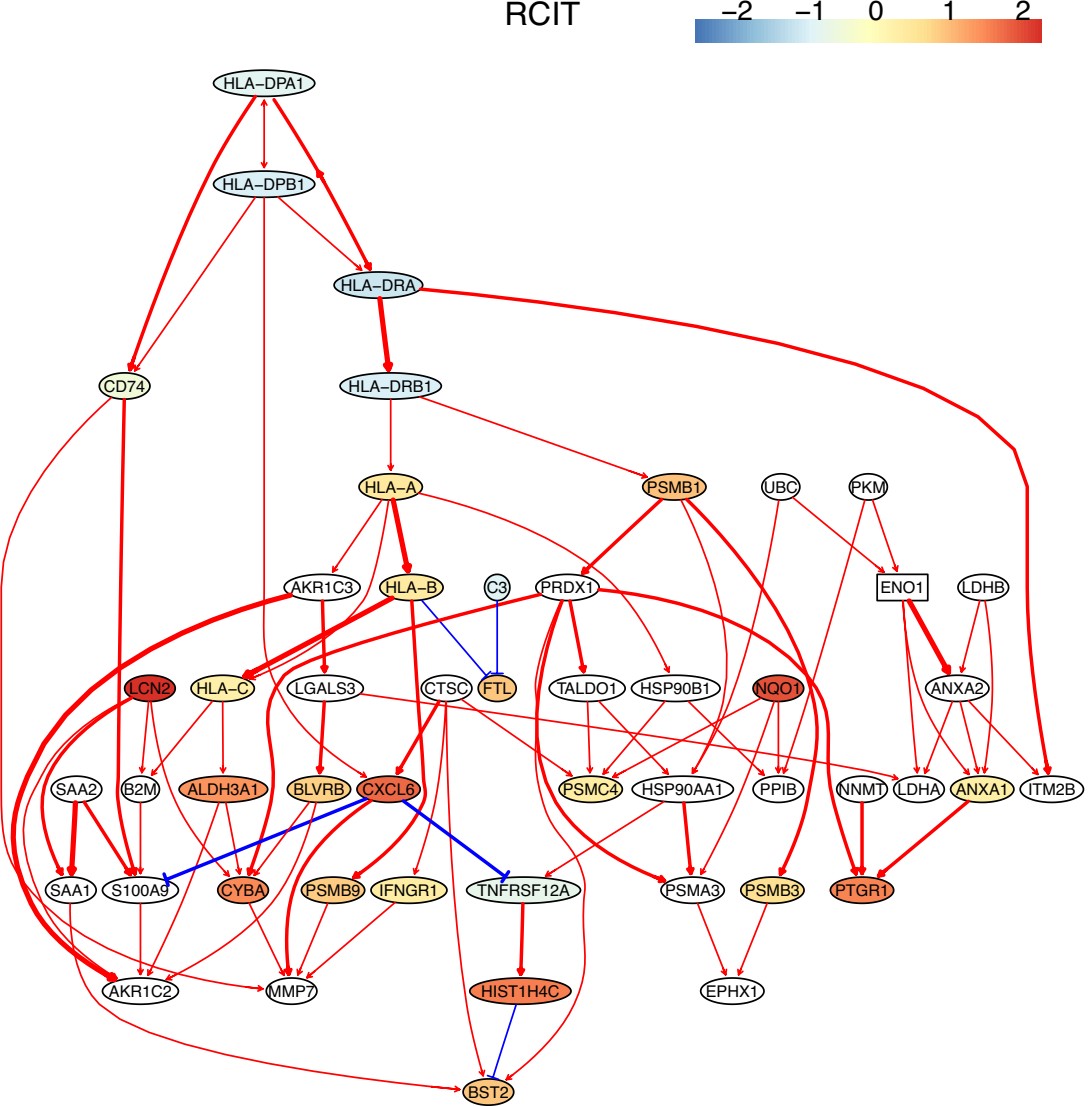

**Appendix 5—figure 3.** The causal network of the 50 feature genes inferred by PC+RCIT from the H2228 dataset (settings: feature genes were expressed in >50% cells, the alpha level for CI test was 0.1, and the 300 cells with more feature genes expressed in cells were used). MHC-II genes were significantly down-regulated compared with the control (the lung alveolar epithelial cells).

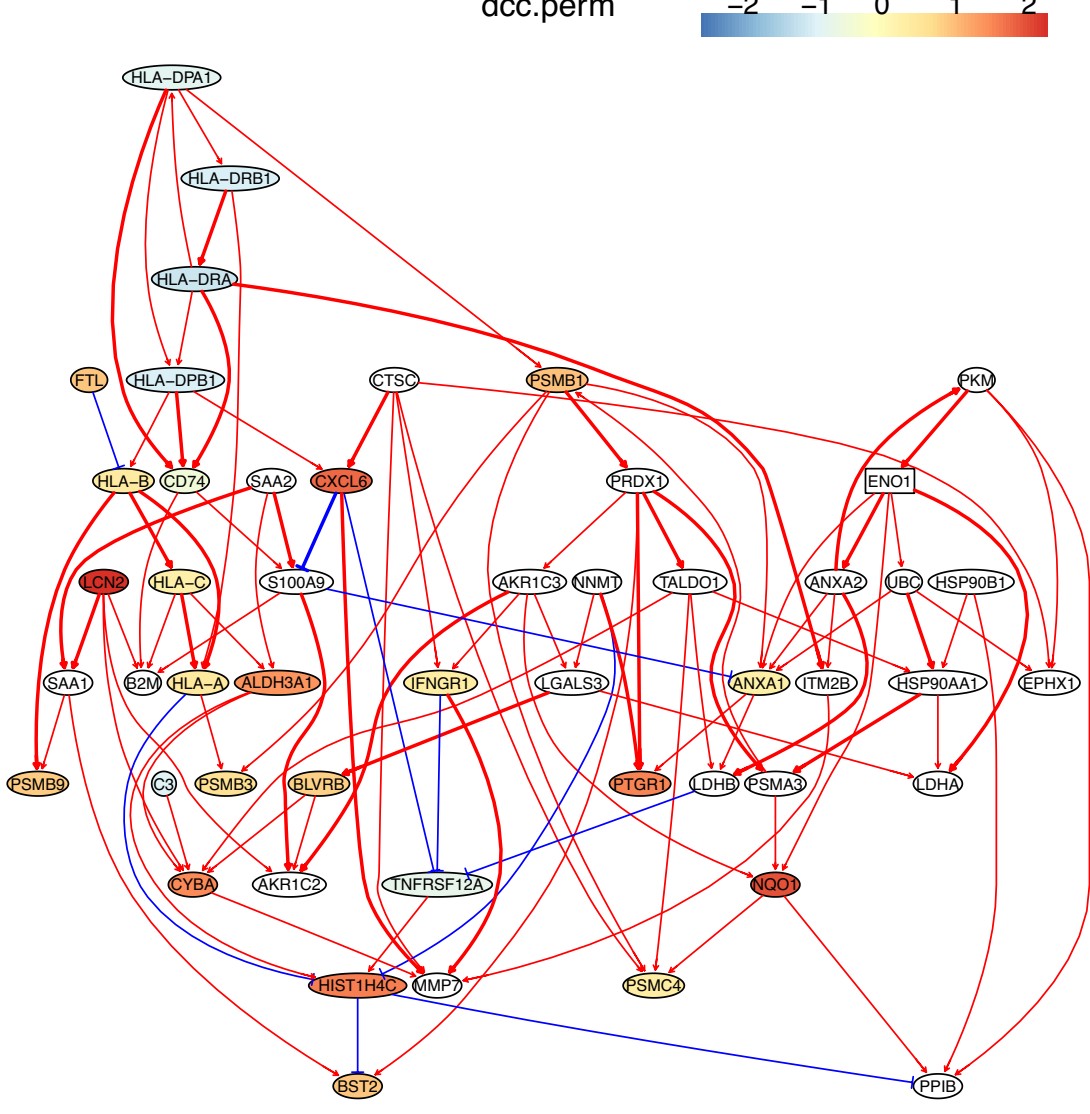

**Appendix 5—figure 4.** The causal network of the 50 feature genes inferred by PC+DCC.perm from the H2228 dataset (settings: feature genes were expressed in >50% cells, the alpha level for CI test was 0.1, and the 300 cells with more feature genes expressed in cells were used). MHC-II genes were significantly down-regulated compared with the control (the lung alveolar epithelial cells).

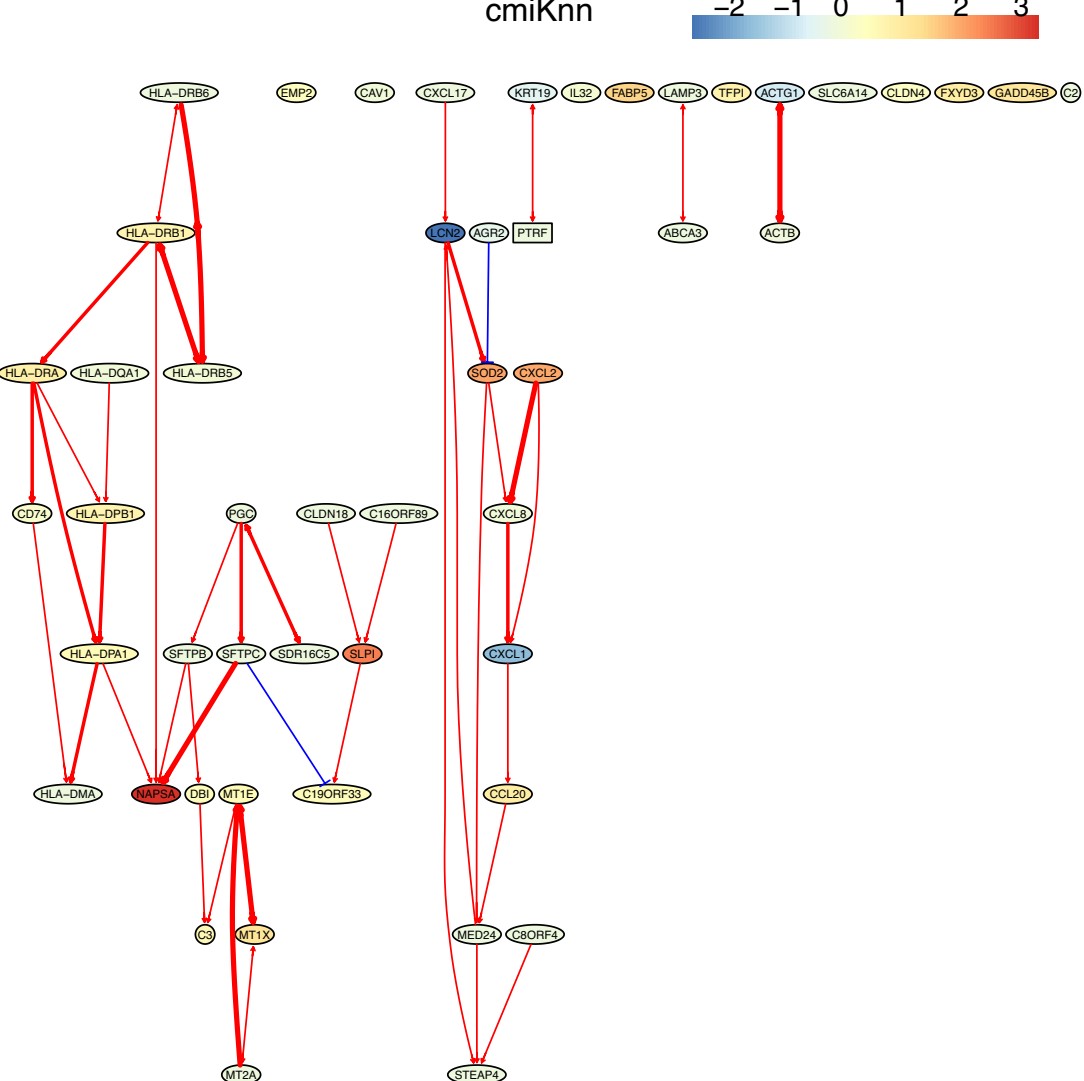

**Appendix 5—figure 5.** The causal network of the 50 feature genes inferred by PC+cmiKnn from the alveolar epithelial cell dataset (settings: feature genes were expressed in >50% cells, the alpha level for CI test was 0.1, and the 300 cells with more feature genes expressed the cells were used). The control of the case was the H2228 cells. Compared with H2228 cells, MHC-II genes in alveolar epithelial cells were highly expressed. The relationships between MHC-II genes and the relationships between MHC-II genes and CD74 (which is a key regulator of MHC-II proteins) are supported by annotated interactions in the STRING database.

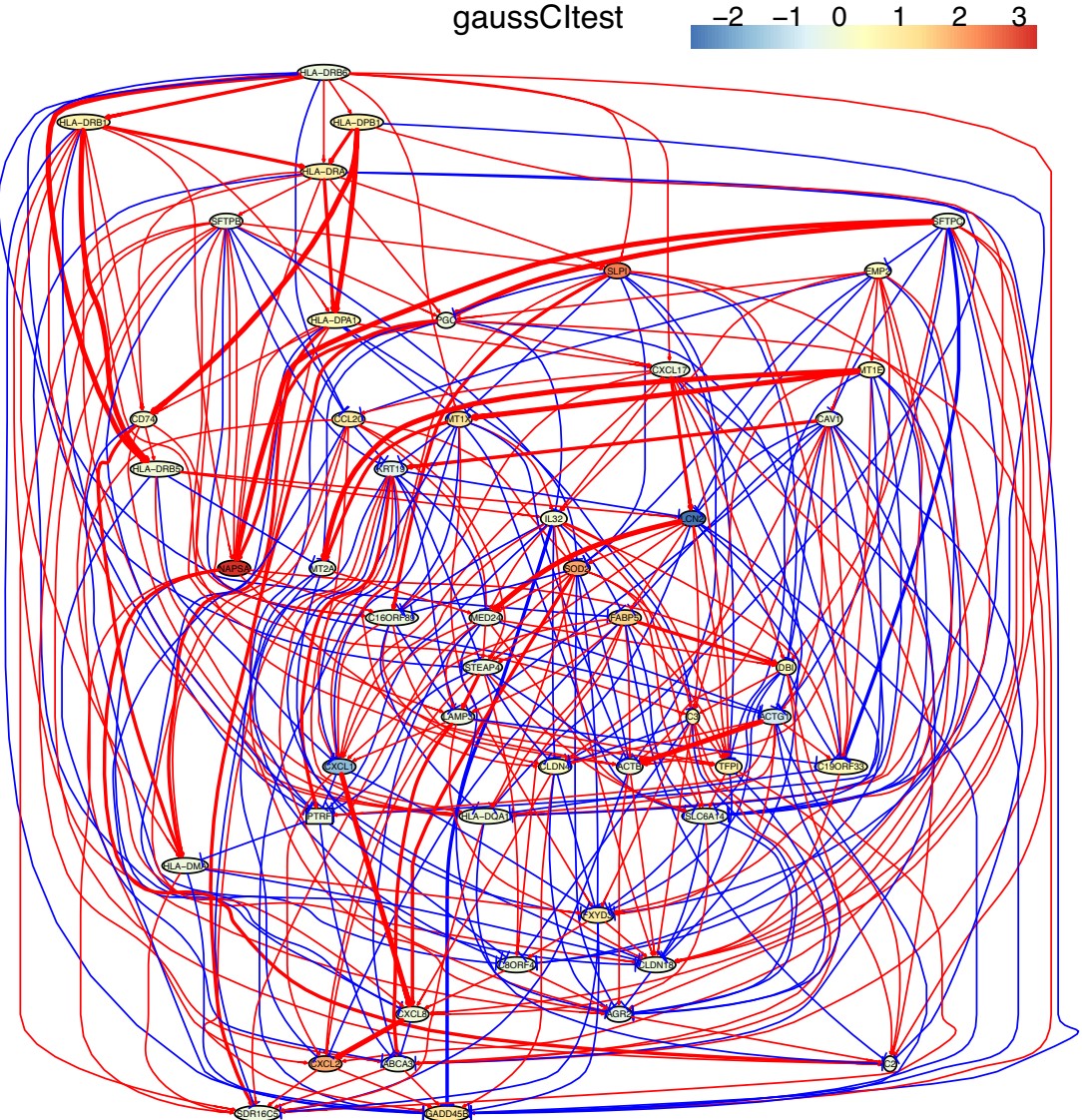

**Appendix 5—figure 6.** The causal network of the 50 feature genes inferred by PC+GaussCItest from the alveolar epithelial cell dataset (settings: feature genes were expressed in >50% cells, the alpha level for CI test was 0.1, and the 300 cells with more feature genes expressed the cells were used). The control of the case was the H2228 cells. Compared with H2228 cells, MHC-II genes in alveolar epithelial cells were highly expressed. The relationships between MHC-II genes and between MHC-II genes and CD74 are much more dense than those inferred by PC+cmiknn.

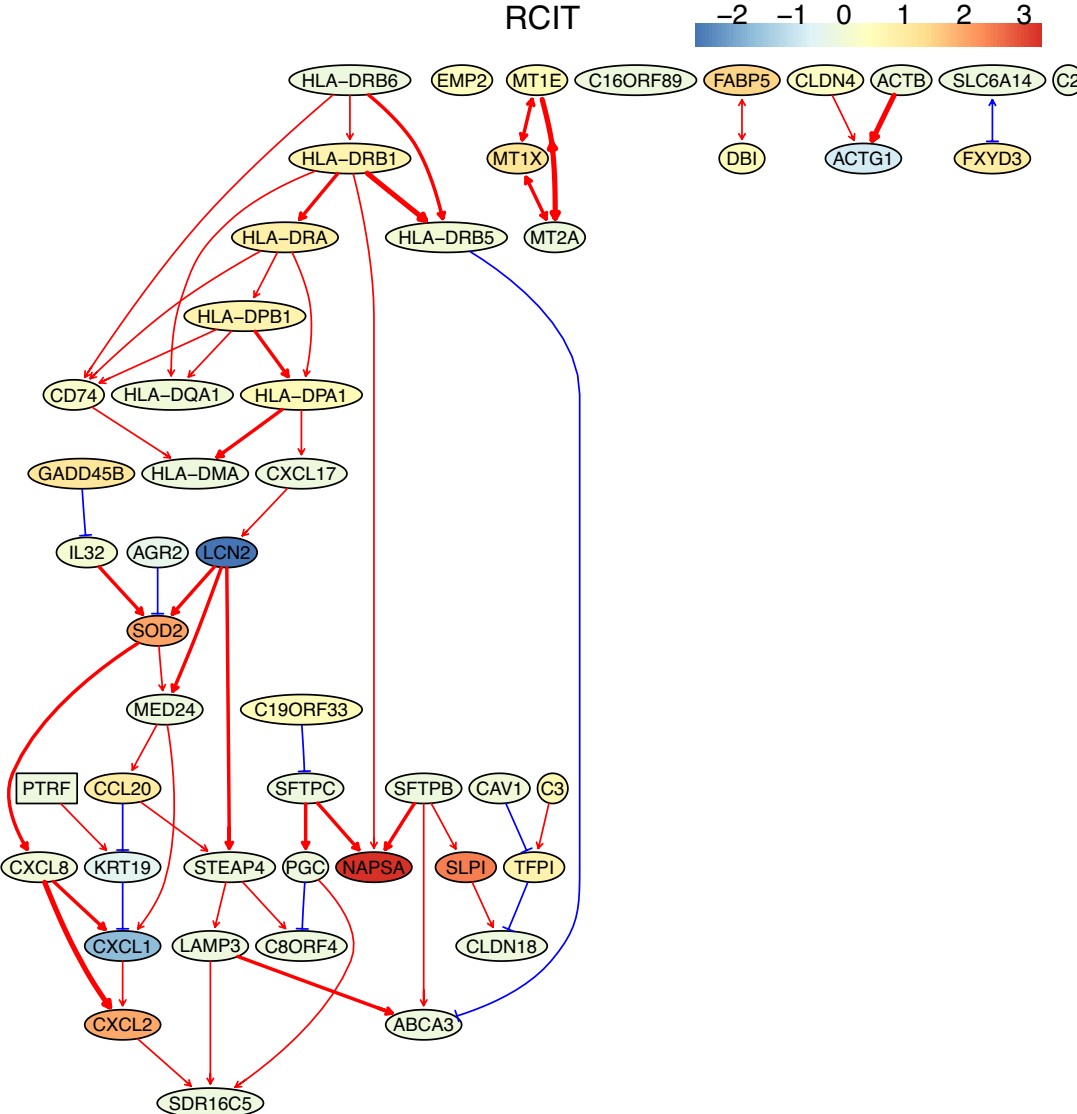

**Appendix 5—figure 7.** The causal network of the 50 feature genes inferred by PC+RCIT from the alveolar epithelial cell dataset (settings: feature genes were expressed in >50% cells, the alpha level for CI test was 0.1, and the 300 cells with more feature genes expressed the cells were used). The control of the case was the H2228 cells. Compared with H2228 cells, MHC-II genes in alveolar epithelial cells were highly expressed.

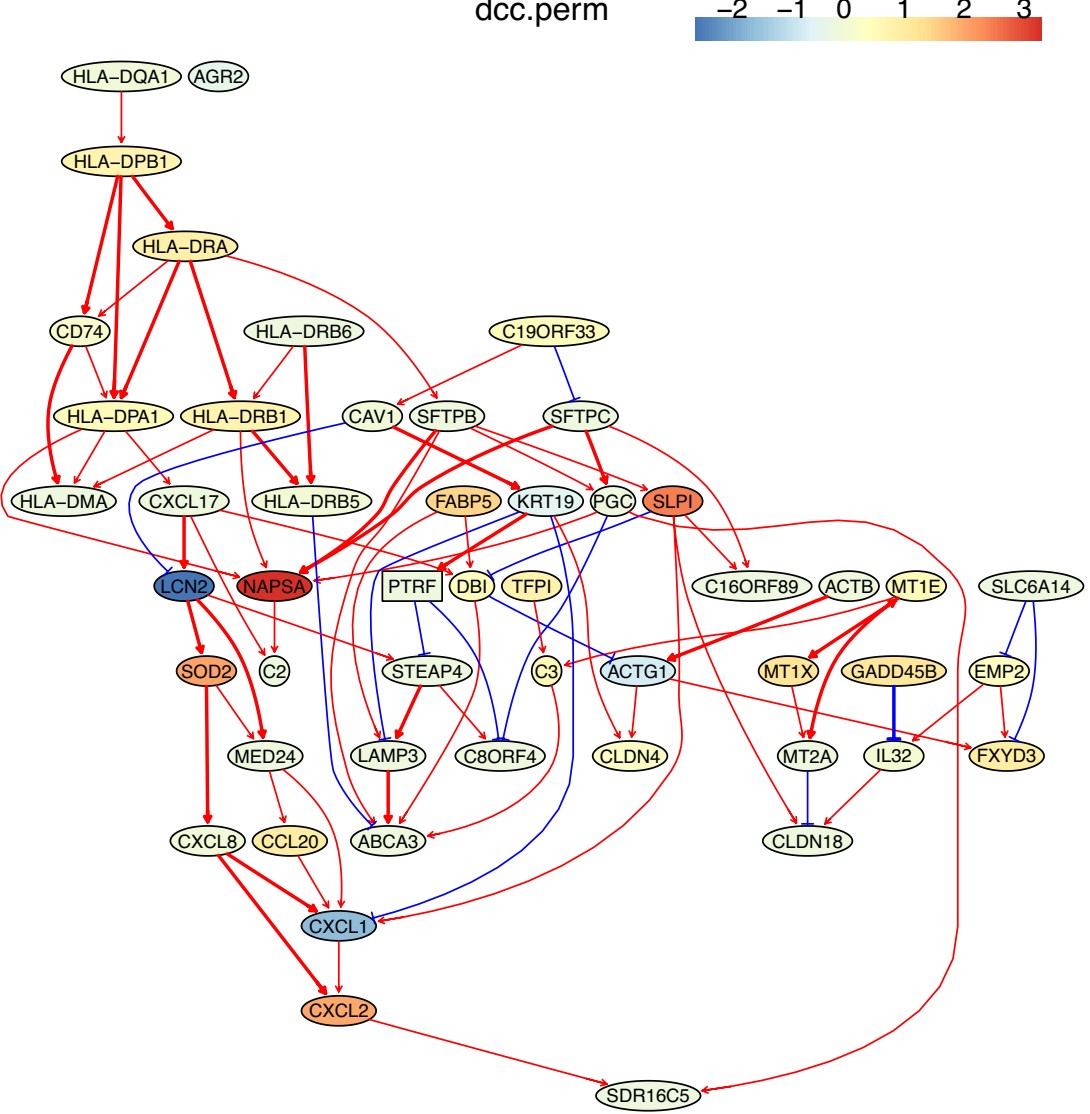

**Appendix 5—figure 8.** The causal network of the 50 feature genes inferred by PC+DCC.perm from the alveolar epithelial cell dataset (settings: feature genes were expressed in >50% cells, the alpha level for CI test was 0.1, and the 300 cells with more feature genes expressed the cells were used). The control of the case was the H2228 cells. Compared with H2228 cells, MHC-II genes in alveolar epithelial cells were highly expressed.

## 2. The analysis of the macrophages from glioblastoma

After using the dataset of macrophage isolated from glioblastoma to examine feature selection algorithms, we also used it to examine causal discovery algorithms. Again, feature genes include HLA genes to examine whether reported interactions are inferred (*Appendix 5—figures 9 and 10*).

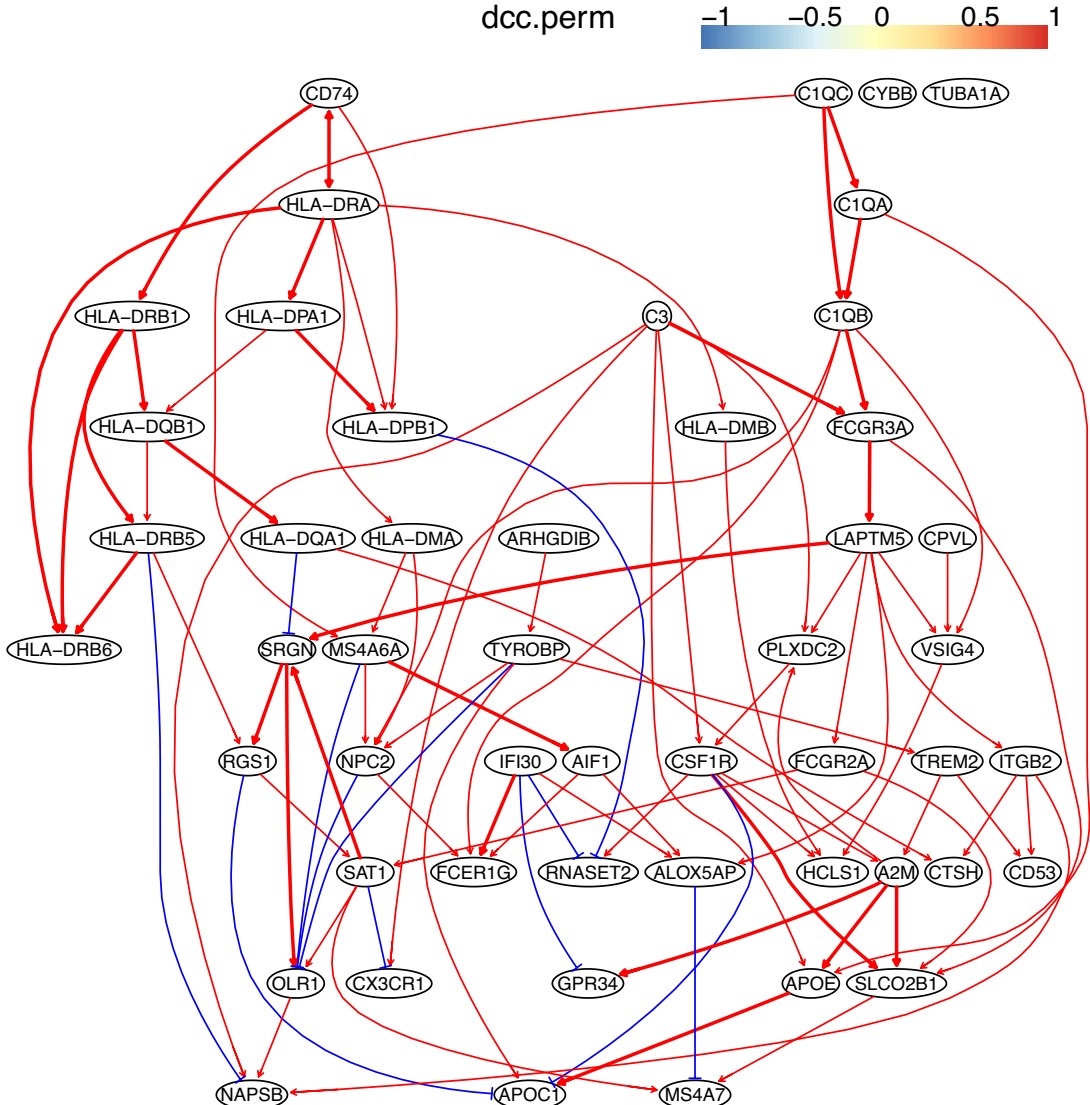

**Appendix 5—figure 9.** The causal network of the 50 feature genes inferred by PC+DCC.perm from the macrophage dataset (macrophages isolated from human glioblastoma) (setting: feature genes were expressed in >50% cells, the alpha level for CI test was 0.1, and the 300 cells with more feature genes expressed were used). Because no control data was used, the differential expression of genes was not computed. Note the interactions between MHC-II genes and CD74, between C1QA/B/C, and the TYROBP→TREM2→A2M→ APOE→APOC1 cascade.

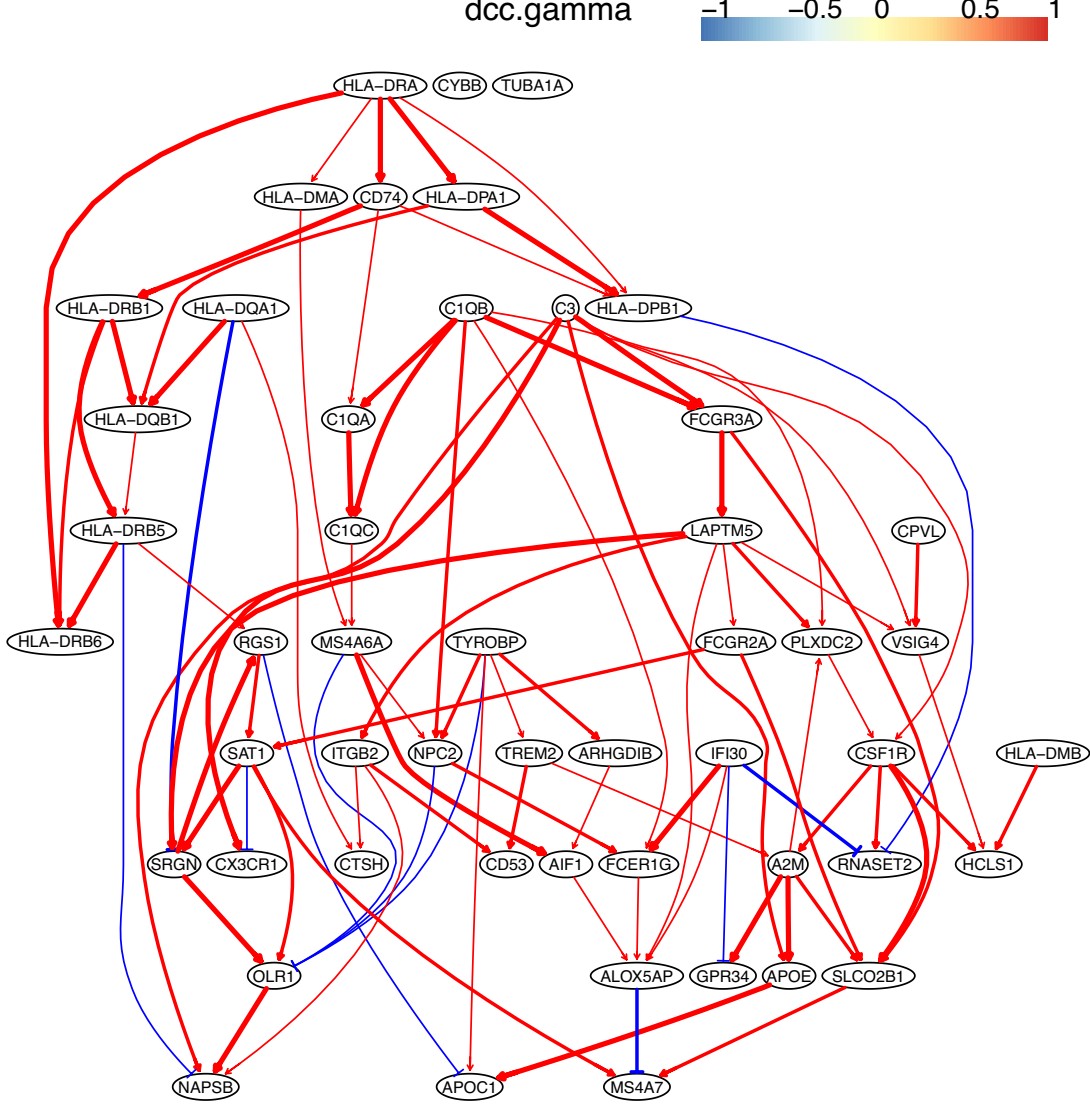

**Appendix 5—figure 10.** The causal network of the 50 feature genes inferred by PC+DCC.gamma from the macrophage dataset (macrophages isolated from human glioblastoma) (setting: feature genes were expressed in >50% cells, the alpha level for CI test was 0.1, and the 300 cells with more feature genes expressed were used). Because no control data was used, the differential expression of genes was not computed. Note the interactions between MHC-II genes and CD74, between C1QA/B/C, and the TYROBP→TREM2→A2M→ APOE→APOC1 cascade.

## 3. The analysis of exhausted CD8 T cells from multiple cancers

We used TOX and PDCD1 as the target gene, respectively, to select 50 genes from genes expressed in >50% exhausted CD8 T cells (from liver, colorectal, and lung cancers) and in >50% non-exhausted CD8 T cells (from the normal tissues neighboring these cancers). Networks with TOX and PDCD1 as the target gene are called TOX-network and PDCD1-network, respectively. In this application case, we demonstrate consensus networks; unless otherwise specified, all panels are consensus networks of the two DCC algorithms. Therefore, we use letters but not algorithms to label panels. Networks were inferred from 500 cells (the case of colorectal cancer) and 463 cells (the case of lung cancer). Exhausted and non-exhausted were mutually used as case and control. In panels, →→ and -|-| represent indirect activation and inhibition (*Appendix 5—figures 11–17*).

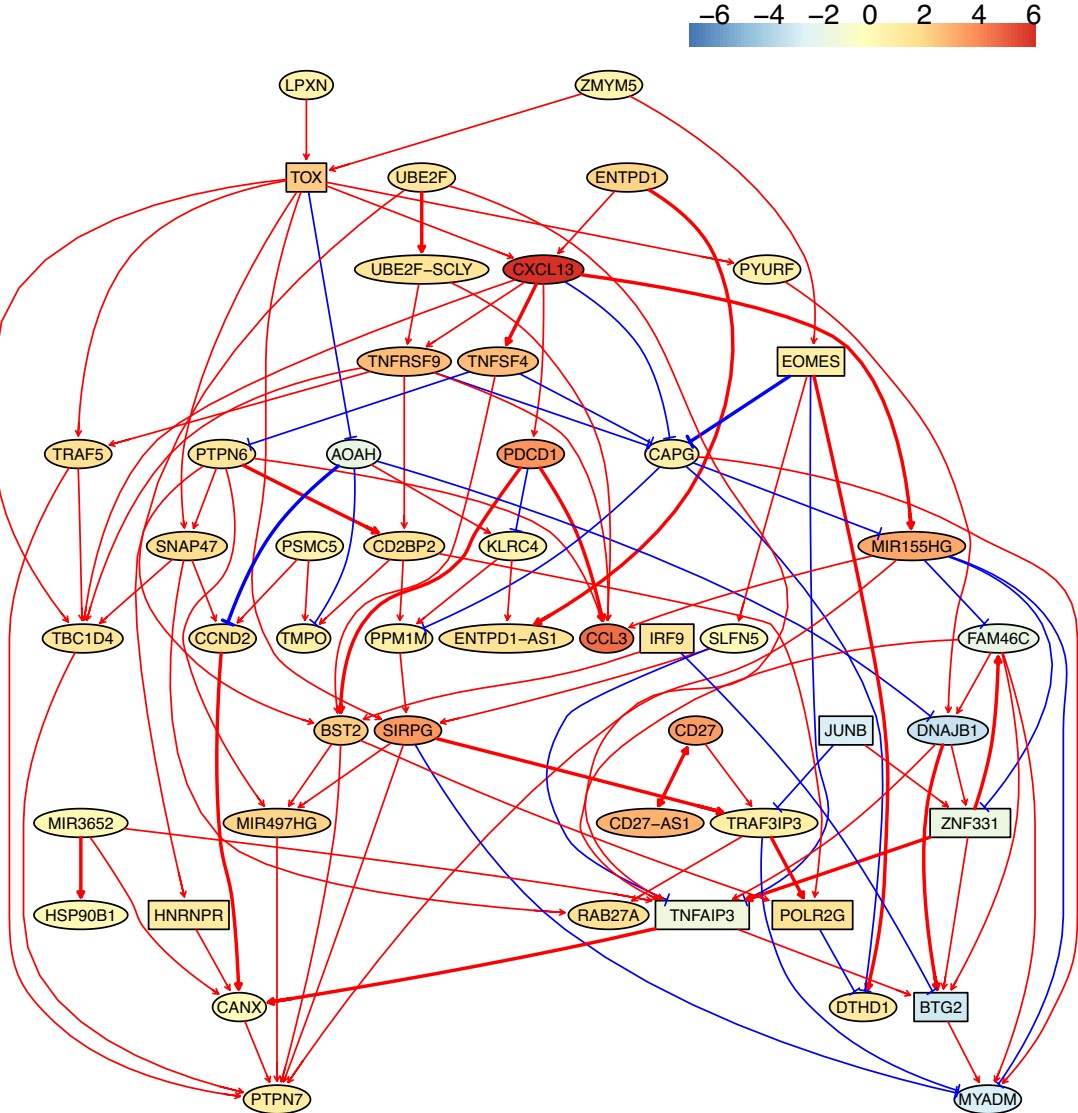

**Appendix 5—figure 11.** The networks of 50 genes in exhausted CD8 T cells and non-exhausted CD8 T cells from colorectal and lung cancers and the normal tissues neighboring the cancers. (**A**) The TOX-network inferred from exhausted CD8 T cells from colorectal cancer. TOX→→PDCD1 (TOX→CXCL13→PDCD1), TNFRSF9→TRAF5, and TOX→→MIR155HG have related reports including (1) TOX up-regulates PDCD1 expression (*Khan et al., 2019*), (2) TNF receptors bind to TRAF2/5 to activate NF-kB signaling, (3) in mice up-regulated miR-155 represses Fosl2 by inhibiting Fosb and causes long-term persistence of exhausted CD8 T cells during chronic infection (*Stelekati et al., 2018*). We found that if more feature genes were selected (to include FOSB), the MIR155HG-|YPEL5→DNAJB1→FOSB were inferred, agreeing with inhibited FOXB by up-regulated MIR155.

# consensusNetwork (dcc.gamma + dcc.perm)

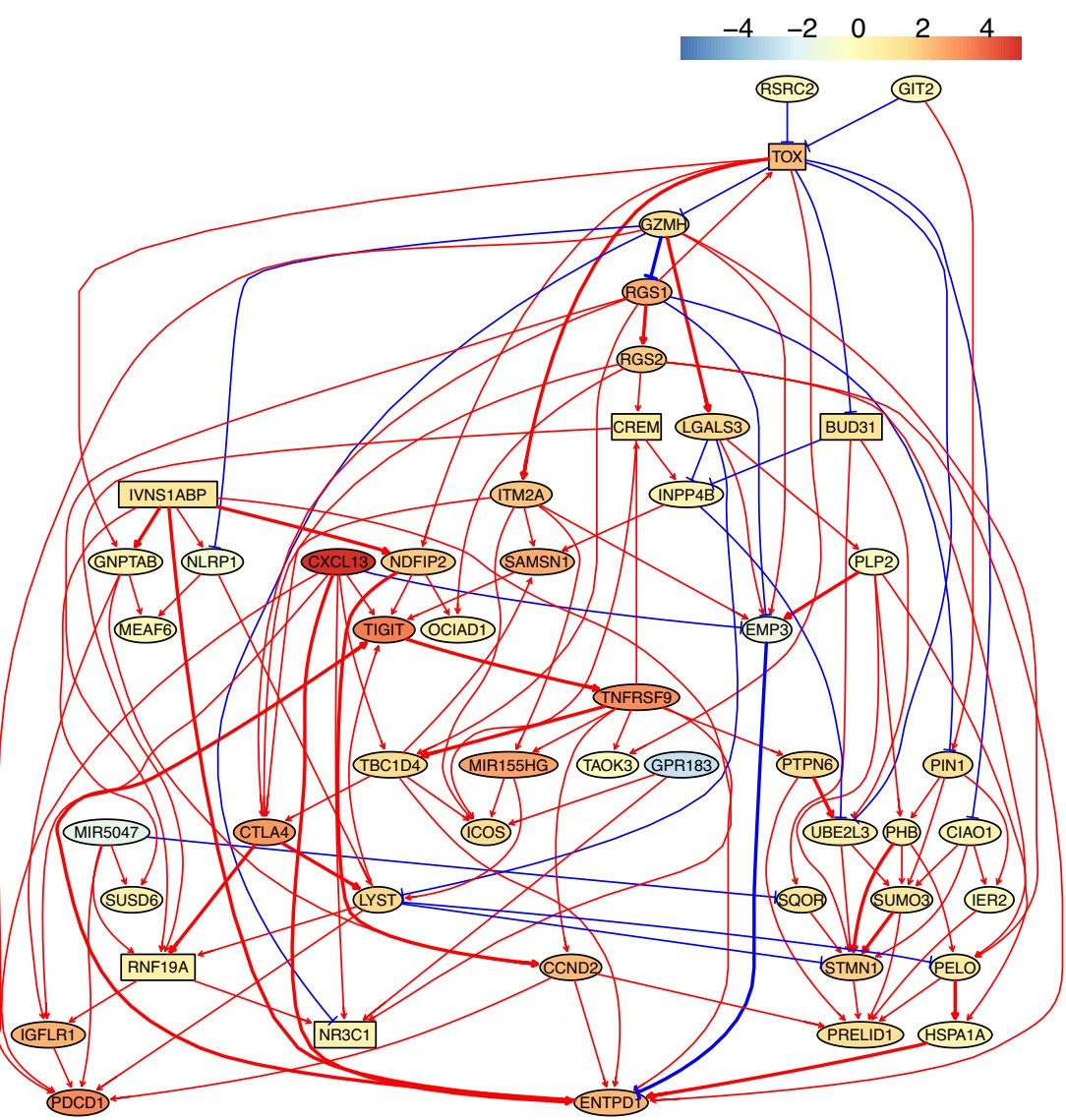

**Appendix 5—figure 12.** The networks of 50 genes in exhausted CD8 T cells and non-exhausted CD8 T cells from colorectal and lung cancers and the normal tissues neighboring the cancers. (**B**) The TOX-network inferred from exhausted CD8 T cells from lung cancer. Different route of TOX→→PDCD1 and TOX→→MIR155HG were inferred (i.e. TOX→GNPTAB→IGFLR1→PDCD1, TOX→ITM2A→MIR155HG).

## consensusNetwork (dcc.gamma + dcc.perm)

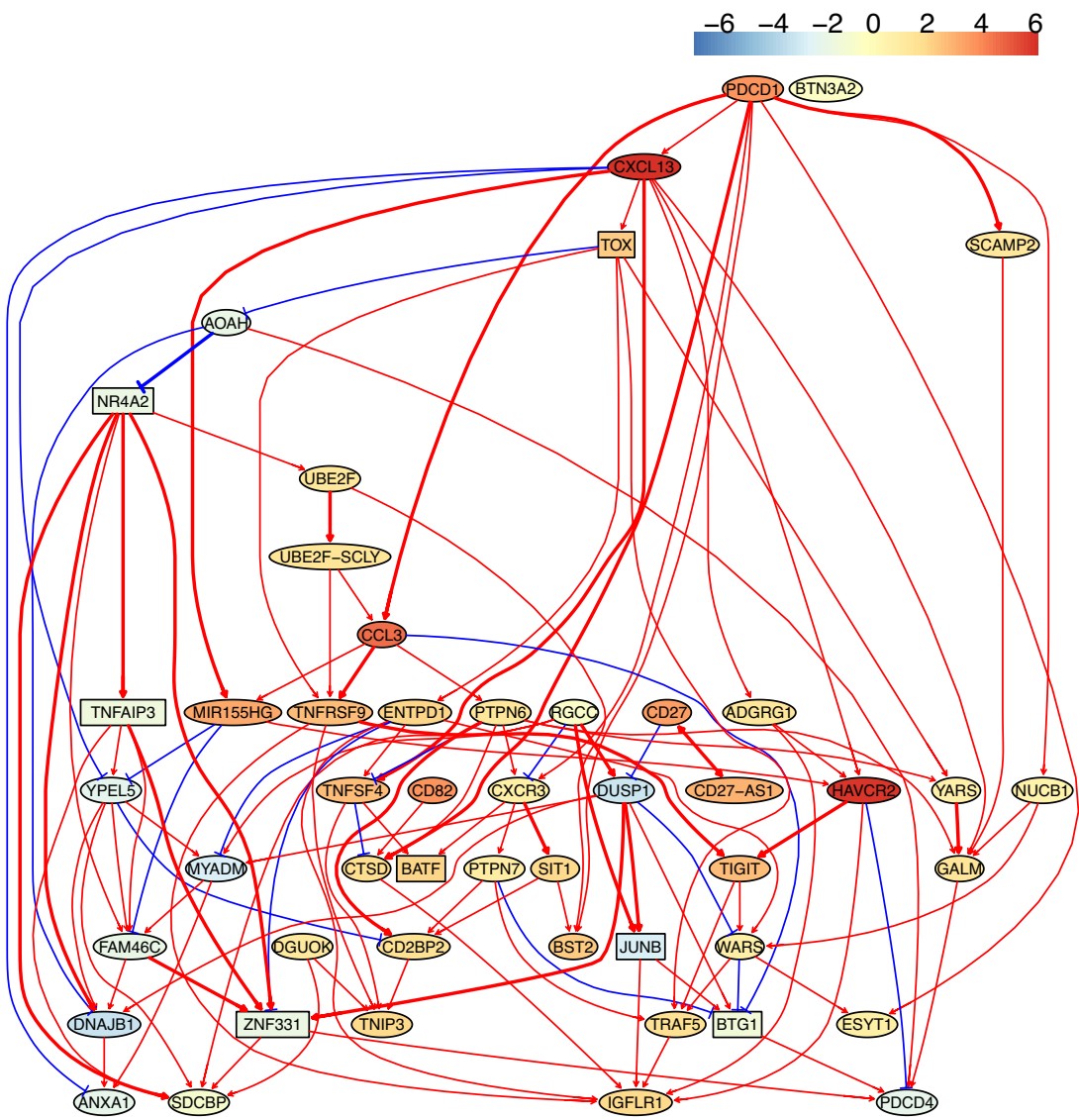

**Appendix 5—figure 13.** The networks of 50 genes in exhausted CD8 T cells and non-exhausted CD8 T cells from colorectal and lung cancers and the normal tissues neighboring the cancers. (**C**) The PDCD1-network inferred from exhausted CD8 T cells from colorectal cancer. PDCD1→→TOX (PDCD1→CXCL13→TOX), PDCD1→→MIR155HG (there were two routes: PDCD1→CXCL13→MIR155HG, PDCD1→CCL3→MIR155HG), MIR155HG-|YPEL5→DNAJB1, and HAVCR2-|PDCD4 were inferred. Related reports of these interactions include (1) TOX transcription factors cooperate with NR4A transcription factors to impose CD8+ T cell exhaustion (***Seo et al., 2019***), (2) CCL3 is one of the up-regulated chemokine genes in exhausted CD8 T cells (***Wherry et al., 2007***).

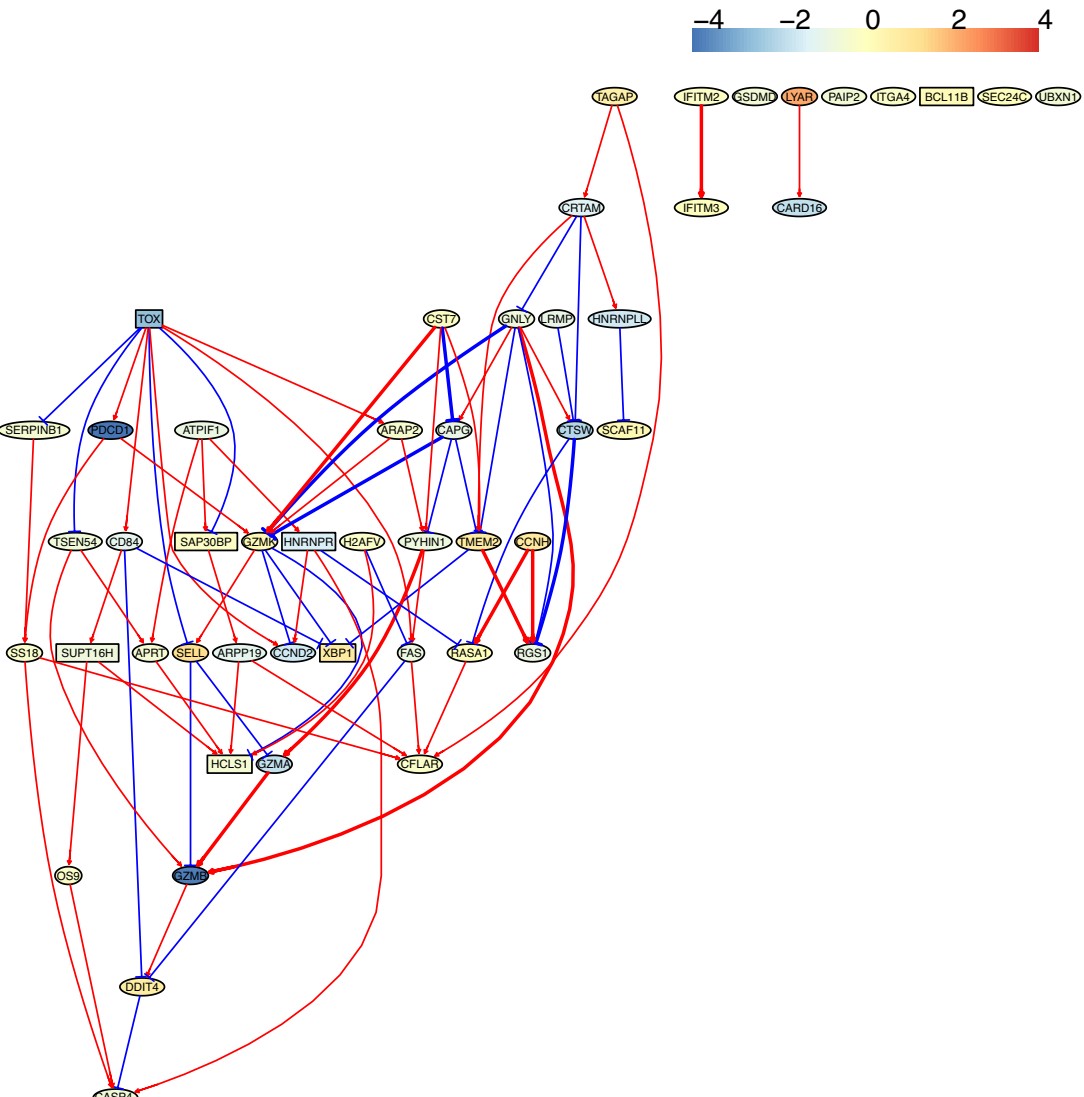

**Appendix 5—figure 14.** The networks of 50 genes in exhausted CD8 T cells and non-exhausted CD8 T cells from colorectal and lung cancers and the normal tissues neighboring the cancers. (**D**) The TOX-network inferred from non-exhausted CD8 T cells from the normal tissue neighboring colorectal cancer. Direct TOX→PDCD1 was inferred, and MIR155HG was not associated with TOX.

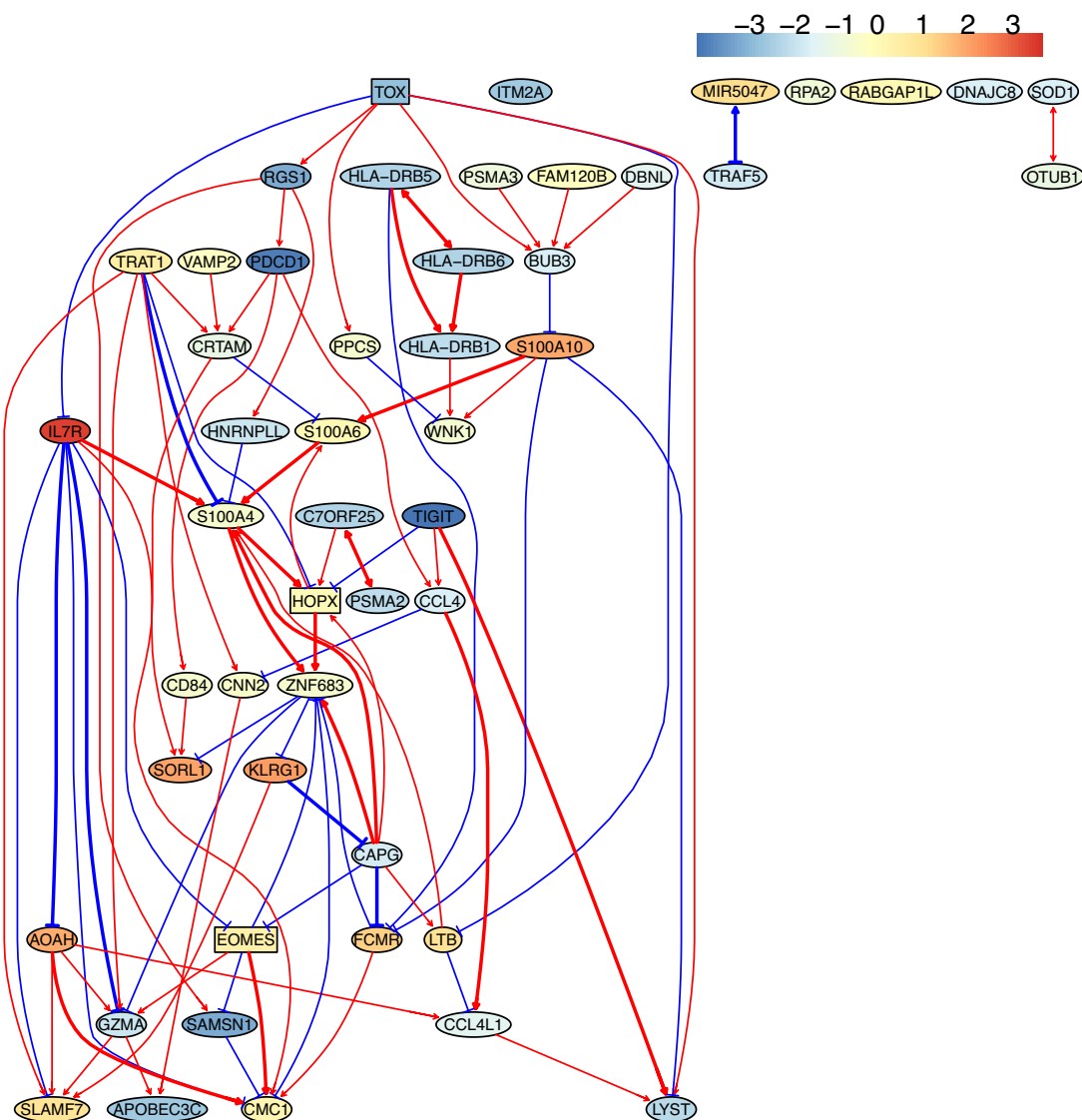

**Appendix 5—figure 15.** The networks of 50 genes in exhausted CD8 T cells and non-exhausted CD8 T cells from colorectal and lung cancers and the normal tissues neighboring the cancers. (**E**) The TOX-network inferred from non-exhausted CD8 T cells from the normal tissue neighboring lung cancer. TOX→RGS1→PDCD1 and interactions between HLA-DRB2, HLA-DRB6, and HLA-DRB1 were inferred.

## consensusNetwork (hsic.gamma + hsic.perm)

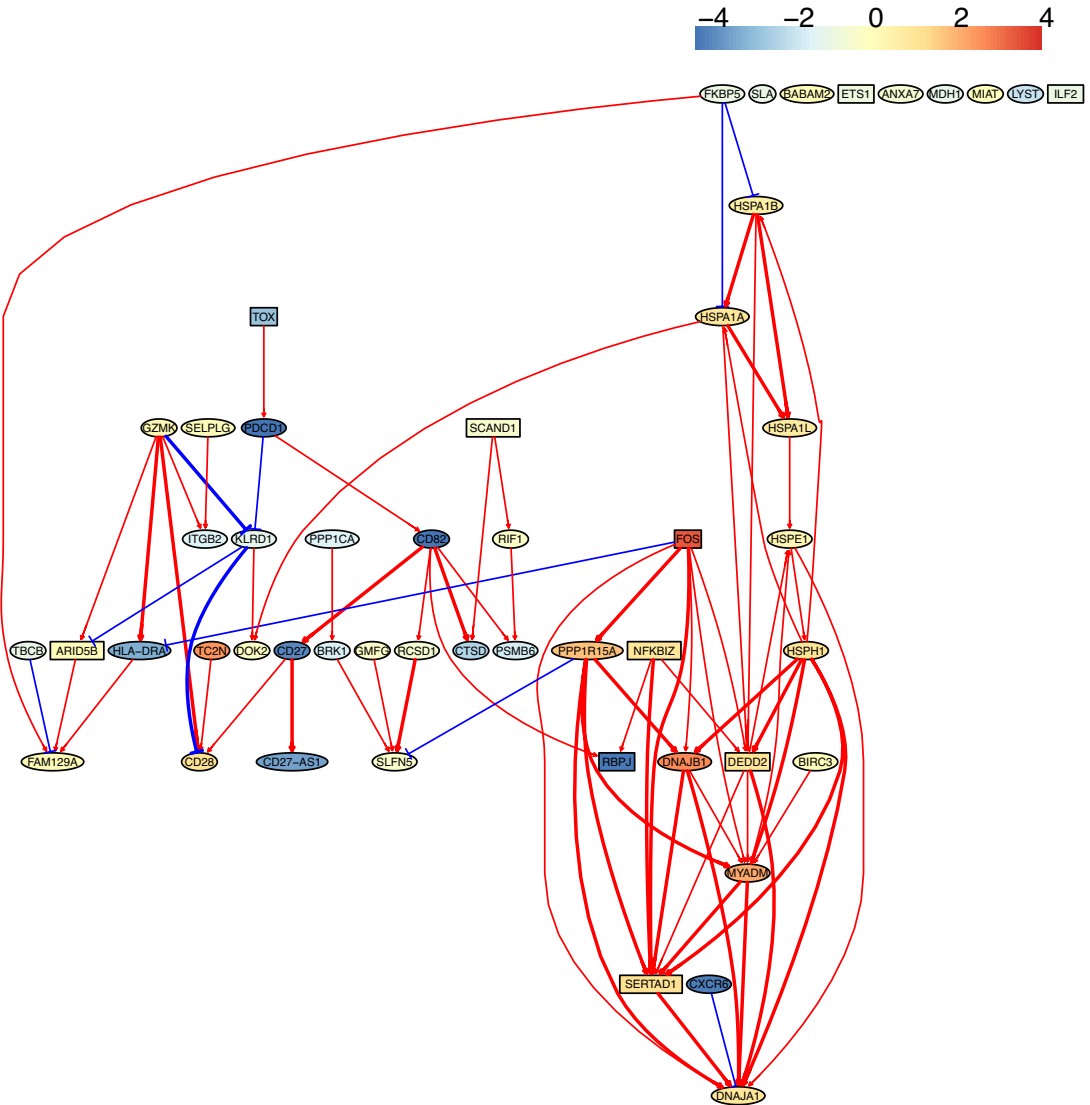

**Appendix 5—figure 16.** The networks of 50 genes in exhausted CD8 T cells and non-exhausted CD8 T cells from colorectal and lung cancers and the normal tissues neighboring the cancers. (**F**) The PDCD1-network of HSIC algorithms inferred from non-exhausted CD8 T cells from the normal tissue neighboring colorectal cancer.

consensusNetwork (dcc.gamma + dcc.perm)

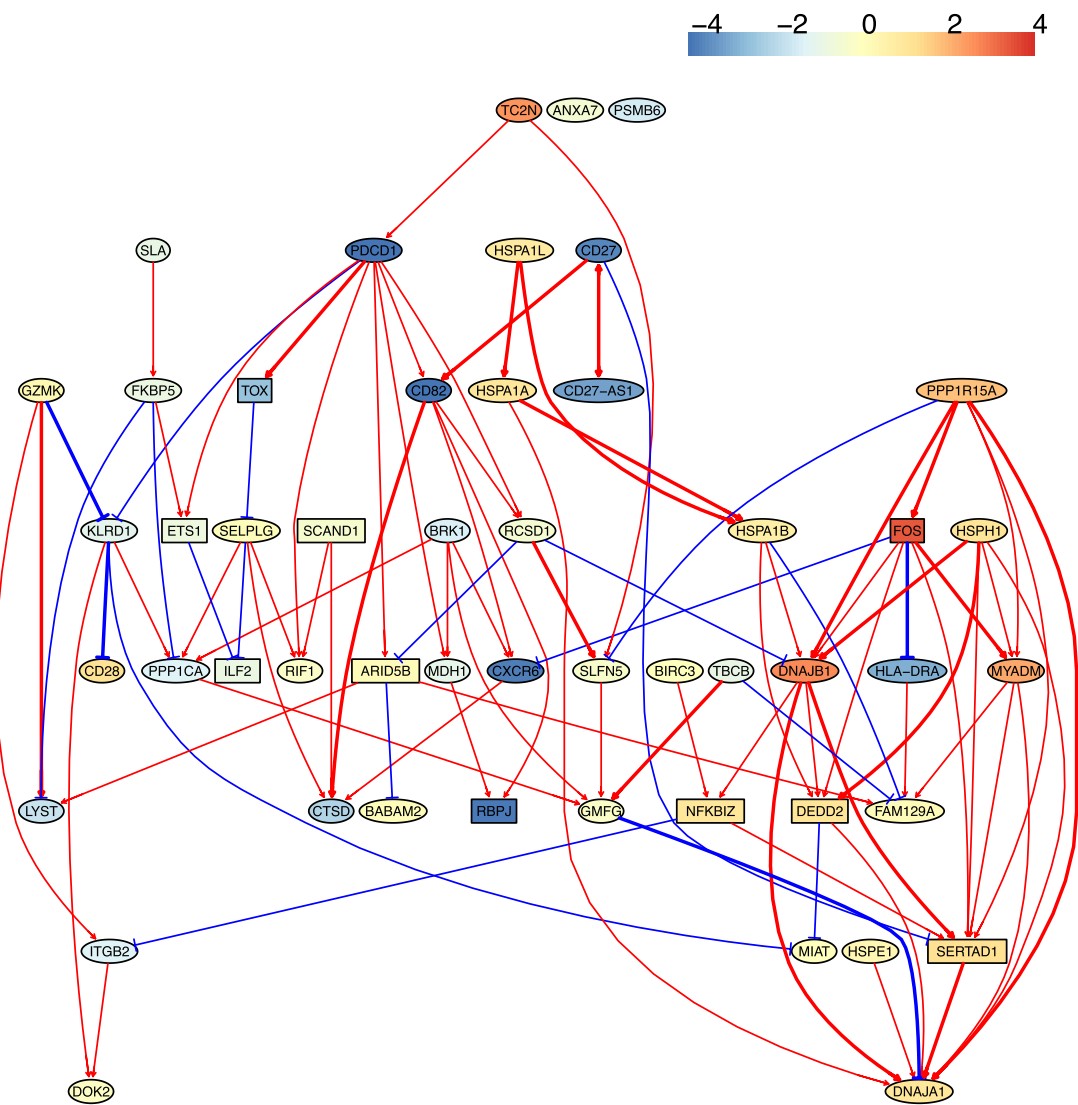

**Appendix 5—figure 17.** The networks of 50 genes in exhausted CD8 T cells and non-exhausted CD8 T cells from colorectal and lung cancers and the normal tissues neighboring the cancers. (**G**) The PDCD1-network inferred from non-exhausted CD8 T cells from the normal tissue neighboring colorectal cancer.

## 4. The analysis of CD4 T cells from young and old mice

Since aging occurs gradually and ubiquitously in almost all cells, we assumed that consistent up- or down-regulation in all CD4 T cell types better defines CD4 aging-related genes than large fold changes. Upon this, we obtained the presumably CD4 aging-related genes (**Appendix 5—table 1**). Many of these genes are not the senescence signatures (**Gorgoulis et al., 2019**), indicating that different genes may be involved in the aging of different cells, but the mitochondrial genes have been well recognized as being important for aging in many cells.

Data in the STRING database support many inferred interactions, especially interactions between the mitochondrial genes, between Ccnd2, Ccnd3, Cdkn1b, and Cdkn2d, between B2m and H2-Q7, between Lck and Cd28, and between Gm9843 and Rps27rt (**Appendix 4—figure 3**). Interactions supported by experimental findings include Cdc42→Coro1a (CDC42 and CORO1A exhibit strong associations both with age) (**Kerber et al., 2009**), Arpc1b→Coro1a (in mouse T cells Coro1a is involved in Arp2/3 regulation) (**Shiow et al., 2008**), B2m→H2-Q7 (B2m is associated with the MHC class I heavy chain) (**Smith et al., 2015**), Lck→Cd28 (Lck is found to associate with CD28 by using its

SH2 domain to bind to a phospho-specific site) (*Rudd, 2021*), Cdc42-|Lamtor2 (mTOR is required for asymmetric division through small GTPases in mouse oocytes) (*He et al., 2013*; *Lee et al., 2012*), Ccnd2-|Lamtor2 (mTORC1 activation regulates beta-cell mass and proliferation by modulation of Ccnd2 synthesis and stability) (*Balcazar et al., 2009*), and Sub1-|Ccnd2-|Lamtor2 (Sub1 can accelerate aging via disturbing mTOR-regulated proteostasis) (*Chen et al., 2021*).

Several inferred results are noticeable. First, interactions among the mitochondrial genes were inferred in all cases, whose expression levels were low in cells from young mice but high in cells from old mice. These indicate that these genes may be common biomarkers of aging for CD4 T cells. Second, in the inferred networks, these mitochondrial genes do not have consistent inputs and outputs, which can probably be explained by the finding that the metabolic system undergoes extensive rewiring upon normal T-cell activation and differentiation (*Zhang et al., 2021b*). With the report of increasing experimental findings, mitochondrial dysfunction in aging and diseases of aging has drawn increasing attention (*Haas, 2019*). Third, Junb is activated. Persistent JUNB activation in human fibroblasts enforces skin aging and the AP-1 family TFs (including FOSL2 and JUNB) are increased in all immune cells during aging (*Maity et al., 2021*; *Zheng et al., 2020*). The findings of Junb/JUNB indicate that JUNB/Junb plays a critical role in aging. Fourth, the Gm9843→Rps27rt→Junb cascade (Rps27rt is also called Gm9846, and both Gm9843 and Rps27rt are mouse-specific genes) was inferred in many cases; it is interesting whether these interactions' counterparts exist in humans (*Appendix 5—figures 18–21*).

**Appendix 5—table 1.** Up- and down-regulated genes in CD4 T cells from young and old mice.

| Gene | FC >0 **cases** | Annotation and evidence | References |
|---|---|---|---|
| Rpl28 | 24 | Ribosomal proteins influence aging. | *Kirkland et al., 1993*; *Steffen and Dillin, 2016* |
| Arpc1b | 24 | Arpc1b may induce senescence in a p53-independent manner. | *Li et al., 2022*; *Yun et al., 2011* |
| Smc4 | 24 | | *Goronzy and Weyand, 2019*; *McCartney et al., 2021* |
| Sub1 | 23 | Sub1 is increased and becomes activated with age, and transgenic expression of PC4 disturbs mTOR-regulated proteostasis and causes global accelerated aging. | *Chen et al., 2021* |
| Cdc37 | 23 | | |
| Lck | 23 | Lck is a positive regulator of inflammatory signaling and a potential treatment target for age-related diseases. | *Garcia and Miller, 2009*; *Kim et al., 2019* |
| Cdc42 | 23 | Mouse model studies have found that aging is associated with elevated activity of the Rho GTPase Cdc42 in hematopoietic stem cells. In humans, CDC42 and CORO1A exhibited strong associations with age. | *Amoah et al., 2021*; *Geiger and Zheng, 2013*; *Kerber et al., 2009* |
| Ccnd2 | 22 | Ccnd2 is an aging marker | *Goronzy and Weyand, 2019*; *McCartney et al., 2021* |
| Ccnd3 | 22 | Ccnd2 is an aging marker | *Goronzy and Weyand, 2019*; *Li et al., 2020*; *McCartney et al., 2021* |
| Foxp1 | 22 | FOXP1 controls mesenchymal stem cell commitment and senescence during skeletal aging. | *Li et al., 2017* |
| Coro1a | 21 | CORO1A is a senescence-related gene. | *Avelar et al., 2020*; *Kerber et al., 2009* |
| Gm26740 | 21 | A mouse-specific gene without annotation. | |
| Lamtor2 | 20 | MAPK and MTOR activator 2. It is involved in the activation of mTORC1. | *Morita et al., 2017*; *Walters and Cox, 2021* |
| Lsp1 | 20 | Lymphocyte-specific protein 1; may play a role in mediating neutrophil activation and chemotaxis. | |
| Gene | FC <0 cases | Annotation and evidence | References |

*Appendix 5—table 1 Continued on next page*

*Appendix 5—table 1 Continued*

| Gene | FC >0 **cases** | Annotation and evidence | References |
|---|---|---|---|
| Rbm3 | 24 | Muscle from aged rats exhibited an increase in heat shock protein (HSP) 25 and HSP70 and in the cold shock protein RNA-binding motif 3 (RBM3). | *Dupont-Versteegden et al., 2008*; *Van Pelt et al., 2019* |
| H2-Q7 | 23 | A strong increase of the MHC class I genes (including H2-Q7) and B2m is observed in the aging lung. | *Angelidis et al., 2019* |
| Btg1 | 23 | Btg1 is involved in neural aging. | *Micheli et al., 2021* |
| Gm9843 | 23 | A mouse-specific gene without annotation. | |
| Rps27rt (Gm9846) | 23 | Ribosomal protein S27 retrogene, mouse-specific. | |
| mt-Atp6 | 22 | | |
| mt-Co1 | 22 | | |
| mt-Co2 | 22 | | |
| mt-Co3 | 22 | Mitochondrial proteins involved in the electron transport chain are overrepresented in cells from older participants, with prevalent dysregulation of oxidative phosphorylation and energy metabolism molecular pathways. | *Bektas et al., 2019*; *Goronzy and Weyand, 2019*; *Haas, 2019* |
| mt-Nd1 | 21 | | |
| Junb | 21 | JUNB is increased in all human immune cells during aging. | *Maity et al., 2021*; *Zheng et al., 2020* |
| Psme1 | 21 | Proteasome activator subunit 1. It is implicated in immuno-proteasome assembly and required for efficient antigen processing. | *Hwang et al., 2007* |
| B2m | 20 | B2m is in GO:0007568, a mouse aging GO term. B2M is elevated in the blood of aging humans and mice. | *Smith et al., 2015* |

| Gene | Other cases | Annotation and evidence | References |
|---|---|---|---|
| Cd28 | 12 | Cd28 is an aging biomarker of T cells. | *Le Page et al., 2018*; *Zhang et al., 2021a* |
| Cdkn2d | 6 | Cdkn2d is an aging biomarker. | *Goronzy and Weyand, 2019* |
| Cdkn1b | 5 | Cdkn1b is an aging biomarker. | *Goronzy and Weyand, 2019* |

#1Genes and numbers in red indicate fold change (FC)>0, genes and numbers in blue indicate FC <0, genes and numbers in black do not show clear differential expression in a majority of cell groups.

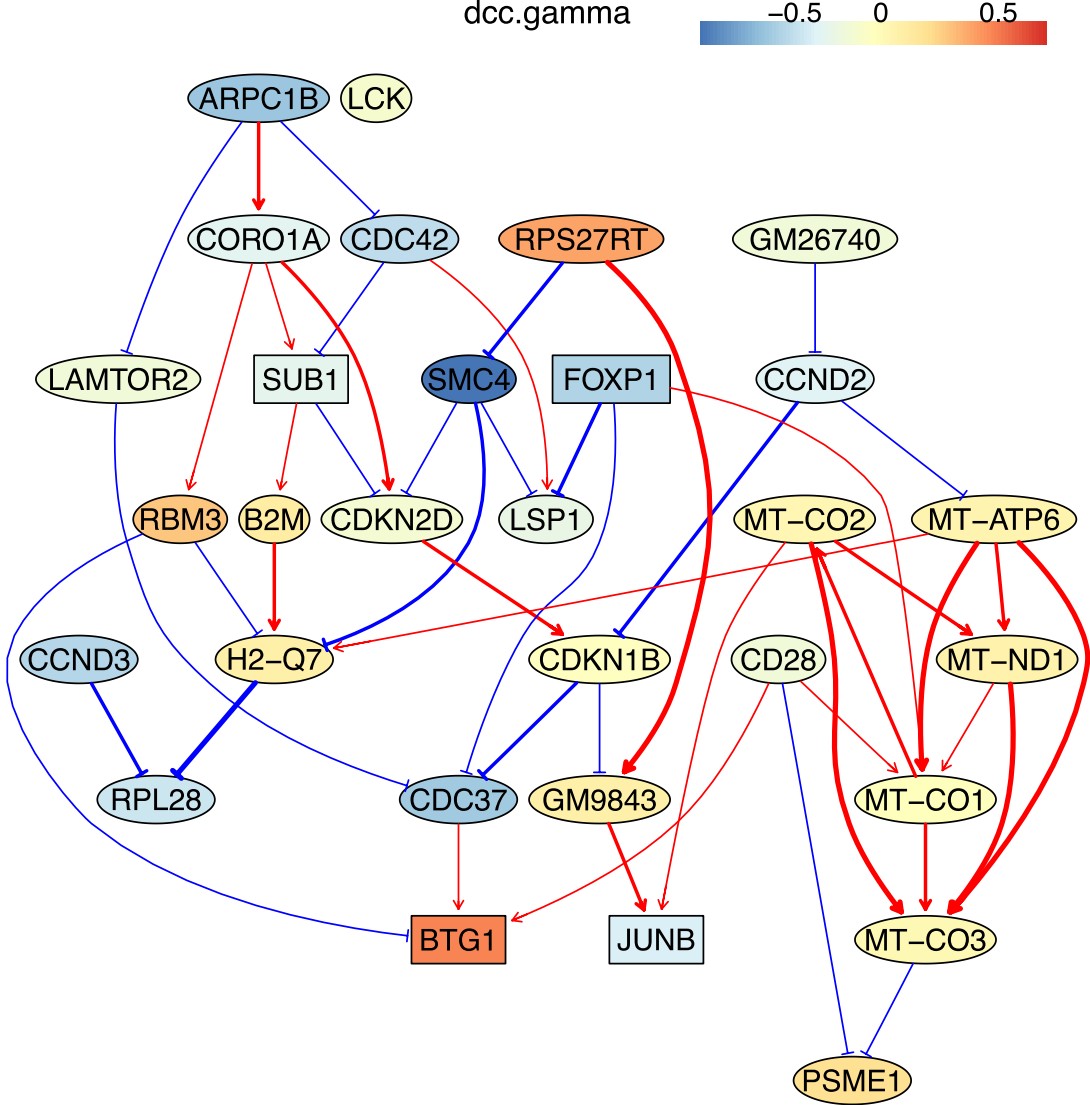

**Appendix 5—figure 18.** The causal relationships between genes that were differentially expressed in old cytotoxic CD4 T cells. The network was inferred from 600 cells.

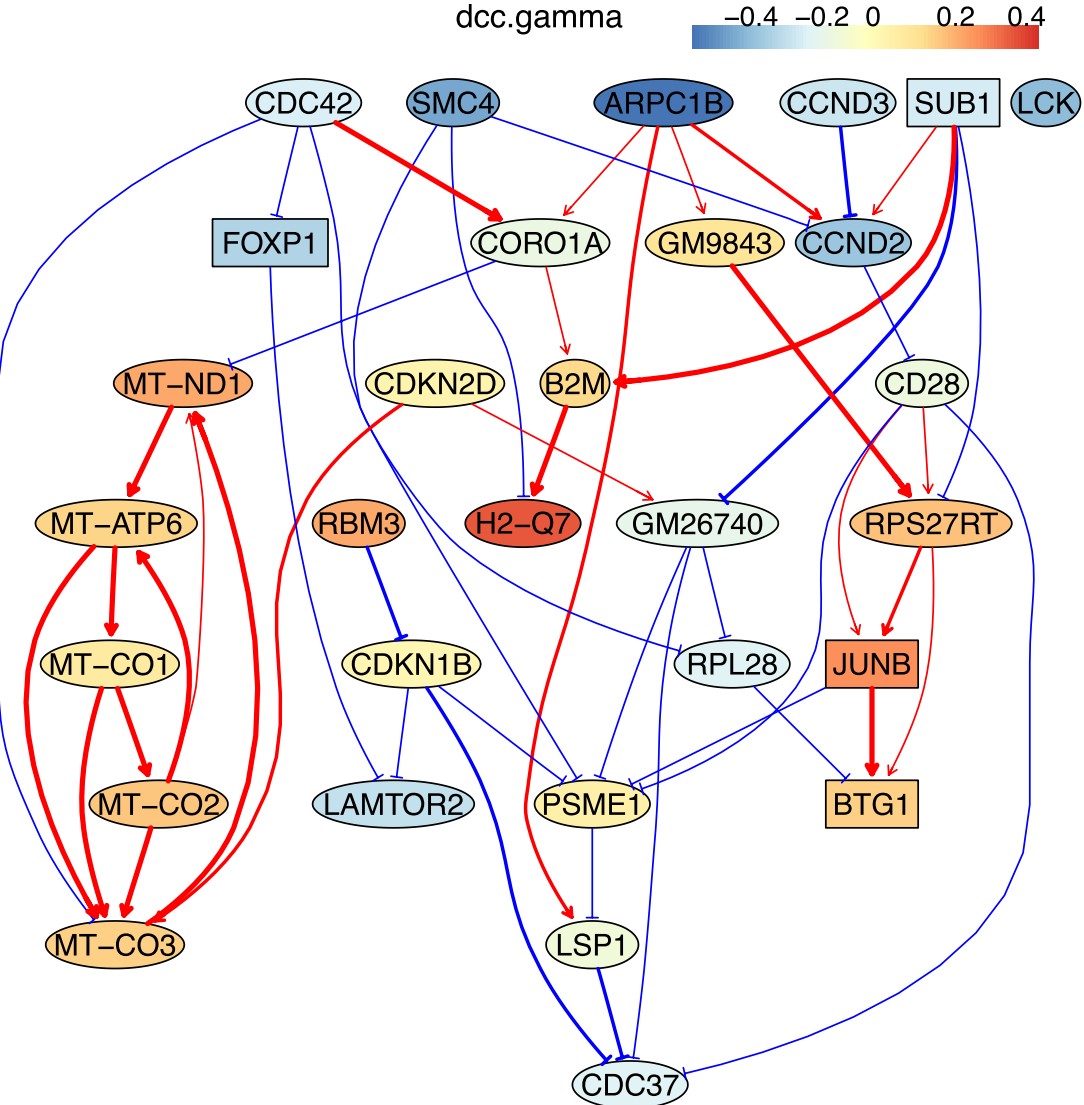

**Appendix 5—figure 19.** The causal relationships between genes that were differentially expressed in old exhausted CD4 T cells. The network was inferred from 600 cells.

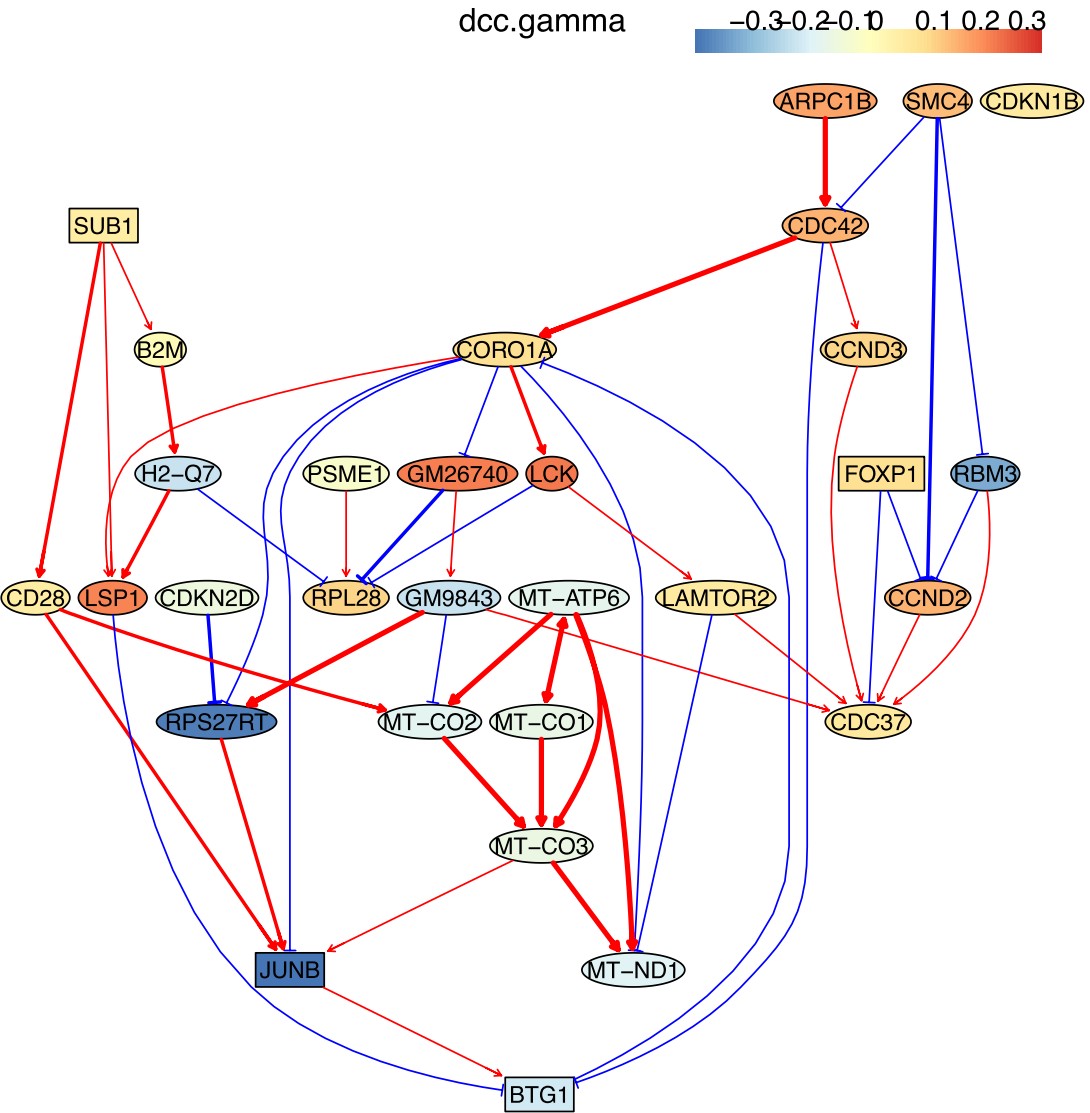

**Appendix 5—figure 20.** The causal relationships between genes that were differentially expressed in young TEM CD4 T cells. The network was inferred from 600 cells.

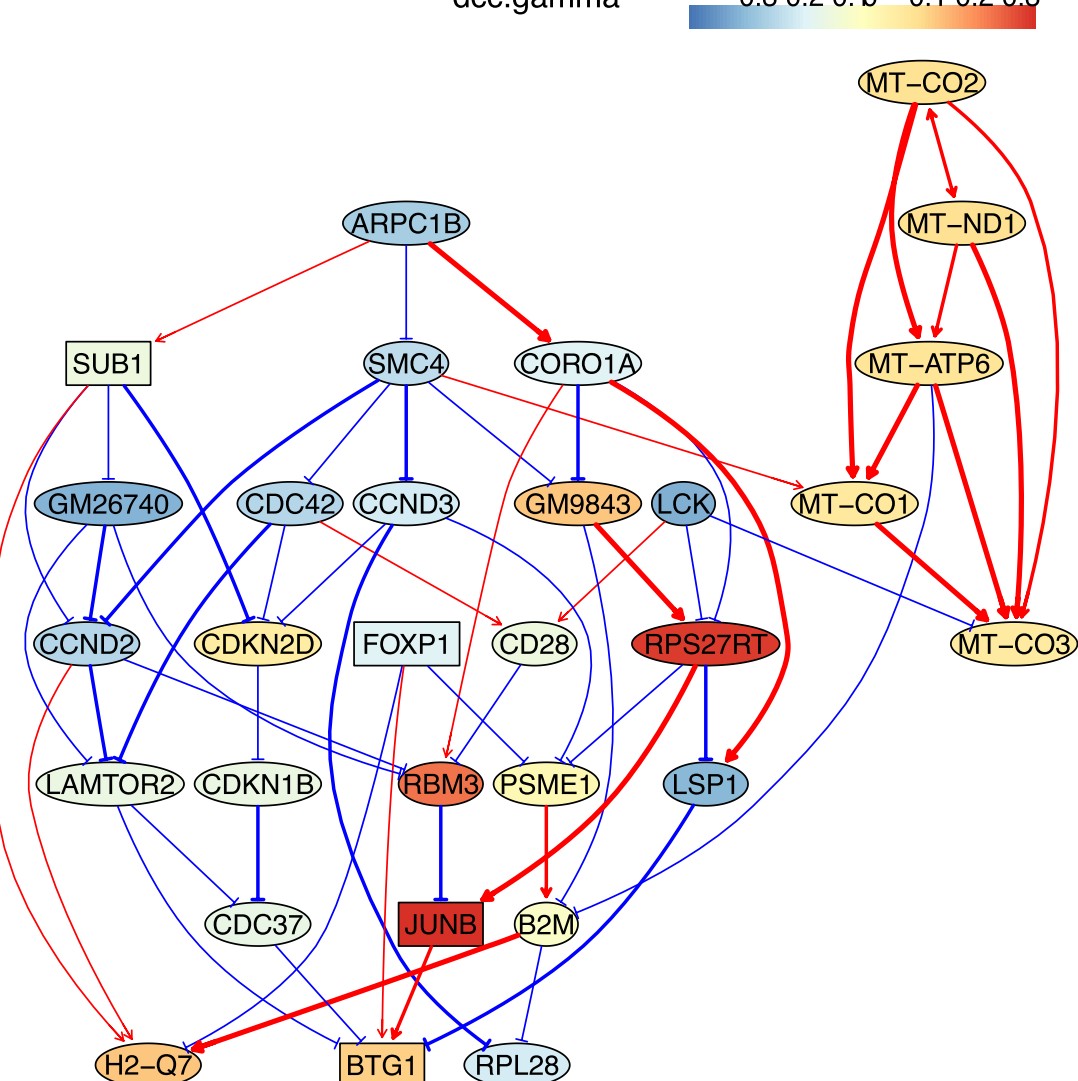

**Appendix 5—figure 21.** The causal relationships between genes that were differentially expressed in old TEM CD4 T cells. The network was inferred from 600 cells.

## 5. The analysis of a flow cytometry dataset

Finally, we analyzed the flow cytometry data reported by Sachs et al. This dataset, due to the ground truth given by the authors, has been used to test other algorithms. The computed structural intervention distance (SID) and SHD between networks inferred by different algorithms and the ground truth network also suggest that the DCC CI tests outperform others. See *Appendix 5—figure 22*.

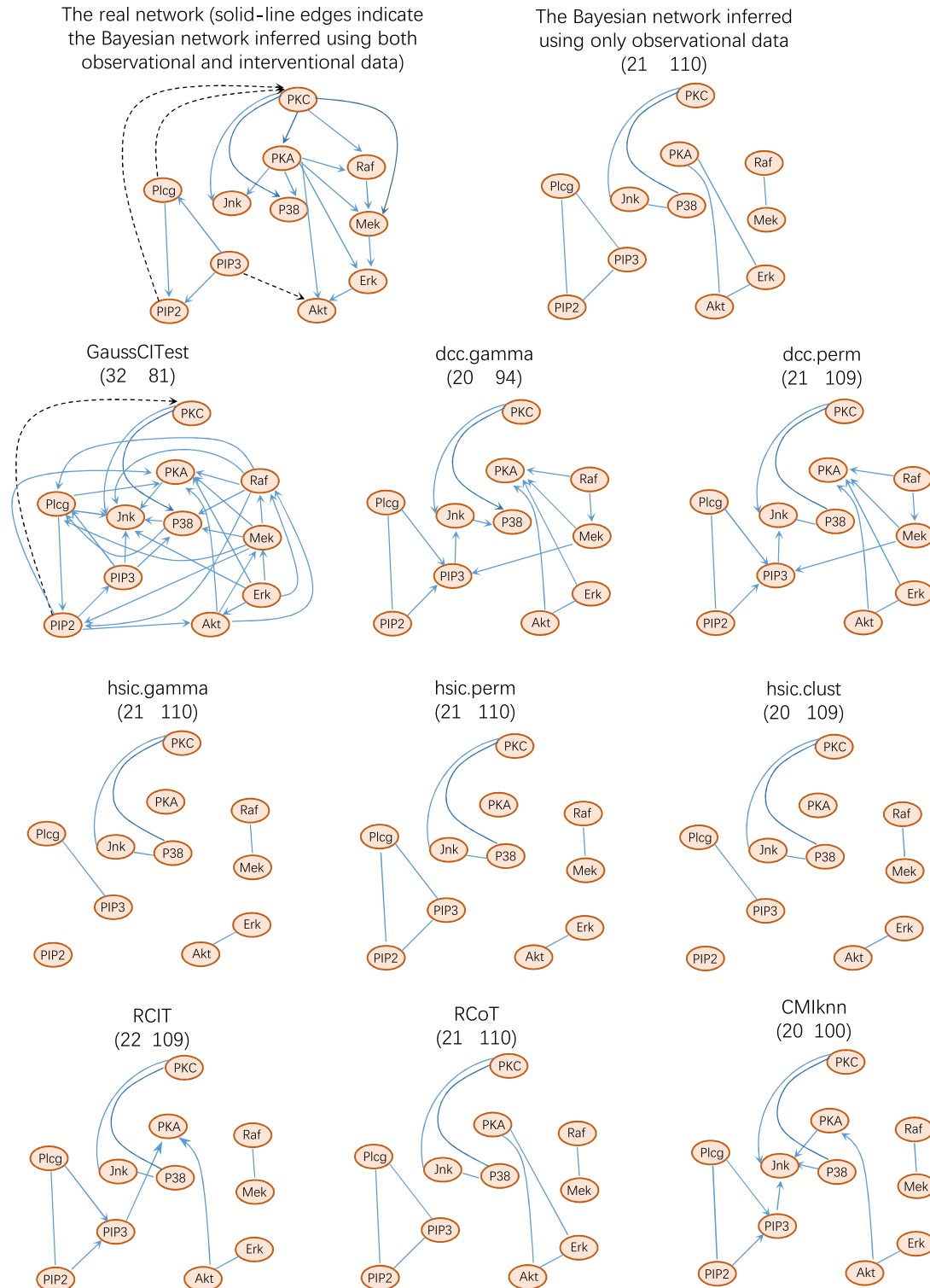

**Appendix 5—figure 22.** The performance of causal discovery algorithms upon the Sachs dataset (*Sachs et al., 2005*). Structural intervention distance (SID) is another important measure for evaluating causal graphs. The numbers in the bracket are structural Hamming distance (SHD) and SID values (the smaller, the better). These values indicate that DCC.gamma and DCC.perm outperform others. The network inferred by Bayesian inference contains only undirected edges.

