## [Editor Report]

This manuscript presents an important tool for causal inference intended for the analysis of single cell datasets but possibly with broader applications. It compares several algorithms and incorporates a number of them in the platform and offers convincing evidence of its usefulness. With the rapid expansion of large datasets, this tool is beneficial in offering several causal inference analysis options and expediting the interpretation of data.

---

## [Decision Letter]

**Decision letter after peer review:**

Thank you for submitting your article "CausalCell: applying causal discovery to single-cell analyses" for consideration by *eLife*. Your article has been reviewed by 2 peer reviewers, one of whom is a member of our Board of Reviewing Editors, and the evaluation has been overseen by Anna Akhmanova as the Senior Editor.

Essential revisions:

This manuscript presents a tool for causal inference intended to be used for the analysis of single-cell datasets. The tool, named CausalCell, attempts to address a quite important question in the network biology field. That is how to infer a directed gene network to reveal causal relationships among genes. Given scRNA-seq data and a set of genes of interest, CausalCell is potentially useful for inferring cell type-specific intracellular causal networks among genes. Although in principle this can be a very helpful tool, the evidence that the incorporated algorithms are the most suitable for the proposed applications is inadequate. Data preprocessing, result illustration, and validation should be substantially strengthened.

1) The authors need to either justify their choice of algorithms in the paper or expand the work to include other important algorithms.

2) Throughout the paper, the authors need to be more explicit about the description of preprocessing as well as the quantification of stated claims.

3) Stronger validation is needed to support and clarify the usefulness of the proposed tool.

*Reviewer #1 (Recommendations for the authors):*

1. There seem to be three general categories of causal inference algorithms: constraint-based, score-based, and hybrid (see for example, https://proceedings.neurips.cc/paper/2017/file/275d7fb2fd45098ad5c3ece2ed4a2824-Paper.pdf).

This paper appears to focus only on different implementations of constraint-based algorithms; however, it appears that score-based algorithms in general produce better results for complex biologically-driven datasets. I have checked this with experts since this is a little outside of my field. In my opinion, this is a major shortcoming that significantly reduces the impact of the current manuscript. In other words, the authors need to justify that the algorithm (or group of algorithms) they have picked is the best option for single-cell analyses (in doing so, the existing efforts and literature should be addressed). Alternatively (and especially if there is no consensus about what algorithm works best), they should incorporate other options into the tool.

2. Some statements in the comparison section need clarification. For example, in Table 1, the authors mention that "Both ExtraTrees and RandomForest perform well, but XGBoost does not." I believe it is necessary to state more details about what the context/condition is, in order to justify such claims.

3. Figure 2: The comparison shows a relatively poor similarity between different algorithms. What is the reason for this?

4. In Discussions, the authors state that kernel-based CI tests perform better than faster methods that make additional assumptions. I am not sure if I see the evidence for this. Especially when different kernel-based algorithms poorly agree with each other, it is unclear how a case can be made. Ideally, this issue can be addressed with a "mock" dataset in which the causal links are already known. In the absence of a known "ground truth" or a "gold standard", their arguments about accuracy and performance are in general less convincing.

5. Line 353: The authors make a statement about the dataset being "large enough". I'd suggest including a formal treatment, in which the causal inference results are compared as the number of data points increases (for example by sub-sampling one of the existing datasets). I believe this will provide more convincing evidence for how many data points are required and what to expect as that number increases/decreases.

*Reviewer #2 (Recommendations for the authors):*

1) Reactome and KEGG databases curated directed links between genes or proteins, which is suitable for result evaluation.

2) 50 feature genes represent a small fraction of the whole genome. This means CausalCell cannot fully take advantage of the high-throughput feature of current scRNA-seq technologies. Any suggestions to deal with this concern?

3) What techniques do the authors use to generate the consensus network?

4) On lines 68-70, what are the differences among regulatory networks, causal networks, (ordinary) networks, and gene networks? Readers could be confused by such an introduction.

5) Figure 4 and other causal network plots should have a legend to indicate the color scheme for the upregulation or downregulation of gene expression.

6) Volume and page information of Reference [49] are missing.

[Editors' note: further revisions were suggested prior to acceptance, as described below.]

Thank you for resubmitting your work entitled "Applying causal discovery to single-cell analyses using CausalCell" for further consideration by *eLife*. Your revised article has been evaluated by Anna Akhmanova (Senior Editor) and a Reviewing Editor.

The manuscript has been improved but there are some remaining issues that need to be addressed, as outlined below:

1. The need to include more recent advances in causal discovery, such as continuous optimization-based, neural network-based methods is brought up by the reviewers again. There is a need to either include these approaches or offer concrete evidence that the PC algorithm works better (or at least equally well) for the type of biological questions the tool is addressing. This is perhaps best done by a direct comparison of results between the methods included in CausalCell and some other representatives such as Notears, Golem, DAG-GNN, lingam, and Dlingam.

2. A more thorough comparison of independence/CI tests with more recent methods such as regression-based, ranking-based, and deep neural network-based CI tests (e.g., MLP-based and GAN-based) needs to be included.

Additionally, please address the more detailed issues raised by Reviewer #3 below.

*Reviewer #3 (Recommendations for the authors):*

This work developed a workflow and platform for effectively performing causal discovery from scRNA‐seq data. The workflow/platform is developed upon the benchmark of 9 feature selection algorithms, 3 causal discovery methods, 9 CI tests, and the analyses of multiple datasets. The authors suggest that kernel-based conditional independence tests generate reliable results. Some key issues are discussed, and tips for best practices are provided. In my opinion, this work has the potential to help biologists discover some causal relationships among single-cell data, but the main drawback of this approach is the lack of new technologies on causal discovery as well as CI tests, and the biological significance of the work is not quite clear. Following are some of my concerns or questions about this work:

1. In recent years, continuous optimization-based methods have become the most popular method for causal discovery, which yield much better performance than the PC algorithm, I think this work should discuss and take some representative continuous optimization-based methods into account, such as Notears, Golem, DAG-GNN. There are also some causal functional model-based methods such as lingam and Dlingam should be discussed.

2. The 9 independence/CI tests might not stand for the state-of-the-art, more kinds of, and more recent methods should be taken into account, such as regression-based, ranking-based, and deep neural network-based CI tests (e.g., MLP-based & GAN-based).

3. Theoretically, not all causal directions can be discovered by the orientation step of the PC algorithm, how to address the Markov equivalence classes in this paper? And there are also some other constraint-based methods, why choose PC?

4. I suggest the author present the time complexity of each method not just 'time consumption', as it seems a little bit confusing. For example, HSIC.gmma should work much faster than HSIC.perm, but they are both '*' at 'time consumption'.

5. How to perform PC with HSIC? only 0-order CI test?

6. See "……However, because most algorithms have been designed for handling limited variables and few algorithms have been evaluated using real data, applying causal discovery to single-cell data remains challenging……". I don't quite agree with this statement, because there are lots of constraint-based methods (or combining with feature selections as dimension reduction step) with different CI tests for causal discovery on RNA-sq/Microarray, and this work also does a similar or same job.

7. See "……These features of CI tests enable causation between any genes and molecules to be inferred……". It should be noted that not all causation between them can be inferred by CI tests.

8. " ……the time consumption of kernel-based CI tests disallows large-scale network inference……", how about the parallel PC? And sometimes one can limit the size of the conditional set.

9. Actually, I wonder whether this workflow/platform can find some interesting biological results (say biomarkers) from the data. This is related to the biological significance of the work. Computational results in the paper do not provide convincing support for this point.

---

## [Author Response]

Essential revisions:This manuscript presents a tool for causal inference intended to be used for the analysis of single-cell datasets. The tool, named CausalCell, attempts to address a quite important question in the network biology field. That is how to infer a directed gene network to reveal causal relationships among genes. Given scRNA-seq data and a set of genes of interest, CausalCell is potentially useful for inferring cell type-specific intracellular causal networks among genes. Although in principle this can be a very helpful tool, the evidence that the incorporated algorithms are the most suitable for the proposed applications is inadequate. Data preprocessing, result illustration, and validation should be substantially strengthened.1) The authors need to either justify their choice of algorithms in the paper or expand the work to include other important algorithms.

(1) We have examined the representative score-based method GES (Chickering 2003) and the representative hybrid method GSP (i.e., score-based + CI tests) (Solus et al. 2021) (see also the NIPS-2017 paper recommended by Reviewer 1, i.e., Wang et al. 2017 cited in this Responses-to-Comments). We used genes in the WikiPathway pathway "Metabolic reprogramming in colon cancer" (WP4290) and the data of the five lung cancer cell lines to compare the PC, GES, and GSP methods. The results suggest that GES is somewhat less accurate, but faster, than PC+DCC.γ and GSP+DCC.γ. At the same significance level, the networks of GES have more isolated nodes and more unreasonable interactions (see Appendix file 3). We think this is partly because GES is a parametric approach and assumes Gaussian distribution of data, and nonlinear relationships, missing values, and latent variables may impair the performance of GES. On the other hand, both PC+DCC.γ and GSP+DCC.γ capture the common features of reprogrammed glucose metabolism in cancer cells, but GSP+DCC.γ infers more interactions than PC+DCC.γ, even with a more stringent significance level (α=0.05 rather than 0.1) (Appendix file 3).

(2) We have incorporated GES and GSP into the CausalCell platform. The original GSP package includes just two CI tests (i.e., GaussCItest and HSIC.γ); we now enable it to run with the 9 CI tests to provide varied options for users.

2) Throughout the paper, the authors need to be more explicit about the description of preprocessing as well as the quantification of stated claims.

(1) On data preprocessing. We have greatly revised the section "3. Data input and preprocessing" and made the following points clear. (a) CausalCell accepts log2-transformed data and z-score data and can turn raw data into either of the two forms. (b) Data can be sorted upon any attribute (e.g., expression value, variance, and fold change). (c) The user can filter genes upon multiple conditions (e.g., expression value, variance, and fold change) so as to, for example, select differentially expressed genes or genes with high variance. (d) Filtering genes upon specific conditions is an important preprocessing step for feature selection, because applying feature selection to genes genome-wide may impair the accuracy of feature selection.

(2) We have added the option "TF=Yes/No" as a new condition for filtering genes, which makes all selected feature genes be transcription factors. More details are given in the responses to Reviewer 2's comments.

(3) On removing batch effects (raised by Review 2): our situations. (a) We analyzed the exhausted and non-exhausted CD8 cells. All samples were sequenced by the same protocol, each dataset was from the same kind of tumor, and data were preprocessed and integrated by the original authors (see related citations). (b) We analyzed the mouse CD4 data. We note that batch effects are very limited, as all samples were sequenced by the same protocol and cells did not cluster topologically by depth of sequencing, experimental batch, or individual mouse. Also, we manually organized the inputs without detecting differentially expressed genes and performing feature selection. (c) We analyzed the five non-small cell lung cancer cell lines and lung alveolar cells. According to the information in the paper and GEO website, these five cell lines were sequenced in the same batch and preprocessed carefully for benchmarking scRNA-seq analysis pipelines (including batch effects removal). We do not have the raw data of the lung alveolar cells and did not remove batch effects. By using the five cell lines to make the 5-to-1 comparisons, we think the revealed key differences between the lung cancer cell lines and alveolar cells are reliable. (d) When using the five lung cancer cell lines and lung alveolar cells to strengthen the validation, we used genes in KEGG/WikiPathway pathways as inputs without detecting differentially expressed genes and performing feature selection.

(4) On removing batch effects: the user's situations. If the user selects feature genes from differentially expressed genes, batch effects may have an influence. We make it clear in the revised manuscript that it is the user's responsibility to remove batch effects. This is because (a) varied methods have been developed to remove batch effects with varied performance (Tran et al. 2020), and (b) removing batch effects should be performed before integrating batches, and we do not know how users' data are generated and whether the uploaded data are integrated or not.

(5) On the quantification of stated claims. We think the claims on algorithms' performance and on sample size are important. (a) We revised Table 1 and Table 2. A note is added to Table 1 and Table 2, respectively, explaining how the performance indicators are estimated and how advantage/disadvantage is made. (b) We recommend 300 and 600 cells for Smart-seq and 10X Genomics data in the main text. The new claim is "Our application and validation examples suggest that 300 and 600 cells are recommended for reliable inference if the input is Smart-seq2 and 10X Genomics data, the input contains about 50 genes, and genes are expressed in >50% cells. Here, reliable inference means that key (but not all) interactions are inferred (e.g., shared in similar datasets and/or with high significance, see Appendix file 4), and 300 and 600 are estimated rather than accurate. More cells are needed if input genes are expressed in less cells and if the input contains more genes. While more cells make more interactions (also more false positives) be inferred, the key interactions are stable". (c) We also explain the recommendation of 300 and 600 cells in the section "Tips for best practices".

(6) In the in the original Appendix file 3, we wrote "Third, to evaluate the impact of sample size on algorithms' performance, we ran the 9 algorithms using 200 (instead of 300) H2228 cells. The results of 200 cells were poorer than the results of 300 cells (compared with the consensus network in Main-figure 2 and Appendix file 3-figure 2), and the two DCC algorithms performed the best and were less sensitive to the decreased sample size than the two HSIC algorithms". On the newly added claim "While more cells make more interactions (also more false positives) be inferred, the key interactions are stable", we used genes in the WP4290 pathway and the dataset of H838 to compare networks inferred using different sub-samples (200 cells, 400 cells, 600 cells, and 800 cells). The results are given in Appendix file 3 and indicate that more cells cause more interactions be inferred but the key interactions are quite stable.

3) Stronger validation is needed to support and clarify the usefulness of the proposed tool.

(1) As described in the original manuscript (mostly in Appendix files), (a) we examined many real datasets (whose number and diversity exceed what are usually used in algorithm papers) to evaluate these algorithms, (b) we developed three methods (i.e., two ways to build consensus networks and the use of spike-in data) to help validate network inference, (c) in the first example of the section "APPLICATIONS", we purposefully focused on inferring interactions between HLA genes, because these interactions have been well studied, and our results agree with experimental reports.

(2) During the revision, we have taken a systematic approach using the five non-small cell lung cancer cell lines and lung alveolar cells. First, we identified differentially expressed genes in each cell line against the alveolar cells upon different conditions. A typical set of conditions was (a) gene expression value >0.1, (b) gene expression > 50% cells, (c) fold change >0.3. Second, we applied GO analysis to the differentially expressed genes in each cell line using g:Profiler (https://biit.cs.ut.ee/gprofiler/gost) (parameters: Significance threshold=Benjamini-Hochberg FDR, User threshold=0.05, Data sources=KEGG and WikiPathways). We found that the WikiPathway and KEGG pathways "Metabolic reprogramming in colon cancer" (WP4290), "Pyrimidine metabolism" (WP4022), and "Nucleotide metabolism" (hsa01232) were commonly enriched in all cancer cell lines (Appendix file 4 gives the result of the cancer cell lines A549). We also performed GO analysis using the GSEA (gene set enrichment analysis) package, which identified the KEGG pathway "Non-small cell lung cancer" (hsa05223) as an enriched pathway. Third, we used the PC+DCC.γ to infer interactions among genes in the three pathways WP4290, hsa01232, and hsa05223 in the five cancer cell lines. Fourth, we also used GSP+DCC.γ and GES to infer interactions among genes in the WP4290 pathway.

(3) Numerous studies report that glucose metabolism is reprogrammed and nucleotide synthesis is increased in cancer cells. The increased glucose uptake and fermentation of glucose to lactate are known as the Warburg Effect and have been documented for over 90 years. Thus, we first examined and compared the WP4290 pathway in the five lung cancer cell lines and lung alveolar cells. The key features of the reprogrammed glucose metabolism in the cancer cell lines are (a) glucose intake is increased, (b) the glycolysis/TCA cycle intermediates are used for synthesizing nucleotide, (c) lactate generation is increased. The inferred networks capture these features (Appendix file 4). (a) Activations of SLC2A1 (which encodes a major glucose transporter and controls glucose intake), PGD (which promotes glucose metabolism into the pentose phosphate shunt), PSAT1 (which encodes a phosphoserine aminotransferase that catalyzes the reversible conversion of 3-phosphohydroxypyruvate to phosphoserine), and LDHA (whose protein catalyzes the conversion of pyruvate to lactate) are inferred in all cancer cell lines but not in alveolar cells. (b) Activations of genes by downstream genes are inferred; this sort of feedback is an important feature of metabolism. (c) In contrast, none of these features occur in the alveolar cells (partly because key genes such as SLC2A1 are not expressed in the alveolar cells). These inferred results are literature-supported (there are many Warburg Effect-related papers) and biologically reasonable, even though the absence of metabolites in the data flaws network inference (Appendix file 4).

(4) We next examined the "Pyrimidine metabolism" (WP4022) pathway. We used genes in hsa00240 ("Pyrimidine metabolism" defined in KEGG) to perform the inference (because WP4022 contains too many POLR gene families) and used the more readable WP4022 pathway to illustrate the results. Compared with glucose metabolism, pyrimidine metabolism has many reversible reactions, making interactions vary more significantly in cells (compared with WP4290). The following genes and reactions are notable. (a) TYMS catalyzes dUMP->dTMP unidirectionally toward DNA synthesis. (b) Tk1/2 catalyze Thymidine->dTMP and Deoxyuridine->dUMP toward DNA synthesis (while NT5C/E/M do the opposite). (c) DUT catalyzes dUTP->dUMP (note that dUMP is the substrate for TYMS). (d) TYMP catalyzes Thymidine->Thymine unidirectionally away from DNA synthesis. (e) ENTPD1/3 catalyze dTTP->dTDP->dTMP, UTP->UDP->UMP, and CTP->CDP->CMP away from DNA and RNA synthesis (but AK9/NME reverse these reactions). (f) NT5C/E/M catalyze dCMP->Deoxycytidine, dUMP-> Deoxyuridine, and dTMP->Thymidine away from DNA synthesis. In accordance with these, the following interactions were inferred from cancer cell lines. (a) TYMS (the key gene that promotes DNA synthesis) is activated (and is not repressed by any gene) in all cancer cell lines but not in alveolar cells. (b) Tk1/2 are activated in cancer cells and alveolar cells. (c) DUT is activated in all cancer cell lines but is not expressed in alveolar cells. (d) activations of TYMP (the critical gene that makes reactions away from DNA synthesis) by multiple others are inferred in alveolar cells. (e) ENTPD1/3 (genes making reactions away from DNA synthesis) are activated only in alveolar cells. (f) NT5C/E/M are repressed in all cancer cell lines but are not expressed in alveolar cells. The most notable may be DUT->Tk1 and DUT->TYMS in all cancer cell lines, indicating feedforward or coordinated regulations that promote DNA synthesis. These features are biologically reasonable, even though the absence of metabolites in the data flaws network inference and there are many reversible reactions (Appendix file 4).

(5) We further examined the "Non-small cell lung cancer" (hsa05223) pathway. We used the "graphite" R package to turn hsa05223 into an adjacency matrix and mapped the inferred causal network to the matrix. An inferred interaction is assumed to be mapped to hsa05223 if it can be mapped to an edge or a path (either forward, inverse, or undirected). About 30% of inferred interactions can be mapped to hsa05223. In addition, hsa05223 contains sub-pathways such as p53 signaling pathway and PI3K-AKT pathway, thus containing considerable unannotated epistatic interactions (Appendix file 4). Synergistic interactions are also unannotated in hsa05223, and many of epistatic and synergistic interactions are literature-supported (e.g., CDKN1A->BAX and EGFR->MET, see Wang et al. 2014 and Dong et al. 2018). We found that many inferred interactions can be mapped to epistatic and synergistic interactions in hsa05223. Taken together, in each cell line, about 50% of inferred interactions can be mapped to hsa05223 (Appendix file 4). These percentages are underestimated, because there are considerable feedback regulations (e.g., by transcription factors) in hsa05223 which are unannotated and we do not handle these interactions (e.g., RB/E2F1-related interactions such as E2F1->EGFR and RB1->ERBB2). If these interactions are annotated and handled, higher percentages of inferred interactions can be mapped to hsa05223. It is also worth noting that the annotated hsa05223 has defects because it is not detected as an enriched pathway by g:Profiler.

(6) We also used the GES and GSP+DCC.γ to repeat the analysis of the WP4290 pathway in the five lung cancer cell lines, and the same features of reprogrammed glucose metabolism were obtained (Appendix file 3).

(7) The above results are described briefly in the main text and in detail in the revised Appendix file 3 and 4. The reader may notice that inferred interactions vary to different extent in different cell lines. These cell lines have different genetic backgrounds, including different numbers of genes in WP4290 are expressed in different cell lines, and it has been recently found that cancer cells show great transcriptional plasticity (Househam et al. 2022).

Reviewer #1 (Recommendations for the authors):1. There seem to be three general categories of causal inference algorithms: constraint-based, score-based, and hybrid (see for example, https://proceedings.neurips.cc/paper/2017/file/275d7fb2fd45098ad5c3ece2ed4a2824-Paper.pdf).This paper appears to focus only on different implementations of constraint-based algorithms; however, it appears that score-based algorithms in general produce better results for complex biologically-driven datasets. I have checked this with experts since this is a little outside of my field. In my opinion, this is a major shortcoming that significantly reduces the impact of the current manuscript. In other words, the authors need to justify that the algorithm (or group of algorithms) they have picked is the best option for single-cell analyses (in doing so, the existing efforts and literature should be addressed). Alternatively (and especially if there is no consensus about what algorithm works best), they should incorporate other options into the tool.

(1) The NIPS-2017 paper (i.e., Wang et al. 2017) reports a hybrid algorithm (IGSP), which features in (a) combining a score-based method and CI tests, (b) based on GSP and being able to handle interventional data, (c) outperforming the representative score-based method GIES. We have examined the representative score-based method GES/GIES and the representative hybrid method GSP/IGSP ('I' means using interventional data) using small-scale simulated and real data (the datasets reported by Sachs et al. and of the lung cancer cell lines). Comparisons of GES, GSP+DCC.γ, and PC+DCC.γ based on these data suggest that GSP+DCC.γ and PC+DCC.γ have comparable performance and somewhat outperform GES, probably because GES assumes Gaussian distribution of data but single-cell data are complex.

(2) We have incorporated GES and GSP into our platform and briefly described the benchmarking of GES, GSP, and PC (Appendix file 3). The GSP package originally contains only two CI tests (i.e., GaussCItest and HSIC.γ); we now enable it to run with the 9 CI tests.

(3) Perturbations can be applied to single cells on the protein level (e.g., the data of Sachs et al. 2000), on the genome level (using the CRISPR/Cas9 system), and on the transcriptome level (using the CRISPR/dCas9 system). The latter two combine pooled CRISPR screen with scRNA-seq (reviewed by Boch et al. 2022). A revised method is direct-capture Perturb-seq, a versatile screening approach in which expressed sgRNAs are sequenced alongside single-cell transcriptomes (Replogle et al. 2020). We do not include GIES and IGSP in the current version of CausalCell because of unsolved issues, including (a) individual cells receive different gRNAs and are perturbed according to the gRNAs received by the cell, thus the numbers of cells targeted by different gRNAs vary substantially in an experiment, and (b) the data (all we examined were generated by 10X Genomics) are much sparser (see, for example, GSE146194) than normal 10X Genomics data.

(4) Please see also the responses to Essential revisions.

2. Some statements in the comparison section need clarification. For example, in Table 1, the authors mention that "Both ExtraTrees and RandomForest perform well, but XGBoost does not." I believe it is necessary to state more details about what the context/condition is, in order to justify such claims.

(1) We revised Table 1 and Table 2. A note is added to Table 1 and Table 2, respectively, explaining how the performance indicators are estimated and how advantage/disadvantage is determined. An important claim is the recommendation of 300 and 600 cells for Smart-seq and 10X Genomics data. We have explained why the two numbers are recommended in the main text and details are also given in Appendix file 3 and 4.

(2) Please see also the responses to Essential Revisions.

3. Figure 2: The comparison shows a relatively poor similarity between different algorithms. What is the reason for this?

(1) We now make it clear that Figure 2 shows that (a) kernel-based CI tests are better than non-kernel-based methods, (b) DCC.perm, DCC.γ, HSIC.perm, and HSIC.γ are more accurate than the other three kernel-based methods, (c) RCIT and RCoT are approximated versions of KCIT, and that is why they are poorer than DCC- and HSIC-based algorithms. Thus, it is not surprising that the comparison shows a relatively poor similarity between different algorithms.

(2) We have greatly revised the legends of Figure 2 and Appendix file 3-figure 2. We note that RCIT and RCoT are much faster than DCC- and HSIC-based algorithms and thus suitable for large-scale inference.

4. In Discussions, the authors state that kernel-based CI tests perform better than faster methods that make additional assumptions. I am not sure if I see the evidence for this. Especially when different kernel-based algorithms poorly agree with each other, it is unclear how a case can be made. Ideally, this issue can be addressed with a "mock" dataset in which the causal links are already known. In the absence of a known "ground truth" or a "gold standard", their arguments about accuracy and performance are in general less convincing.

(1) Figure 2 and Appendix file 3-figure 2 give the evidence. On *how a case can be made,* we explain in the revised legend of Figure 2 that the accuracy of PC+9 CI tests was evaluated with four steps. First, 9 causal networks were inferred using the 9 CI tests. Second, pairwise structural Hamming distances (SHD) between these networks were computed, which indicate that the networks of DCC.γ, DCC.perm, HSIC.γ, and HSIC.perm share the highest similarity. Third, a consensus network was built using the networks of the above four CI tests and was assumed to be closer to the ground truth than any network inferred by a single algorithm. Fourth, each of the 9 networks was compared with the consensus network.

(2) As mentioned above, RCIT and RCoT are poorer than DCC- and HSIC-based algorithms because the two are approximated versions of KCIT.

(3) We used multiple synthetic datasets to examine these algorithms but found that performance evaluation upon synthetic datasets is unreliable, probably because the defined relationships between variables poorly catch the real relationships between varied genes in varied cells. Appendix file 3-figure 1BD shows an example. That is why we chose the five lung cancer cell lines and alveolar cells to make 5-to-1 comparisons, and think this may be the best way one can use to estimate algorithms' performance.

(4) We have made more analyses during the revision using the five lung cancer cell lines and genes in specific pathways. Key features of interactions in these genes are well annotated.

5. Line 353: The authors make a statement about the dataset being "large enough". I'd suggest including a formal treatment, in which the causal inference results are compared as the number of data points increases (for example by sub-sampling one of the existing datasets). I believe this will provide more convincing evidence for how many data points are required and what to expect as that number increases/decreases.

We evaluated the impact of sample size on algorithms' performance by running the 9 algorithms (CI tests) using 200 and 300 H2228 cells. The results indicate that 200 cells generate poorer networks than 300 cells (Main-figure 2; Appendix file 3-figure 2). In both situations, complete kernel-based CI tests performed better than other CI tests. We also inferred interactions between genes in the "Metabolic reprogramming in colon cancer" (WP4290) pathway using 200, 400, 600, and 800 H838 cells (Appendix file 3-figure 3). The results indicate that more cells make more interactions be inferred, but the interactions with high significance are stable. Upon these results, we explain in the main text (the section "5. Causal discovery" and the section "Tips for best practices") that 300 and 600 cells for Smart-seq and 10X Genomics data are suitable for typical situations, that the required sample size depends on multiple factors, and that performing inference using large datasets is highly time-consuming.

Reviewer #2 (Recommendations for the authors):1) Reactome and KEGG databases curated directed links between genes or proteins, which is suitable for result evaluation.

Substantial analyses have been made using the data of the five lung cancer cell lines and alveolar cells and the genes in multiple pathways in the KEGG and WikiPathway databases. Please see the responses to Essential Revisions.

2) 50 feature genes represent a small fraction of the whole genome. This means CausalCell cannot fully take advantage of the high-throughput feature of current scRNA-seq technologies. Any suggestions to deal with this concern?

(1) Trade-offs between time consumption, network accuracy, and network size are inevitable. We also note that statisticians are working on improving the time consumption of kernel-based CI tests (for example, RCIT and RCoT are two approximated versions of KCIT) and that a researcher usually cares only about specific genes linked to a phenotype or disease.

(2) In most situations, the genes linked to a phenotype or disease are unknown. Thus, a tool should take advantage of the high-throughput feature of current scRNA-seq to identify such genes. Instead, inferring networks genome-wide is often not required.

(3) In the section “Tips for best practices”, we wrote, “a solution to fully taking advantage of the high-throughput feature of scRNA-seq data is to infer multiple networks with shared genes and then try to merge these networks”.

3) What techniques do the authors use to generate the consensus network?

We have added in the section "5. Causal discovery" that there are two ways to build a consensus network: using shared interactions inferred by different algorithms or using shared interactions inferred by running an algorithm for multiple times.

4) On lines 68-70, what are the differences among regulatory networks, causal networks, (ordinary) networks, and gene networks? Readers could be confused by such an introduction.

These methods were historically named and may confuse a reader. We feel this manuscript is not a proper place to explain the differences in detail and refer the reader to these reviews. We also use the quotation mark to cite them in the revised manuscript and briefly explain the difference between signaling networks and regulatory networks in the Discussion.

5) Figure 4 and other causal network plots should have a legend to indicate the color scheme for the upregulation or downregulation of gene expression.

A color bar is added.

6) Volume and page information of Reference [49] are missing.

The mistake is corrected.

References

1. Boch et al. High-content CRISPR screening. Nat Rev. Methods Primers. 2022, 2, 8.

2. Chickering, D.M. Optimal structure identification with greedy search. Journal of Machine Learning Research 2003, 3, 507-554.

3. Dong et al. EGFR and c-MET cooperate to enhance resistance to PARP inhibitors in hepatocellular carcinoma. Cancer Res. 2918, 79, 819–829.

4. Househam et al. Phenotypic plasticity and genetic control in colorectal cancer evolution. Nature 2022, 611, 744-753.

5. Replogle et al. Combinatorial single-cell CRISPR screens by direct guide RNA capture and targeted sequencing. Nature Biotechnology 2020, 38, 954-961.

6. Sachs et al. Causal protein-signaling networks derived from multiparameter single-Cell Data. Science 2005, 308, 523-529.

7. Solus et al. Consistency guarantees for greedy permutation-based causal inference algorithms. Biometrika 2021, 108, 795–814.

8. Tran et al. A benchmark of batch-effect correction methods for single-cell RNA sequencing. Genome Biology 2020, 21, 12.

9. Wang et al. BAX and CDKN1A polymorphisms correlated with clinical outcomes of gastric cancer patients treated with postoperative chemotherapy. Medical Oncology 2014, 31, 249.

10. Wang et al. Permutation-based causal inference algorithms with interventions. NIPS 2017.

[Editors' note: further revisions were suggested prior to acceptance, as described below.]

The manuscript has been improved but there are some remaining issues that need to be addressed, as outlined below:1. The need to include more recent advances in causal discovery, such as continuous optimization-based, neural network-based methods is brought up by the reviewers again. There is a need to either include these approaches or offer concrete evidence that the PC algorithm works better (or at least equally well) for the type of biological questions the tool is addressing. This is perhaps best done by a direct comparison of results between the methods included in CausalCell and some other representatives such as Notears, Golem, DAG-GNN, lingam, and Dlingam.

(1) Among the various causal discovery methods, we think that PC + kernel-based CI tests may best suit various situations, including missing values in data, different distributions of data, noise in data, and complex relationships between genes. Therefore, we initially focused on identifying what kernel-based CI tests perform best. Agreeing with theoretical studies (see the PhD dissertation of Patra Verbyla and other related papers), we found that DCC and HSIC-based CI tests outperform others. Now, we have benchmarked eight new methods (including the continuous optimization-based NOTEARS and its follow-up methods), together with two new CI tests, using simulated and real data. In the benchmarking, these methods are called NOTEARS-linear (Zheng X et al. 2018), NOTEARS-nonlinear (Zheng X et al. 2020), GOLEM (Ng I et al. 2021), DAGMA-linear (Bello K et al. 2022), DAGMA-nonlinear (Bello K et al. 2022), ICALiNGAM (Ikeuchi T et al. 2023; Shimizu et al. 2006), DirectLiNGAM (Ikeuchi T et al. 2023; Shimizu et al. 2011), and DAG-GNN (Yu Y et al. 2019), and the CI tests are called GCM (Shah R, Peters J 2019) and KRESIT (Zhang Q et al. 2019). Simulated data are generated using the methods described in NOTEARS-linear and NOTEARS-nonlinear. Real data are the scRNA-seq data of five non-small cell lung cancer (NSCLC) cell lines (A549, H2228, H838, HCC827, H1975), which have been used widely to evaluate scRNA-seq data analysis algorithms.

(2) Here we describe seven cases of benchmarking. First, we used the method in NOTEARS-linear to simulate 600 samples of 10 variables. In this case, continuous optimization-based methods (especially the linear versions) perform very well (Author response table 1). Since the time consumption of PC+KRESIT is more than an order of magnitude higher than other methods but the accuracy of PC+KRESIT’s network is low, we excluded KRESIT in the following benchmarking.

**Author response table 1. sa2table1:** The results of 600 samples of 10 variables in linear relationships (* Dual Intel(R) Xeon(R) CPU E7-4820 v4 @ 2.00GHz，256G RAM).

Methods	FDR	TPR	FPR	SHD	Precision	Recall	F1	#Edges	Time*
NOTEARS_linear	0.00	0.95	0.00	1	1.00	0.95	0.97	19	13.93
NOTEARS_nonlinear	0.17	0.75	0.12	6	0.83	0.75	0.79	18	505.72
GOLEM	0.00	1.00	0.00	0	1.00	1.00	1.00	20	21.89
ICALiNGAM	0.16	0.80	0.12	5	0.84	0.80	0.82	19	0.74
DirectLiNGAM	0.52	0.55	0.48	19	0.48	0.55	0.51	23	0.38
DAGMA_linear	0.00	1.00	0.00	0	1.00	1.00	1.00	20	4.59
DAGMA_nonlinear	0.15	0.55	0.08	10	0.85	0.55	0.67	13	508.95
DAG_GNN	0.05	1.00	0.04	1	0.95	1.00	0.98	21	522.86
PC+DCC.γ	0.17	0.50	0.08	10	0.71	0.50	0.59	12	296.28
PC+Gauss	0.13	0.65	0.08	8	0.72	0.65	0.68	15	0.20
PC+GCM	0.00	0.70	0.00	6	0.70	0.70	0.70	14	1515.40
PC+KRESIT	0.36	0.35	0.16	13	0.54	0.35	0.42	11	20432.41

Second, we benchmarked these methods using simulated 600 samples of 10 variables in nonlinear relationships. Data were simulated using the method in NOTEARS-nonlinear. In this case, nonlinear versions of continuous optimization-based methods (NOTEARS-nonlinear and DAGMA-nonlinear) perform better than others. But, no method reaches TPR=1.0 (Author response table 2).

**Author response table 2. sa2table2:** The results of 600 samples of 10 variables in nonlinear relationships.

Methods	FDR	TPR	FPR	SHD	Precision	Recall	F1	#Edges	Time
NOTEARS_linear	0.00	0.30	0.00	14	1.00	0.30	0.46	6	3.18
NOTEARS_nonlinear	0.00	0.80	0.00	4	1.00	0.80	0.89	16	90.91
GOLEM	0.55	0.45	0.44	21	0.45	0.45	0.45	20	21.83
ICALiNGAM	0.33	0.30	0.12	17	0.67	0.30	0.41	9	0.76
DirectLiNGAM	0.14	0.30	0.04	14	0.86	0.30	0.44	7	0.40
DAGMA_linear	0.00	0.30	0.00	14	1.00	0.30	0.46	6	3.53
DAGMA_nonlinear	0.00	0.80	0.00	4	1.00	0.80	0.89	16	257.68
DAG_GNN	0.33	0.30	0.12	16	0.67	0.30	0.41	9	485.38
PC+DCC.γ	0.28	0.65	0.20	9	0.684	0.65	0.67	18	369.64
PC+Gauss	0.31	0.55	0.20	11	0.65	0.55	0.60	16	0.17
PC+GCM	0.14	0.60	0.08	9	0.86	0.60	0.70	14	1036.18

Third, we evaluated these methods by inferring the network of 49 genes in the "non-small cell lung cancer" (hsa05223) KEGG pathway using 600 A549 cells (note that A549, H2228, H838, HCC827, H1975 are non-small cell lung cancer cell lines). We used the graphite package in Bioconductor to derive the DAG of the human hsa05223 pathway and used the DAG as the ground truth to evaluate the accuracy of inferred networks (Author response image 1). In this case, PC+DCC.γ performs best, PC+Gauss generates too many edges (see also Appendix 3 figure 2), most continuous optimization-based methods infer no or few edges, and most edges inferred by GOLEM are wrong (Author response table 3; Author response image 2).

**Author response table 3. sa2table3:** The results of 600 samples of 10 variables in nonlinear relationships.

Methods	FDR	TPR	FPR	SHD	Precision	Recall	F1
NOTEARS_linear	-	-	-	-	-	-	-
NOTEARS_nonlinear	-	-	-	-	-	-	-
GOLEM	0.97	0.09	0.03	127	0.029	0.010	0.015
ICALiNGAM	-	-	-	-	-	-	-
DirectLiNGAM	-	-	-	-	-	-	-
DAGMA_linear	-	-	-	-	-	-	-
DAGMA_nonlinear	-	-	-	-	-	-	-
DAG_GNN	-	-	-	-	-	-	-
PC+DCC.γ	0.92	0.15	0.07	162	0.085	0.071	0.077
PC+Gauss	0.94	0.31	0.36	451	0.057	0.232	0.091

**Author response image 1. sa2fig1:** The DAG of the hsa05223 "non-small cell lung cancer" pathway.

**Author response image 2. sa2fig2:** The inferred networks of the 49 genes in the hsa05223 pathway by the ten methods in A549 cells. (A) The network inferred by GOLEM. (B) The network inferred by PC+DCC.γ. (C) The network inferred by PC+GaussCItest.

The poor performance of continuous optimization-based methods on real data may be due to (a) missing values in scRNA-seq data, (b) inference with missing variables (i.e., the 49 genes are an incomplete causal model), (c) complex relationships between genes, and (d) noise in data. Note that many relationships between the 49 hsa05223 genes should be indirect because some genes remain unannotated in the pathway and not all annotated genes are highly expressed in these cells.

We next evaluated the power of these methods to infer indirect relationships between genes. To perform the fourth case of benchmarking, we first performed pathway enrichment analysis using the g:Profiler program to identify enriched pathway genes in NSCLC cells. A small and well annotated pathway is the pentose phosphate pathway WP134 (“WP” indicates pathways defined in the WikiPathways database). Branching from glycolysis at the first committed step of glucose metabolism, the pentose phosphate pathway is critical for cancer cells because it generates pentose phosphates to supply their high rate of nucleic acid synthesis and provides NADPH that is required for both the synthesis of fatty acids and cell survival under stress (Patra and Hay, 2014). This pathway is a potential target for cancer therapy (Cho et al., 2018; Ghanem et al., 2021). Since the five enriched WP134 genes G6PD, PGD, TKT, TALDO1, PGLS in A549 cells do not encode transcription factors, we used BAHSIC as the method and the five genes as the response variables to perform feature selection to identify these genes' potential transcription factors. The top two transcription factors are ENO1 and NPM1. We then used the seven genes and 600 A549 cells to infer the regulatory network. The result is notable in that (a) some continuous optimization-based methods (e.g., DAGMA-linear and DAGMA-nonlinear) can infer edges this time (Author response table 4), (b) the inferred edges by these methods are mainly from the transcription factor ENO1 and NPM1 to their target genes (the network of GOLEM is quite unreasonable) (Author response image 3). The result suggests that direct interactions are easier to be inferred and that DAGMA-nonlinear can perform well when there are direct interactions between genes.

**Author response table 4. sa2table4:** The results of 600 A549 cells, 5 enriched WP134 genes, and 2 of their transcription factors.

Methods	FDR	TPR	FPR	SHD	Precision	Recall	F1	#Edge	Time
NOTEARS_linear	-	-	-	-	-	-	-	-	0.01
NOTEARS_nonlinear	-	-	-	-	-	-	-	-	0.61
GOLEM	0.83	0.08	0.56	11	0.17	0.08	0.11	6	19.53
ICALiNGAM	0.00	0.33	0.00	8	1.00	0.33	0.50	4	0.58
DirectLiNGAM	0.50	0.08	0.11	11	0.50	0.08	0.14	2	0.14
DAGMA_linear	0.00	0.17	0.00	10	1.00	0.17	0.29	2	3.17
DAGMA_nonlinear	0.56	0.33	0.56	10	0.44	0.33	0.38	9	374.06
DAG_GNN	-	-	-	-	-	-	-	-	546.36
PC+DCC.γ	0.40	0.50	0.44	8	0.50	0.50	0.50	10	234.71
PC+Gauss	0.38	0.42	0.33	8	0.63	0.42	0.50	8	0.14
PC+GCM	0.45	0.50	0.56	7	0.50	0.50	0.50	11	1056.60

**Author response image 3. sa2fig3:** Some inferred networks of the WP134 pathway in the A549 cells. (A) The network of PC+DCC.γ. (B) The network of PC+GCM. (C) The network of DAGMA-linear. (D) The network of DAGMA-nonlinear. (E) The network of GOLEM, (F) The network of ICLiNGAM.

To examine the conjecture continuous optimization-based methods perform better when there are direct interactions between genes, we performed the fifth case of benchmarking to examine only the enriched hsa05223 genes. We used the GSEA program to identify 14 highly and differentially expressed hsa05223 genes in H2228 cells compared with the lung alveolar cells (see Appendix 1 table 1) (AKT1, BAD, BAX, CCND1, CDK4, EGFR, EML4, GRB2, HRAS, KIF5B, KRAS, MAP2K2, MET, RXRA). Then, we used the ten methods and 600 H2228 cells to infer the networks of the 14 genes. This time, all continuous optimization-based methods perform worse (worse than the case of 49 hsa05223 genes). However, PC+DCC.γ still performs well (Author response image 4), as the network of PC+DCC.γ maps well the hsa0523 pathway (Figure 1; Appendix 4 figure 14). This suggests that PC+DCC.γ can perform causal discovery with missing variables.

**Author response image 4. sa2fig4:** The networks of the 14 enriched hsa05223 genes inferred by the ten methods. Most continuous optimization-based methods do not infer relationships between genes. In the network of PC+DCC.γ, solid and dashed red edges map direct and indirect interactions in the hsa05223 pathway and red edges with paper citations mark relationships supported by experimental findings. Edges between BAX, CDK3, and CCND1 (functioning for uncontrolled proliferation and driving cell division) are unannotated in hsa05223, but some of these relationships may be true because they reflect feedback regulations in cancer cells.

A challenge when benchmarking methods with scRNA-seq data is identifying or building a reasonable ground truth. Working also on causal hypothesis testing, we find that systematically analyzing accepted and rejected causal hypotheses can help build approximations to the ground truth of causal relationships between genes. Author response image 5 shows an approximate ground truth for the 14 enriched hsa05223 genes built upon causal hypothesis testing, and we find that the PC+DCC.γ network maps the approximate ground truth well.

**Author response image 5. sa2fig5:** An approximate ground truth for the 14 enriched hsa05223 genes in H2228 cells built upon causal hypothesis testing using gene expression data in H2228 cells.

To further confirm that the poor performance of continuous optimization methods on real data is due greatly to missing variables (inference with an incomplete model), we simulated nonlinear relationships between 20 variables using the method of NOTEARS-nonlinear and inferred the relationships between 18 variables. In the first round of inference X8 and X20 were removed from the list of variables, and in the second round of inference X4 and X7 were removed from the list of variables. The results indicate that PC+DCC.γ identifies more indirect relationships between the 18 variables than DAGMA-nonlinear in both rounds (Author response image 6).

**Author response image 6. sa2fig6:** DAGMA-nonlinear and PC+DCC. γ show different ability to infer indirect causal relationships from simulated data. Red and blue edges indicate mapped direct relationships between variables with different direction, dashed edges indicate mapped indirect relationships between variables (e.g., X11X4 in panel (B) maps X11X8X4 in the true DAG). When an edge maps both direct and indirect relationships, it is assumed to map the direct one. We do not care about the edges' orientation when checking whether it maps any edge in the true DAG. (A) The true DAG of the nonlinear relationships between the 20 variables. (B) The network of DAGMA-nonlinear without X8 and X20, in which two edges map indirect relationships. (C) The network of PC+DCC.γ without X8 and X20, in which five edges map indirect relationships. (D) The network of DAGMA-nonlinear without X4 and X7, in which four edges map indirect relationships. (E) The network of PC+DCC.γ without X4 and X7, in which six edges map indirect relationships.

Finally, we examined whether these methods are sensitive to missing values in data. We randomly removed 10% and 20% of values per variable from the simulated data that describe nonlinear relationships between 10 variables (see Author response table 2) and performed causal discovery. The results show that the performance of almost all methods decreased when values were randomly removed (compared (Author response table 5) and Author response table 6 with Author response table 2), especially in terms of TPR and SHD. The performance of DAGMA-nonlinear decreased drastically, as it performs well not only when there is no missing value in simulated data (Author response table 2) but also on some real datasets (Author response image 3). To discovery why DAGMA-nonlinear showed such a difference, we examined the percentages of missing values for the 7 WP134 genes in the A549 dataset (DAGMA-nonlinear performs well) and the percentages of missing values for the 14 hsa05223 genes in the H2228 dataset (DAGMA-nonlinear performs poorly). The percentages of missing values are 0% (ENO1), 0% (NPM1), 0% (G6PD), 5.5% (PGLS), 0.17% (PGD), 0.17% (TALDO1), and 0% (TKT) in the A549 dataset, and 42.2% (AKT1), 12.3% (BAD), 30.2% (BAX), 13% (CCND1), 0% (CDK4), 20.2% (EGFR), 27.3% (EML4), 28.3% (GRB2), 30.8% (HRAS), 14.4% (KIF5B), 32.2% (KRAS), 27.5% (MAP2K), 20.7% (MET), and 37.3% (RXRA) in the H2228 dataset. The differences in percentages of missing values reveal why DAGMA-nonlinear performs well when there are no or few missing values, but poorly otherwise.

**Author response table 6. sa2table6:** The results of 600 samples of 10 variables in nonlinear relationships, with 20% missing values.

Methods	FDR	TPR	FPR	SHD	Precision	Recall	F1	#Edges	Time
NOTEARS_linear	0.20	0.2	0.04	17	0.80	0.2	0.32	5	1.66
NOTEARS_nonlinear	0.60	0.8	0.96	25	0.40	0.8	0.53	40	77.95
GOLEM	0.71	0.3	0.6	24	0.29	0.3	0.29	21	14.98
ICALiNGAM	0.50	0.1	0.08	20	0.50	0.1	0.17	4	0.07
DirectLiNGAM	0.25	0.15	0.04	18	0.75	0.15	0.25	4	0.23
DAGMA_linear	0.43	0.2	0.12	19	0.57	0.2	0.30	7	1.91
DAGMA_nonlinear	-	-	-	-	-	-	-	0	266.44
DAG_GNN	0.60	0.1	0.12	20	0.40	0.1	0.16	5	182.71
PC_dcc_γ	0.44	0.25	0.16	18	0.45	0.25	0.32	11	49.63
PC_gauss	0.70	0.15	0.28	19	0.27	0.15	0.19	11	0.07
PC_GCM	0.00	0.25	0	15	0.71	0.25	0.37	7	48.18

**Author response table 5. sa2table5:** The results of 600 samples of 10 variables in nonlinear relationships, with 10% missing values.

Methods	FDR	TPR	FPR	SHD	Precision	Recall	F1	#Edges	Time
NOTEARS_linear	0.4	0.15	0.08	19	0.6	0.15	0.24	5	1.60
NOTEARS_nonlinear	0.56	0.8	0.8	21	0.44	0.8	0.57	36	97.66
GOLEM	0.52	0.55	0.48	19	0.48	0.55	0.51	23	14.75
ICALiNGAM	0.25	0.15	0.04	18	0.75	0.15	0.25	4	0.04
DirectLiNGAM	0.4	0.15	0.08	19	0.6	0.15	0.24	5	0.23
DAGMA_linear	0.43	0.2	0.12	19	0.57	0.2	0.30	7	1.88
DAGMA_nonlinear	0	0.15	0	17	1	0.15	0.26	3	277.04
DAG_GNN	0.67	0.15	0.24	23	0.33	0.15	0.21	9	215.29
PC_dcc_γ	0.22	0.35	0.08	14	0.64	0.35	0.45	11	82.42
PC_gauss	0.5	0.3	0.24	18	0.5	0.3	0.38	12	0.08
PC_GCM	0.29	0.25	0.08	17	0.5	0.25	0.33	10	68.43

(3) The above results suggest that continuous optimization-based methods (especially DAGMA-nonlinear) perform very well if missing values and missing variables are not serious. We have incorporated DAGMA-nonlinear into CausalCell.

2. A more thorough comparison of independence/CI tests with more recent methods such as regression-based, ranking-based, and deep neural network-based CI tests (e.g., MLP-based and GAN-based) needs to be included.

(1) (a) We have examined multiple other CI tests, including the regression-based tests SCIT and KRESIT (Zhang H et al. 2022; Zhang Q et al. 2017), the GAN-based test GCIT (Bellot A, van der Schaar M. 2019), and the generalized covariance measure-based GCM (which is also classified as regression-based) (Shah R, Peters J. 2019). (b) We ran PC+SCIT using the authors' MATLAB code and found that the inferred networks are skeletons (as the authors wrote in their paper, "In the experiments above, we compare SCIT and ReCIT in terms of learning causal skeletons of small DAGs"). Also, since SCIT is much faster than KCIT, but Type I error is comparable to KCIT, SCIT may not outperform HSIC and DCC. (c) We compared PC+KRESIT and PC+GCM with PC+DCC.γ and PC+GaussCItest, together with the other eight new causal discovery methods. We found that PC+KRESIT is 69 times slower than PC+DCC.γ and underperforms PC+DCC.γ in terms of accuracy (Author response table 1). PC+GCM performs slightly poorer than PC+DCC.γ (Author response image 3, Author response image 7) and is also slower (Tables 1, 2).

**Author response image 7. sa2fig7:** The network inferred by PC+GCM without X4 and X7 (left) and X8 and X20 (right). Compared with the network of PC+DCC.γ (Author response image 6), PC+GCM infers fewer edges.

(2) (a) Based on the paper of Azadkia and Chatterjee (Azadkia M, Chatterjee S. 2021), Shi and Han indicate that CI test using the Azadkia–Chatterjee coefficient is inefficient (Shi H et al. 2022), and Lin and Han propose a revised method to overcome the rate-inefficiency (Lin Z, Han F. 2021). But, Lin and Han do not give an implementation of their revised method. (b) Both Figure 2 in Bellot and van der Schaar (2019) and our running of GCIT suggest that the set of conditional variables should not be empty. Thus, the current GCIT cannot work with PC, as it cannot handle the 0-order situation. (c) We used Google to search the internet, with the keywords "multilayer perceptron conditional independence" and "mlp conditional independence". The top and most relevant papers are "Shen Y et a. Conditional independence in Testing Bayesian Networks. Proc. Machine Learning Res. 2019; Shrivastava H et al. A deep learning approach to recover conditional independence graphs. NeurIPS 2022; Duong B, Nguyen T. Conditional independence testing via latent representation learning. arXiv 2022". Shen Y et al. examined CI in a specific class of Bayesian network (Testing Bayesian Network). The uGLAD test reported by Shrivastava H et al. assumes that the input comes from an underlying multivariate Gaussian distribution. The LCIT test reported by Duong B and Nguyen T also demands that the set of conditional variables should not be empty. Therefore, we did not examine these CI tests.

Summary:

(a) Using conditional independence of variables to infer the underlying causal network does not require specific assumptions about the functional forms of causal relationships or distribution of data. But, this approach only infers a set of equivalent networks. Other methods make some assumptions, for example, a differentiable score function. The new methods (including the continuous optimization-based) recently reported raise two questions – which ones perform well when being applied to scRNA-seq data, and whether the inferred networks are reliable. The above results suggest that continuous optimization-based methods (especially DAGMA-nonlinear) perform well if the inference is without missing variables and the data has no missing values, but otherwise not, and that PC + kernel-based CI tests (especially DCC.γ) can tolerate inference with missing variables and missing values in scRNA-seq data. (b) Missing values in scRNA-seq have drawn researchers’ attention, but inference with missing variables may present a more serious problem for causal discovery. (c) The benchmarking results suggest that GCM is also a good CI test. (d) As inferred causal networks help the researcher better draw causal hypotheses, it is sensible to use causal hypothesis testing to evaluate inferred networks. We develop a method using causal hypothesis testing to build approximate ground truths of causal relationships between variables. By comparing inferred networks with two kinds of approximate ground truth – the KEGG pathways and the approximate ground truth built upon causal hypothesis testing, one can better ensure the reliability of causal discovery (especially by PC+DCC.γ). (e) We have incorporated DAGMA-nonlinear and GCM into CausalCell. (f) The above results allow us to look at causal discovery methods from a more balanced perspective, and we have revised the main text and Appendix 3 significantly.

Additionally, please address the more detailed issues raised by Reviewer #3 below.Reviewer #3 (Recommendations for the authors):This work developed a workflow and platform for effectively performing causal discovery from scRNA‐seq data. The workflow/platform is developed upon the benchmark of 9 feature selection algorithms, 3 causal discovery methods, 9 CI tests, and the analyses of multiple datasets. The authors suggest that kernel-based conditional independence tests generate reliable results. Some key issues are discussed, and tips for best practices are provided. In my opinion, this work has the potential to help biologists discover some causal relationships among single-cell data, but the main drawback of this approach is the lack of new technologies on causal discovery as well as CI tests, and the biological significance of the work is not quite clear. Following are some of my concerns or questions about this work:1. In recent years, continuous optimization-based methods have become the most popular method for causal discovery, which yield much better performance than the PC algorithm, I think this work should discuss and take some representative continuous optimization-based methods into account, such as Notears, Golem, DAG-GNN. There are also some causal functional model-based methods such as lingam and Dlingam should be discussed.

(1) Many thanks for the valuable comments. We have used both simulated (using the methods described in NOTEARS-linear and NOTEARS-nonlinear) and real data (the scRNA-seq data of the non-small cell lung cancer cell lines) to evaluate the accuracy and time consumption of multiple methods. The results are presented in the responses to the editor's first suggestion. Evaluating the accuracy of networks inferred using scRNA-seq data is challenging because of the problems of missing values and missing variables and the lack of ground truth. Upon our recent work on causal hypothesis testing, we develop a method to reasonably build an approximation to the ground truth of a set of genes in specific cells. We use two kinds of approximate ground truth – KEGG pathways and the approximation to ground truth built upon causal hypothesis testing – to evaluate inferred networks. We also use gene annotations and reported gene functions to evaluate inferred networks.

(2) We examined the representative continuous optimization-based method NOTEARS (Zheng X et al. 2018) and its follow-up methods, including NOTEARS-nonlinear (Zheng X et al. 2020), GOLEM (Ng I et al. 2021), DAGMA-linear (Bello K et al. 2022), and DAGMA-nonlinear (Bello K et al. 2022). We also examined the LiNGAM-based methods, including ICALiNGAM (Ikeuchi T et al. 2023; Shimizu et al. 2006) and DirectLiNGAM (Ikeuchi T et al. 2023; Shimizu et al. 2006). Some of these methods perform very well on simulated data but poorly on real data. We identified that missing variables (inference with incomplete models) and missing values in data are two causes of poor performance of continuous optimization-based methods.

2. The 9 independence/CI tests might not stand for the state-of-the-art, more kinds of, and more recent methods should be taken into account, such as regression-based, ranking-based, and deep neural network-based CI tests (e.g., MLP-based & GAN-based).

We have examined more CI tests (please see the responses to the editor's second suggestion). The results suggest that GCM is also a good one and we have added the GCM test into the platform.

3. Theoretically, not all causal directions can be discovered by the orientation step of the PC algorithm, how to address the Markov equivalence classes in this paper? And there are also some other constraint-based methods, why choose PC?

(1) We chose PC for three reasons. (a) It is a well-recognized constraint-based method. (b) We actually examined FCI in the study, and found that, as FCI infers more kinds of interactions than PC does, the inferred causality is too complex for biologists to serve as reliable leads for subsequent experimental studies.

(2) There is a trade-off between performing inference without assumptions about the functional forms of causal relationships or distribution of data and distinguishing networks of the same Markov equivalence class. As the above benchmarking results show (Author response image 2, 4, 6), PC+DCC.γ is bad at addressing the causal Markov assumption but good at addressing the causal sufficiency assumption. Now the users can make the most of PC+DCC.γ and DAGMA-nonlinear to better perform causal discovery.

(3) We have discussed the two issues in the revised Discussion.

4. I suggest the author present the time complexity of each method not just 'time consumption', as it seems a little bit confusing. For example, HSIC.gmma should work much faster than HSIC.perm, but they are both '*' at 'time consumption'.

(1) Indeed, time complexity and time consumption are different concepts. We did not describe the accurate time complexity because (a) the time complexity of some algorithms may be too complex for ordinary biologists, (b) the time complexity of some algorithms is empirical (e.g., an estimated upper-bound) instead of theoretical, (c) time complexity does not accurately indicate time consumption because the latter also depends on sample size and parameters (e.g., the size of the conditional set).

(2) We have added time complexity in Appendix 3.

5. How to perform PC with HSIC? only 0-order CI test?

(1) HSIC is a measure of dependency between two variables. Petras Verbyla developed a method to apply HSIC to CI test. Performing two extra transformations can determine if X and Y are conditionally independent given the conditioning set Z: first, performing nonlinear regressions for X and Z and for Y and Z, respectively, to generate the residuals Xresid and Yresid based on Z; then, calculating HSIC(Xresid, Yresid) that indicates whether X and Y are conditionally independent given the conditioning set Z (X⫫Y|Z) (Verbyla P et al. 2017) (see the kpcalg package). This strategy is also used in the causaldag package (https://github.com/uhlerlab/conditional_independence/blob/main/conditional_independence/ci_tests/nonparametric/hsic.py#L114).

6. See "……However, because most algorithms have been designed for handling limited variables and few algorithms have been evaluated using real data, applying causal discovery to single-cell data remains challenging……". I don't quite agree with this statement, because there are lots of constraint-based methods (or combining with feature selections as dimension reduction step) with different CI tests for causal discovery on RNA-sq/Microarray, and this work also does a similar or same job.

(1) The sample sizes of almost all microarray and RNA-seq experiments are <100 per tissue. In addition, a tissue sample (e.g., a lump of cancer tissue) contains varied cells. Thus, the reliability of inferring causal networks from microarray data and RNA-seq data is questionable. One specific study reporting very large RNA-seq samples was performed by Maathuis M et al. (Maathuis M et al. 2010). This study was specific in that RNA-seq was applied to the single-cell organism yeast. Basically, only scRNA-seq can accurately detect gene expression in hundreds of isolated single cells and generate data for reliable causal discovery.

(2) We have deleted this sentence.

7. See "……These features of CI tests enable causation between any genes and molecules to be inferred……". It should be noted that not all causation between them can be inferred by CI tests.

This sentence has been revised.

8. " ……the time consumption of kernel-based CI tests disallows large-scale network inference……", how about the parallel PC? And sometimes one can limit the size of the conditional set.

(1) We indeed use the parallel PC, which allows multiple tasks to be run simultaneously.

(2) Limiting the size of the conditional set is a good point. We have now set the size of the conditional set as a parameter.

9. Actually, I wonder whether this workflow/platform can find some interesting biological results (say biomarkers) from the data. This is related to the biological significance of the work. Computational results in the paper do not provide convincing support for this point.

(1) We analyzed many scRNA-seq datasets (see Appendix files). We previously found that some inferred interactions are difficult to interpret. As we use two kinds of approximate ground truth to evaluate inferred networks in this round of revision, we find that (as Tables and Figures in the Responses-To-Reviewers show) many inferred relationships (especially by PC+DCC.γ) are quite reliable but are indirect causal relationships between genes.

(2) Instead of stressing the novelty and significance of our findings, we hope the tips for best practices are helpful. To our knowledge, no such tips have been reported.

References

1. Azadkia M, Chatterjee S. A simple measure of conditional dependence. Ann. Statist 2021.

2. Bello K et al. DAGMA: Learning DAGs via M-matrices and a Log-Determinant Acyclicity Characterization. NeurIPS 2022.

3. Bellot A, van der Schaar M. Conditional Independence Testing using Generative Adversarial Networks. NeurIPS 2019.

4. Cho ES et al. The Pentose Phosphate Pathway as a Potential Target for Cancer Therapy. Biomol Ther (Seoul) 2018.

5. Dragoj M et al. Association of CCND1 overexpression with KRAS and PTEN alterations in specific subtypes of non-small cell lung carcinoma and its influence on patients' outcome. Tumour Biol. 2015.

6. Ghanem N et al. The Pentose Phosphate Pathway in Cancer: Regulation and Therapeutic Opportunities. Chemotherapy 2021.

7. Ikeuchi T et al. Python package for causal discovery based on LiNGAM. Journal of Machine Learning Research 2023.

8. Lin Z, Han F. On boosting the power of Chatterjee's rank correlation. arXiv 2021.

9. Maathuis MH et al. Predicting causal effects in large-scale systems from observational data. Nature Methods 2010.

10. Ng I et al. On the Role of Sparsity and DAG Constraints for Learning Linear DAGs. arXiv 2021.

11. Patra KC, Hay N. The pentose phosphate pathway and cancer. Trends Biochem Sci. 2014.

12. Rao G et al. Inhibition of AKT1 signaling promotes invasion and metastasis of non-small cell lung cancer cells with K-RAS or EGFR mutations. Sci. Rep. 2017.

13. Sasaki H et al. CCND1 messenger RNA expression is correlated with EGFR mutation status in lung cancer. Clin Lung Cancer. 2007.

14. Shah R, Peters J. The hardness of conditional independence testing and the generalized covariance measure. Annals of Statistics 2019

15. Shi H et al. On Azadkia–Chatterjee’s conditional dependence coefficient. arXiv 2022.

16. Shimizu S et al. DirectLiNGAM: A direct method for learning a linear non-Gaussian structural equation model. Journal of Machine Learning Research 2011.

17. Shimizu S et al. A linear nonGaussian acyclic model for causal discovery. Journal of Machine Learning Research 2006.

18. Verbyla P et al. Exploiting general independence criteria for network inference. bioRxiv 2017.

19. Yu Y et al. DAG-GNN: DAG Structure Learning with Graph Neural Networks. arXiv 2019.

20. Zhang R et al. RXRα provokes tumor suppression through p53/p21/p16 and PI3K-AKT signaling pathways during stem cell differentiation and in cancer cells. Cell Death Dis. 2018.

21. Zhang H et al. Residual Similarity Based Conditional Independence Test and Its Application in Causal Discovery. AAAI 2022.

22. Zhang Q et al. Feature-to-feature regression for a two-step conditional independence test. UAI 2017.

23. Zheng X et al. DAGs with NO TEARS: Continuous Optimization for Structure Learning. arXiv 2018.

24. Zheng X et al. Learning Sparse Nonparametric DAGs. arXiv 2020.Zheng X et al. Learning Sparse Nonparametric DAGs. arXiv 2020.